# Foundation Molecular Grammar: Multi-Modal Foundation Models Induce Interpretable Molecular Graph Languages

Michael Sun [1]   Weize Yuan [2]   Gang Liu [3]   Wojciech Matusik [1]   Jie Chen [4]

## Abstract

Recent data-efficient molecular generation approaches exploit graph grammars to introduce interpretability into the generative models. However, grammar learning therein relies on expert annotation or unreliable heuristics for algorithmic inference. We propose Foundation Molecular Grammar (FMG), which leverages multi-modal foundation models (MMFMs) to induce an interpretable molecular language. By exploiting the chemical knowledge of an MMFM, FMG renders molecules as images, describes them as text, and aligns information across modalities using prompt learning. FMG can be used as a drop-in replacement for the prior grammar learning approaches in molecular generation and property prediction. We show that FMG not only excels in synthesizability, diversity, and data efficiency but also offers built-in chemical interpretability for automated molecular discovery workflows. Code is available at https://github.com/shiningsunnyday/induction.

## 1. Introduction

Substructure-based graph grammar is a data-efficient approach for molecular generation, offering significant advantages in terms of interpretability and performance (Guo et al., 2022b;a; Sun et al., 2024) compared to directly training on base molecular formal languages like SMILES and SELFIES. However, learning a graph grammar is challenging, as the resulting production rules are rarely guaranteed to be interpretable, e.g. containing functional groups. Interpretable approaches typically require expert annotation to construct the vocabulary of substructures (Sun et al., 2024), which is time-intensive and resource-demanding. One open-ended direction is to leverage MMFMs for this task, but their potential for advanced chemistry reasoning has been limited to simple (predominantly text-based) tasks (Guo et al., 2023; Bran et al., 2023) and remains to be validated by experts.

Recent success of Large Language Models (LLMs) on natural language has begun to permeate the molecular domain. Using the formal languages SMILES (Weininger, 1988) and SELFIES (Krenn et al., 2020), transformer-based decoder architectures have been used to directly generate molecular strings (Bagal et al., 2021). Encoder-decoder architectures have been adopted for translating between natural language and formal language (Edwards et al., 2022). The text-molecule translation task has been reformulated for in-context learning tasks for chatbots like ChatGPT (Guo et al., 2023; Li et al., 2024). These efforts have also enabled the creation of instruction datasets (Fang et al., 2023; Liu et al., 2024a) that enable further finetuning for custom tasks. When combined with abundant textual data, using natural language as the surrounding context also enables masked language modeling pretraining (Christofidellis et al., 2023; Liu et al., 2023c; Born & Manica, 2023).

Instead of modeling molecular graphs as strings, specialized graph encoders have been adopted for enhanced cross-modality pretraining (Su et al., 2022; Luo et al., 2023; Liu et al., 2023b; 2024a;b), leading to downstream tasks such as text-based structure retrieval (Liu et al., 2023a) and structure-based design through an interactive conversation (Zeng et al., 2024). However, graph encoders do not come with the same advanced features of foundation models like understanding and reasoning. Aware of these challenges, our work explores an unconventional solution: rendering molecules as images alongside self-generated textual descriptions, implicitly aligning the two modalities in-context. This comes at a ripe opportunity when cheminformatics APIs like RDKit (Landrum) are becoming prevalent enough that MMFMs are likely to have seen sufficient artifacts of the standardized rendering procedure during pretraining.

MMFMs such as GPT-4o possess strong image understanding and reasoning capabilities when interacting with natural language. In this work, we demonstrate that these models can identify molecular substructures *directly* from rendered

[1]MIT CSAIL [2]MIT Chemistry [3]University of Notre Dame [4]MIT-IBM Watson AI Lab, IBM Research. Correspondence to: Michael Sun <msun415@csail.mit.edu>.

*Proceedings of the 42nd International Conference on Machine Learning*, Vancouver, Canada. PMLR 267, 2025. Copyright 2025 by the author(s).

molecular images. These substructures form an interpretable vocabulary that serves as the foundation for building graph grammar. While an interpretable vocabulary is essential, it is insufficient to form a complete and consistent rule set for molecular generation. To address this, we propose a novel and sound algorithmic approach for data-driven grammar rule induction, integrating chain-of-thought (Wei et al., 2022) reasoning with LLM debate (Khan et al., 2024) to execute the heuristics. This method enables the model to iteratively refine its reasoning and decisions, ensuring that the resulting grammar is of high-quality and robust. We call our method Foundation Model Grammar (FMG). FMG leverages the generalist reasoning capabilities (Wei et al., 2021) of foundation models to integrate cross-modal understanding into a robust framework for molecular graph generation.

Our contributions include:

- Introducing the novel FMG framework that uses MMFMs to learn formal graph languages by asking it to make decisions within a hierarchical decomposition algorithm
- Innovative prompting mechanisms to execute the algorithm in a step-by-step manner with MMFM in the loop, e.g. rendering intermediate variables as annotated images
- Proposing LLM-based judge and tournament system to rank decompositions for chemical soundness and validating the approach with expert ground-truth labels
- Demonstrating that FMG outperforms existing state-of-the-art methods on popular molecular generation benchmarks in terms of superior data efficiency, diversity, and synthesizability
- Evaluating FMG's step-by-step reasoning via comprehensive case studies and quantitative analysis.

FMG bridges the gap between elusive expert domain knowledge and emergent capabilities of MMFMs using an interpretable workflow backed by technical rigor, semantic flexibility, and expert validation.

## 2. Related Works

### 2.1. Multi-modal Foundation Models for Molecules

The existing works are classified primarily by: (a) how molecules are represented – whether as formal languages (e.g., SMILES, SELFIES), graphs, or more excitingly, a combination of multiple modalities (Su et al., 2022; Luo et al., 2023; Liu et al., 2023b; 2024a;b); (b) whether individual molecules are paired with natural language (e.g., captions (Edwards et al., 2022), literature context (Liu et al., 2023c)); (c) what features of FMs are used (e.g., encoder (Liu et al., 2023b), decoder (Bagal et al., 2021), encoder-decoder (Edwards et al., 2022), in-context learning (Guo et al., 2023; Li et al., 2024), instruction tuning (Fang et al., 2023; Liu et al., 2024a), few-shot learning (Li

et al., 2024), reasoning); and (d) the task involved (e.g., sequence modeling (Born & Manica, 2023), cross-modality pretraining (Christofidellis et al., 2023; Liu et al., 2023b), (Liu et al., 2023c), generation (Bagal et al., 2021), translation (Christofidellis et al., 2023), retrieval (Liu et al., 2023a), interaction (Zeng et al., 2024), generalist agents (Bran et al., 2023; Chaves et al., 2024)).

### 2.2. Learning Molecular Grammars

Recently, a number of grammar-based generative models have been created (Dai et al., 2018; Nigam et al., 2021; Krenn et al., 2020; Kajino, 2019; Jin et al., 2018; 2020; Guo et al., 2022a; Sun et al., 2024), which have shown enhanced interpretability, synthesizability and data-efficiency. In many cases, the grammar is written manually or created algorithmically, hindering its quality or practicality. For example, Sun et al. (2024) instead prioritizes quality and interpretability by advocating to integrate expert annotations within a graph grammar construction and learning workflow, but its quality is contingent on experts. Guo et al. (2022b) tries to optimize the graph grammar construction process indirectly by parameterizing the sampling function as a neural network, but its decisions are unexplainable. Our approach, by contrast, requires no human contingency and optimizes for the intrinsic quality of the formal language as judged by non-expert LLM agents. At the same time, its decisions are explainable. We use an innovative technique of saving the chain-of-thought reasoning steps for creating "design narratives", which are both explainable artifacts of rule induction and certificates of quality for further scrutiny by humans.

## 3. FMG Learning and Inference

FMG combines the sound framework of the clique tree decomposition algorithm with the adaptability of MMFM decision-making modules for molecular generation. FMG formulates grammar induction as constructing a clique tree and serializes the construction into intuitive selection steps for the MMFM module to follow. In Fig. 1, we show a concrete example of acrylates. The algorithm first initializes most basic units – the base cliques – then proceeds to hand over control to the MMFM's selection modules. The MMFM can merge base cliques to form chemically meaningful substructures (3.3.2 and 3.3.3), remove connections between cliques in the process of spanning tree construction (3.3.4), and finally selec a root motif to anchor the parse tree (3.3.5). The algorithm outputs a junction tree of the input molecule, where each parent-child relationship is converted to a grammar rule (Aguinaga et al., 2018), forming the minimal specialized grammar (MSG) (Wang et al., 2024) for the input molecule. Section 3.5 explains how all MSGs of a given dataset are pooled into the final grammar and Section 3.6 shows how new molecules are generated given

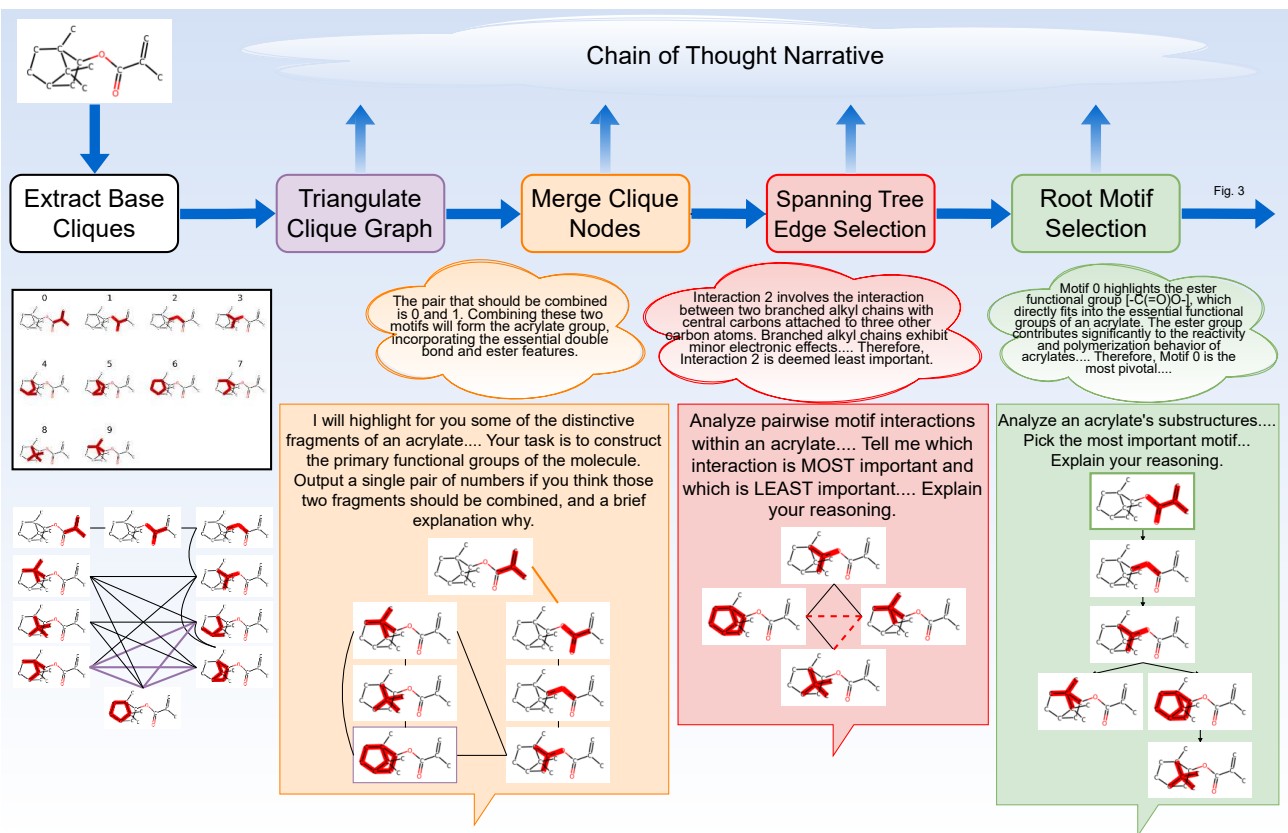

Figure 1. Main modules of FMG algorithm (left) we initialize base cliques using bonds and minimal rings, (left-middle) we triangulate the clique graph to guarantee existence of a clique tree, (middle) we prompt MMFM to meaningfully merge pairs of motifs, (middle-right) we eliminate cycles in the clique graph by prompting MMFM to identify the least important interactions, (right) we prompt MMFM to select the root motif, completing the tree.

the grammar.

## 3.1. Preliminaries

**Molecular Clique Graph**. A base molecular hypergraph is a pair $H = (V_H, E_H)$, where $V_H$ (nodes) is a set of bonds, and $E_H$ (hyperedges) is a set of non-empty subsets of $V_H$. We follow prior work (Kajino, 2019; Guo et al., 2022b) and define $E_H :=$ $\{\{u, v\} \text{ if } u, v \text{ share an atom}\} \bigcup \{\{u\}|\{u\} \text{ is a minimal ring}\}$. Given $H$, we obtain $G_H$, the graph of $H$, where two nodes $u, v$ that share a common hyperedge in $E_H$ are connected. If we can construct a $G_C = (V_C, E_C)$ by extracting the maximal cliques ($V_C$) from $G_H$, and setting $E_C$ to be the clique pairs sharing a common node, we call $G_C$ the molecular clique graph and denote this operation as $CLIQUE(G_H) \rightarrow G_C$. $G_C$ forms the building blocks for further operation. For each $c \in V_C$, we use $V_c$ to denote the clique nodes of $G_H$ within the clique $c$.

**Clique Tree Decomposition**. The clique tree, also known as the junction tree, of $G_H$ is a tree $T$, where each node $\eta$ is labeled with a $V_\eta \subseteq V_H$ and $E_\eta \subseteq E_H$, such that the

following properties hold: 1) For each $v$ in $G_H$, there is at least a node $\eta \in T$ such that $v \in V_\eta$. 2) For each hyperedge $e_i \in E$, there is exactly one node $\eta \in T$ such that $e \in E_\eta$ and $u \in e_i \rightarrow u \in V_\eta$. 3) For each $v \in G_H$, the set $\{\eta \in T \mid v \in V_\eta\}$ is connected. The last property is the running intersection property and is relevant during the clique tree construction phase, as it needs to be checked after each step. The Junction Tree Algorithm achieves this by finding a subset $E'_C \subseteq E_C$, such that $(V_C, E'_C)$ is a spanning tree of $G_C$. There is a theoretical guarantee that if $G_H$ is triangulated, there is always a valid tree decomposition via the existence of an elimination ordering (Koller, 2009). Choosing the best spanning edges $E'_C$ is somewhat an art (Jensen & Jensen, 1994). There is the "optimal" clique tree, the one with minimal treewidth, or $\min_T \max_{\eta \in T} |V_\eta| - 1$, but finding it is NP-hard (Arnborg et al., 1987). Instead, common heuristics, such as the maximum cardinality heuristic (Tarjan & Yannakakis, 1984), are used to find one close to the minimal width.

**Hyperedge Replacement Grammar**. A Hyperedge Replacement Grammar (HRG) is a tuple $(N, T, S, P)$ where: $N$ is a set of non-terminal hyperedge labels in $\mathbb{N}$ (shorthand

$|\cdot|$); $T$ is a set of terminal hyperedge labels; $S \in N$ is the starting non-terminal hyperedge with label $0 := |S|$; $P$ is a set of production rules, each consisting of $A \in N$ (LHS) and a hypergraph with labeled hyperedges and $|A|$ external nodes (RHS).

We adopt an automatic way to convert a clique tree into an HRG by interpreting the clique tree as a parse tree (Aguinaga et al., 2018), where each intermediate node $V_\eta$ becomes the RHS of a production rule and its immediate parent and/or children are used to compute its non-terminal hyperedges and external nodes, as depicted in Fig. 2. The common bonds between $V_\eta$ and its parent become numbered external nodes (black circles). These external nodes are used to "anchor" the replacement, as they also appear in the RHS. For each child of $V_\eta$ (if any), a non-terminal hyperedge (circles labeled N) is added and connected to the common bonds (blue).

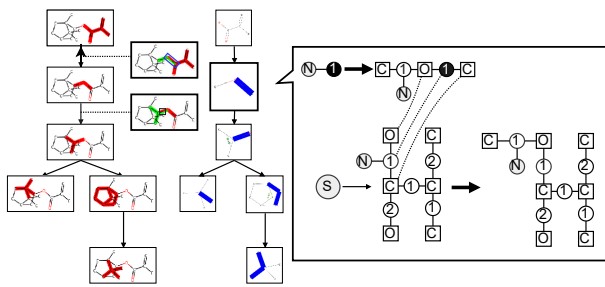

*Figure 2.* Example of conversion from clique tree to HRG production rules; (Left) Each node of the clique tree contains a substructure (red), with edges corresponding to shared bonds between substructures; (Right-top) Rule extracted from second clique of the tree, with a non-terminal hyperedge for the LHS and the clique's substructure being the RHS; (Right-bottom) example of applying the rule, dashed connections are corresponding bonds and atoms

### 3.2. Role of MMFM

For inducing a desirable grammar for molecular generation, a combination of hard and soft constraints must be in place. The hard constraints ensure soundness of the tree decomposition, while soft constraints seek the "optimal" decomposition among all valid decompositions, that best captures the specific characteristics of the data, with the gold standard being a domain expert. The essence of our approach is to automatically handle the hard constraints within a sound algorithmic framework, then leave the exercises of judgment to an MMFM. To modularize the MMFM's involvement, we standardize all tasks to be the task of selecting among a finite set of choices. All MMFM tasks in Section 3.3 can be captured by only two fundamental selections:

**Single Selection**. Given a set $S \subseteq V_C^{(t)}$, the MMFM is

asked to select $s \in S$ or refrain from selection. That is, to select a clique out of all cliques in the current $G_C$.

**Pair Selection**. Given a subset of pairs, $P \subseteq V_C^{(t)} \times V_C^{(t)}$, the MMFM is asked to select $p \in P$ or refrain from selection. That is, to select a pair of bond cliques out of a given set of feasible pairs.

When the context is clear, we denote the raw responses $F_1(S^{(t)})$ and $F_2(P^{(t)})$. We use answer extraction utility prompts to obtain the answers. These selections map to triangulation, merging, cycle removal, and root selection operations on $G_C^{(t)}$. We can execute the full tree decomposition of a molecular clique graph, $G_C^{(0)} \Rightarrow G_C^{(T)}$, using only these operations, driven by MMFM's selections. We will describe each operation $G_C^{(t)} \to G_C^{(t+1)}$, in detail, in the context of constructing the clique tree in Section 3.3.

### 3.3. MMFM Guided Tree Decomposition

See Figure 1 for an illustration of the steps described in this subsection.

#### 3.3.1. CONSTRUCTION OF CLIQUE GRAPH

We initialize $G_H^{(0)}$ to the graph of the base molecular hypergraph. We automatically extract the maximal cliques of $G_H^{(0)}$, thus constructing $G_C^{(0)} \leftarrow CLIQUE(G_H^{(0)})$.

#### 3.3.2. TRIANGULATE CLIQUE GRAPH

We now triangulate $G_H^{(0)}$ to ensure the soundness of the junction tree algorithm. We adopt a chordality testing algorithm (Tarjan & Yannakakis, 1984) that iteratively detects pairs $(u, v) \in V_H \times V_H$ that would form chordless cycles of length $> 3$ if left unaddressed. At each iteration $t$ in which the algorithm returns a pair $(u, v)$ connected by a chord, we set $P^{(t)} \to \{(c_1, c_2) \mid c_1 \in V_u \cap c_2 \in V_v\}$. Let $c^*_1, c^*_2 \leftarrow F_2(P^{(t)})$. We then merge $c^*_1, c^*_2$ by adding all edges, $E_H^{(t+1)} \leftarrow E_H^{(t)} \cup V_{c^*_1} \times V_{c^*_2}$. We update $G_C^{(t+1)} \leftarrow CLIQUE(G_H^{(t+1)})$. Let $G_C^{(T_1)}$ denote the clique graph once $G_H$ is triangulated. We proceed to the next phase.

#### 3.3.3. MERGE CLIQUE NODES

We now would like to give the MMFM the option to further merge cliques that form more cohesive motifs, e.g. functional groups, in the context of the base molecule. Starting with $t = T_1$, we set $P^{(t)} \leftarrow E_C^{(t)}$. If $F_2(P^{(t)})$ does not return, we terminate and proceed to the next phase. Otherwise, at each iteration, we let $c^*_1, c^*_2 \leftarrow F_2(P^{(t)})$. We merge $c^*_1, c^*_2$ following the same operation steps as in Step 2. Let $G_C^{(T_2)}$ denote the clique graph upon termination of this phase.

### 3.3.4. SPANNING TREE EDGE ELIMINATION

We now extract a spanning tree over $E_C^{(T_2)}$ using a top-down approach of detecting and eliminating cycles of $G_C^{(T_2)}$. We terminate and proceed to the next phase once there are no more cycles. Otherwise at each step t, let $c_1, c_2, \ldots, c_k, c_1$ be one such cycle. We set $P^{(t)} \leftarrow \{(c_i, c_{(i+1)\%k}) \mid$ removing $c_i, c_{i+1}$ will not violate running intersection $, i = 1, 2, \ldots, k\}$. We then update $E_C^{(t+1)} \leftarrow E_C^{(t)} \setminus \{F_2(P^{(t)})\}$. Let $G_C^{(T_3)}$ denote the clique tree once all cycles have been removed.

### 3.3.5. ROOT MOTIF SELECTION

Lastly, we root $G_C^{(T_3)}$ at $F_1(V_C^{(T_3)})$. The final clique tree is $G_C^{(T)}$ ($T = T_3 + 1$). We obtain the multi-set of production rules using this decomposition, $\mathcal{P}(G_C^{(T)})$.

### 3.4. Prompting Setup

For each selection, we prompt the MMFM with rendered images from the rdkit and dynamical textual descriptions related to the current state of the decomposition ($G_C$), in addition to the static prompt, which includes some background on the domain and detailed task instructions.

**Rendering Images.** For single selection (root motif selection), we use the Python package rdkit to render the molecule and highlight the bonds ($V_c$) of a single substructure ($c \in S^{(t)} \subseteq V_C^{(t)}$) in a cell. We use matplotlib.pyplot to enact a grid cell layout so all choices are shown together. For double selection where the number of choices is small (edge selection), we highlight each pair ($c_1, c_2 \in P^{(t)} \subseteq V_C^{(t)} \times V_C^{(t)}$) using different colors in the same cell. For double selection where the number of choices are large (merging cliques), we render each clique in a separate cell, just like with single selection, but the task instruction is to select a pair of cliques.

**Dynamic Textual Descriptions.** Motivated by the success of prompt-based learning techniques, we assist GPT's reasoning during selection tasks by plugging in isolated descriptions of each element of $S$ or $P$ into the task prompt, enabling multi-modal alignment. These are obtained by rendering each substructure (or pair of substructures) in isolation and asking GPT to describe those. An example of an isolated description is "Motif 5. Benzene - A six-membered aromatic ring entirely consisting of carbon atoms", whereas an in-context description is "Motif 5. A six-membered ring, similar to benzene, but includes distinct locations for double bonds from Motif 1."

**Rephrasing Prompts.** We then use format conversion prompts to convert GPT's sometimes elaborative answers into simple phrases that can be grammatically inserted into

subsequent task prompts (example: "Motif 9. This motif is another carbocyclic structure, specifically a bicyclic system with carbon double bonds..." $\rightarrow$ "a bicyclic carbocyclic structure with carbon double bonds").

**Task Prompts.** These are the primary prompts that instruct GPT to perform the selection tasks. We substitute rephrased dynamic descriptions of individual cliques (motifs) where appropriate into these templates and specifically instruct GPT to explain its reasoning. Example walkthroughs featuring all the task prompts are given in App. H.

**Answer Extraction Prompts.** We use low-level utility prompts to postprocess an answer prompt into a fixed format for regex extraction (example: "After extensive deliberation, the interaction between Motif 5 and Motif 7 seems weakest of the ones shown" $\rightarrow$ "5,7")

**Thought Collection Prompts.** We collect GPT's responses into summarized reasons for a particular selection, as they will be composed into a narrative (more in Section 3.5). For a particular selection at time $t$, let $COT(F_{j(t)})$ be the prompt chaining composition to return a summarized reasoning over the selection. We denote the output as $COT^{(t)}$.

### 3.5. FMG Inference

**LLM-based Feedback Mechanism.** Our MMFM-guided algorithm is inherently stochastic, as repeated runs may produce different decompositions. In the absence of human experts, it's difficult to judge how "good" the rules produced by each decomposition are. The key challenge is that, given only the grammar rules, absent of any context, it is difficult to evaluate its semantic qualities. We channel the built-in explainability of our approach into a clever solution. We reuse the natural language artifacts (e.g. chain of thought, explanations) logged during execution of our algorithm as a certificate of the quality of that single decomposition. With this in mind, we adopt a simple yet effective learning procedure to optimize the FMG. We first perform $K$ passes (i.e., independent runs of the algorithm) over, the molecule $H$, producing decompositions $[G_{C_k}, k = 0, \ldots, K - 1]$. Denoting $[COT_k^{(t)}, t = 0, \ldots, T - 1]$ as the chain of thoughts for the $k$'th pass over molecule $H$, we combine it with knowledge of the timestep delimiters $T_1, T_2, T_3$ to compose a step-by-step story of how the molecule was decomposed. Instead of evaluating rules in isolation, we group rules into specialized grammars, defined as the set of rules used in the tree decomposition of a single example. The resulting story provides much needed context for an LLM to assist in the evaluation.

**LLM-based Tournaments.** We use a tournament-based evaluation system by pitting stories of discrepant decompositions for comparison by a non-expert LLM. We pit discrepant runs (A and B) for the same molecule against each

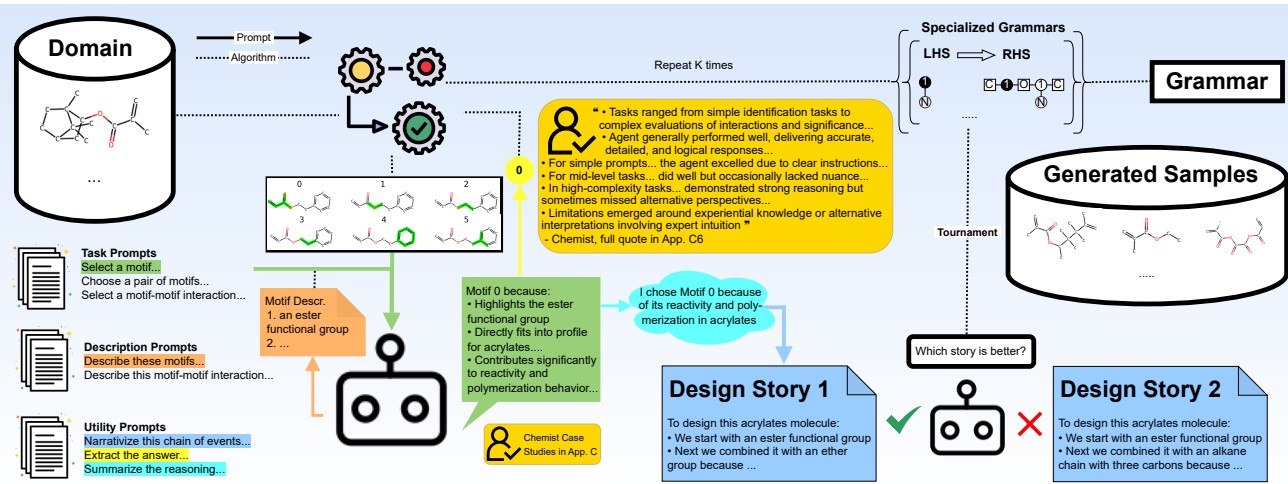

*Figure 3.* Our workflow takes as input a class-specific dataset and a collection of prompts (left); executes the tree decomposition algorithm with MMFM as a decision-making module (left middle); converts the parse tree into production rule set (left-right), resolving discrepancy across runs with a non-expert LLM judge; and infers a grammar which can generate new class-specific samples (right).

other and ask the vanilla LLM to decide which story wins (A or B) on the basis of validity, soundness, and perceived depth of understanding. We adopt a Swiss tournament format, with 4 rounds, and use the logits of the first token in the response to assign weighted outcomes of the matchup, similar to how Khan et al. (2024) designed the preference model. We consolidate all outcomes using the Bradley-Terry model (Bradley & Terry, 1952), a statistical model used for paired comparisons, where each debater's ability is inferred from the pairwise results. We rank and order the participants $[0, 1, \ldots, K-1] \overset{\text{permute}}{\to} [r_1, r_2, \ldots, r_K]$ according to the tournament results and define the "Top $k$" FMG as the HRG inferred by the production rule multi-set $\bigcup_{r \in \{r_1, \ldots, r_k\}} P(G_{C_r})$, where $\bigcup$ is the multiset union.

### 3.6. Stochastic Sampling for Molecular Generation

So far, we have only considered the contribution to the HRG by decomposing a single molecule, $H$. In the domain-specific setting, we are given a small dataset of class-specific molecules, which we convert into our base molecular hypergraphs: $\mathcal{D} := \{H^{(i)} \mid 1 \leq i \leq N\}$, to create a set of specialized grammars $\mathcal{P}_\mathcal{D} := \{P(G_C^{(i)})\}$. Instead of naively pooling all grammars together, we maintain a count for the number of times each rule is applied, aggregated across $\mathcal{P}_\mathcal{D}$. During generation, the algorithm finds all applicable rules and chooses one with probability proportional to its count to apply, according to the definition in Drewes et al. (1997). We adopt Kajino (2019)'s technique to ensure valid conversion from hypergraph to molecule.

## 4. Results

**Baselines.** We evaluate our method against a variety of similar and alternative approaches.

- Grammar Generative Models (MHG (Kajino, 2019), DEG (Guo et al., 2022b), Random Walk (RW) (Sun et al., 2024), STONED (Nigam et al., 2021)): How does MMFM compare with algorithmic and expert-based annotation?
- Grammar/Scaffold based Molecular VAEs (JT-VAE (Jin et al., 2018), Hier-VAE (Jin et al., 2020), MoLeR (Maziarz et al., 2021)): How does grammar-based generation fare against neural parametrizations that still consider motifs?
- Chemical Language Models (CLMs) (MolGPT (Bagal et al., 2021), MolT5 (Edwards et al., 2022), Text+Chem T5 (Christofidellis et al., 2023)): How do LLMs trained on large databases of molecular strings transfer or adapt to domain-specific settings?
- In Context Learning (ICL): How does directly prompting an LLM, in the absence of any formal framework, do?

More details on each baseline is in App. B.

**Metrics.** We use common unconditional generation metrics adopted by molecular generative models (Polykovskiy et al., 2020):

- **Valid/Unique/Novelty** (percent of valid/uniq/novel mols)
- **Diversity** (avg. pairwise Tanimoto distance (Rogers & Hahn, 2010))
- **Retro\* Score (RS)** (success rate of Retro* model (Chen et al., 2020)
- **Membership** (percentage of molecules belonging to the dataset's chemical class)[1].

---

[1]We generate 10000 for small datasets and 1000 for

**Datasets.** We evaluate on three small monomer datasets used by Guo et al. (2022b) curated from literature, as well as two real-world datasets from the photovoltaic and toxicology domains used by Sun et al. (2024).

*Table 1.* Results on small datasets Isocyanates (11 examples), Acrylates (32) and Chain Extenders (11). Best rounded result(s) for each metric **bolded**. (T): Training from scratch; (FT): Fine-tuning, (I): Inference, using posterior interpolation. Since T5 (I) methods struggle to generate sufficient valid and unique samples, we exclude them from ranking.

| Method | Valid (Avg.) | Unique | | | Div. | | | RS | | | Memb. | | |
|---|---|---|---|---|---|---|---|---|---|---|---|---|---|
| Train Data | 100% | 100% | 100% | 100% | 0.61 | 0.67 | 0.80 | 100% | 100% | 100% | 100% | 100% | 100% |
| JT-VAE (T) | 100% | 5.8% | 0.5% | 2.3% | 0.72 | 0.29 | 0.62 | 5.5% | 4.9% | 2.2% | 66.5% | 48.64% | 79.6% |
| Hier-VAE (T) | 100% | 99.6% | 99.7% | 99.8% | 0.83 | 0.83 | 0.83 | 1.85% | 3.04% | 2.69% | 0.05% | 0.82% | 43.6% |
| MHG (T) | 100% | 75.9% | 86.8% | 87.4% | 0.88 | 0.89 | 0.90 | 2.97% | 36.8% | 50.6% | 12.1% | 0.93% | 41.2% |
| MoLeR (FT) | 100% | 87.1% | 40.7% | **100%** | 0.86 | 0.80 | 0.91 | 69.2% | **97.7%** | 70.7% | 77.3% | 72.2% | 97.3% |
| MoLeR (I) | 100% | 65.7% | 45.4% | 51.1% | **0.90** | **0.90** | 0.90 | 61.3% | 76.2% | 92.3% | 0.08% | 32.0% | 95.5% |
| STONED | 100% | **100%** | 99.8% | 99.8% | 0.85 | 0.84 | **0.93** | 5.63% | 11.2% | 6.78% | 79.8% | 47.9% | 61.0% |
| DEG | 100% | **100%** | **100%** | **100%** | 0.86 | 0.86 | **0.93** | 27.2% | 43.9% | 67.5% | 96.3% | 69.6% | 93.5% |
| GPT4 (ICL) | 91% | 73.0% | 35.1% | 63.5% | 0.86 | 0.78 | 0.87 | **84.4%** | 95.0% | 98.0% | 93.7% | 99.7% | 99.5% |
| MolT5 (I) | 76% | 0.9% | 0.3% | 7.1% | 0.09 | 0.21 | 0.75 | 98.1% | 99.6% | 48.5% | 99.8% | 100% | 100% |
| Text+Chem T5 (I) | 42% | 26.2% | 46.4% | 49.8% | 0.55 | 0.71 | 0.80 | 87.6% | 58.3% | 43.9% | 100% | 100% | 100% |
| FMG | 100% | **100%** | **100%** | **100%** | 0.73 | 0.46 | 0.85 | 61.7% | 93.0% | **99.1%** | **99.6%** | **100%** | **99.8%** |
| FMG-Text | 100% | **100%** | **100%** | **100%** | 0.73 | 0.46 | 0.84 | 33.1% | 87.1% | 98.5% | **99.6%** | **100%** | 99.6% |

**Results on Small Datasets.** We first observe in Tables 1 and 2 that VAE methods struggle to generate unique molecules, suggesting they collapse in this extreme setting, consistent with findings by Guo et al. (2022b); Sun et al. (2024). Hier-VAE fares better, as it incorporates inductive bias of larger substructures, but this comes at the expense of RS and Memb., suggesting an undesirable shift in distribution.

Meanwhile, the CLMs struggle with task comprehension, sometimes mixing natural language with SMILES, or violating SMILES syntax, e.g. forgetting to close parentheses. The few valid examples are often repetitive or similar (as low as <1% uniqueness and <0.6 Div.), despite efforts to find balanced decoding strategies like varying beam width or adjusting temperature. To their credit, the few valid and unique examples achieve high RS and Memb., suggesting that the LMs may have memorized a few relevant examples in the domain but failed to generalize.

The grammar-based methods do better on validity and coverage, but struggle across RS and Memb.. Despite *optimizing* for RS and Div., DEG still falls short of FMG. This indicates that these methods can learn the syntax of the underlying language, but not its semantics like synthesizability or membership criteria, signs of expert-level understanding. FMG's high RS scores are hence impressive considering that we only prompted GPT to "highlight the primary functional groups of the molecule". FMG also achieves nearly 100% class membership in Table 1. In fact, ICL also achieves high RS and Memb., suggesting MMFMs are sufficiently knowledgeable about these niche chemical classes that it can to some extent learn the molecular generation task in-context. However, ICL has similar struggles as CLMs in generating valid and unique examples. By combining the semantic com-

prehension of ICL with our formal framework, we implicitly capture the knowledge into the selections while ensuring validity and robust performance.

However, FMG still leaves some to be desired across coverage. Our investigation reveals that the learning procedure is inclined towards forming cliques representing more complex substructures that are characteristic of the chemical class or known to be synthetically accessible. The applicability of a rule decreases as the RHS becomes more complex, and so the grammar's coverage decreases. We suspect that the low diversity is due to this phenomenon occurring in the extreme setting of having ≈ 30 or fewer samples, as that creates fewer rules which are less applicable.

*Table 2.* Results on Real-World Datasets HOPV (316 examples) and PTC (348). Same protocol as Table 1. Since VAE (T) methods struggle to generate sufficient unique samples, we exclude them from ranking.

| Method | Valid (Avg.) | Unique | | Novelty | | Div. | | RS | | Memb. | |
|---|---|---|---|---|---|---|---|---|---|---|---|---|
| Train Data | 100% | 100% | 100% | N/A | N/A | 0.86 | 0.94 | 51% | 87% | 100% | 30% |
| JT-VAE (T) | 100% | 11% | 8% | 100% | 80% | 0.77 | 0.83 | 99% | 96% | 84% | 27% |
| Hier-VAE (T) | 100% | 43% | 20% | 96% | 85% | 0.87 | 0.91 | 79% | 92% | 76% | 25% |
| Hier-VAE +expert (T) | 100% | 29% | 28% | 92% | 75% | 0.86 | 0.93 | 84% | 90% | 82% | 17% |
| MoLeR (FT) | 100% | **100%** | 99% | **100%** | 99% | 0.90 | 0.92 | 42% | 71% | 60% | 30% |
| MoLeR (I) | 100% | **100%** | 94% | **100%** | 97% | 0.90 | 0.92 | 25% | 79% | 74% | 45% |
| DEG | 100% | 98% | 88% | 99% | 87% | **0.93** | **0.95** | 19% | 38% | 46% | 27% |
| RW (expert) | 100% | **100%** | **100%** | **100%** | **100%** | 0.89 | 0.93 | 58% | 60% | 71% | 22% |
| GPT4 (ICL) | 71% | 95% | 84% | 99% | 98% | 0.91 | 0.93 | 46% | 56% | 53% | **89%** |
| MolGPT (T) | 26% | 86% | 41% | 71% | 47% | 0.84 | 0.88 | 43% | **91%** | 86% | 70% |
| MolT5 (I) | 61% | 12% | 20% | **100%** | 95% | 0.41 | 0.77 | 8% | **91%** | 0% | 45% |
| Text+Chem T5 (I) | 48% | 81% | 95% | 95% | 67% | 0.87 | 0.91 | 42% | 50% | **88%** | 47% |
| FMG | 100% | **100%** | **100%** | **100%** | 92% | **0.93** | 0.93 | **70%** | 78% | 38% | 46% |
| FMG-Text | 100% | **100%** | **100%** | **100%** | 91% | 0.92 | 0.93 | 69% | 77% | 43% | 33% |

**Results on Real-World Datasets.** We see similar relative trends across the baselines in Table 2. VAE (T) methods still suffer from poor data-efficiency, but MoLeR (FT) becomes quite competitive. However, it still struggles on RS and Memb. The low uniqueness and novelty of the VAE baselines invalidates its seemingly high RS score, achieved by sampling smaller molecules. Meanwhile, FMG keeps high RS scores while lifting coverage to be far more reasonable, as the size of the dataset becomes larger. FMG is one of only two methods that achieve 100% uniqueness (the other being RW with access to expert annotations) while tied for first and second on diversity for HOPV and PTC, respectively.

CLM baselines exhibit similar struggles and perform far worse on HOPV, likely due to the photovoltaic domain shifting further out-of-distribution compared to the drug-pervasive pretraining distribution, e.g. ChEMBL. Amongst grammar-based methods, FMG surpasses even RW on RS (by 12% and 18%), suggesting that FMG is more amenable to synthesis considerations even for larger, more hand-engineered molecules. Though membership is not strictly defined for these two domains, FMG appears to do exceptionally well for PTC (halides) but poor for HOPV (thiophenes), which is surprising. As we see later in App. E, $k$ imposes a sharp tradeoff between Memb. and {Div., RS}, though FMG is capable of achieving exceptional numbers for either/or. Our results suggest that domain-general FMs

HOPV/PTC, use the same Retro* parameters and adopt the same membership criteria as Guo et al. (2022b); Sun et al. (2024).

Table 3. Snapshot of PTC expert case study (App. H.1). Full case studies for all datasets in App. H.

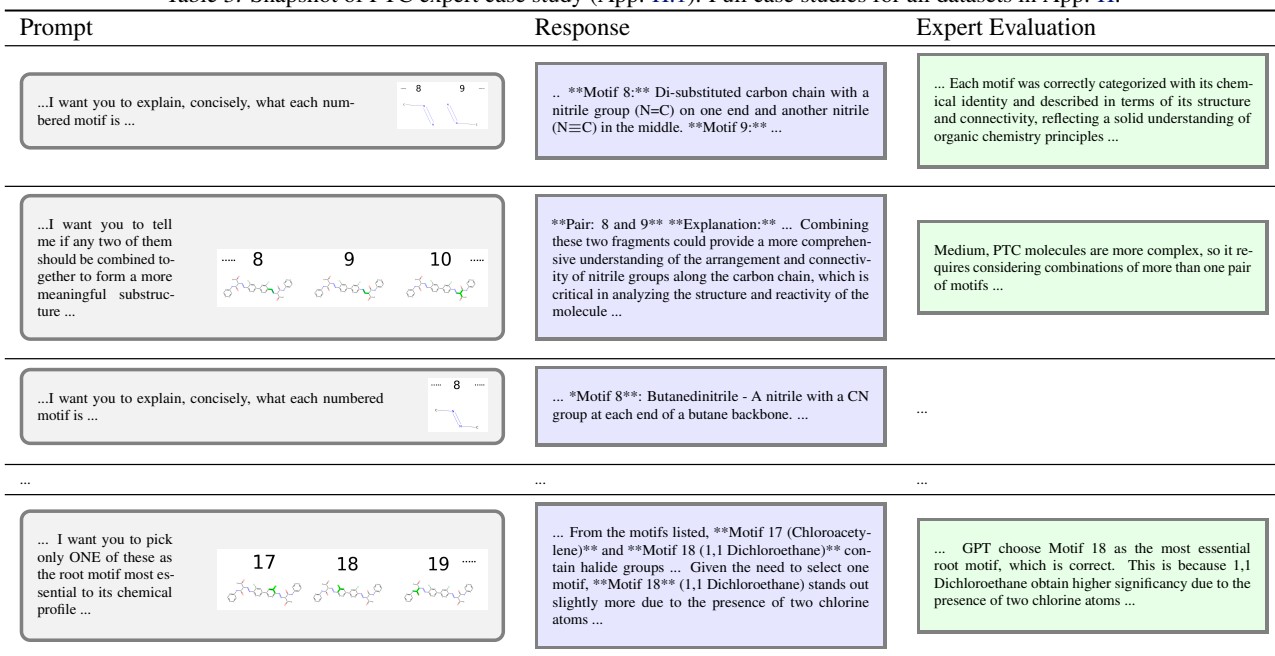

| Prompt | Response | Expert Evaluation |
|---|---|---|
| ...I want you to explain, concisely, what each numbered motif is ... | .. **Motif 8:** Di-substituted carbon chain with a nitrile group (N=C) on one end and another nitrile (N≡C) in the middle. **Motif 9:** ... | ... Each motif was correctly categorized with its chemical identity and described in terms of its structure and connectivity, reflecting a solid understanding of organic chemistry principles ... |
| ...I want you to tell me if any two of them should be combined together to form a more meaningful substructure ... | **Pair: 8 and 9** **Explanation:** ... Combining these two fragments could provide a more comprehensive understanding of the arrangement and connectivity of nitrile groups along the carbon chain, which is critical in analyzing the structure and reactivity of the molecule ... | Medium, PTC molecules are more complex, so it requires considering combinations of more than one pair of motifs ... |
| ...I want you to explain, concisely, what each numbered motif is ... | ... *Motif 8**: Butanedinitrile - A nitrile with a CN group at each end of a butane backbone. ... | ... |
| ... | ... | ... |
| ... I want you to pick only ONE of these as the root motif most essential to its chemical profile ... | ... From the motifs listed, **Motif 17 (Chloroacetylene)** and **Motif 18 (1,1 Dichloroethane)** contain halide groups ... Given the need to select one motif, **Motif 18** (1,1 Dichloroethane) stands out slightly more due to the presence of two chlorine atoms ... | ... GPT choose Motif 18 as the most essential root motif, which is correct. This is because 1,1 Dichloroethane obtain higher significancy due to the presence of two chlorine atoms ... |

are already aligned with chemistry-specific desiderata like synthesizability and specificity, promoting the intrinsic quality of the grammar. We include visualizations and further interpretation of the results in App. C.

**Expert Evaluation of Interpretability.** One of FMG's main advantages is its interpretability. The algorithm execution yields step-by-step reasoning steps for decomposing each molecule. Table 3 shows a snapshot from our five in-depth step-by-step walkthroughs in App. H.1-H.5. Each reasoning step is scrutinized by an expert and the input prompt classified for its difficulty. We tallied all the steps across the five case studies by difficulty. The expert completely agrees with FMG in a majority of cases and partially agree in all cases. We report the full findings in App. H.6.

**Expert Agreement with LLM Judge.** FMG's step-by-step reasoning steps are not only milestones of interpretability; they directly influence the final grammar by closing the feedback loop. Our LLM-as-a-judge protocol pits discrepant decompositions and their "design stories" against each other (Sec. 3.5) in a tournament setting. To validate this protocol, we prepare some ground-truth pairwise outcomes. We ask both the expert and LLM judge to decide which decomposition is more chemically sound, solely based on their design stories. The results are in App. D, with the takeaway being: against a random coin toss baseline, the MMFM's agreement with the expert yields $p = 1.1e - 5$, indicating statistically significant alignment with expert judgement. In the next study, we quantify the downstream effect of refining the grammar by choosing higher-ranked decompositions.

**Behavior on Small Subset of a Large Benchmark.** For a bonus challenge, we evaluate FMG's behavior when run on a tiny (0.05%) subset of the MOSES benchmark (Polykovskiy et al., 2020). We generated 30k samples and computed standard metrics. To our surprise, we find FMG *leads* on all unconditional generation metrics, over leaderboard methods trained on 100% of data, but lags behind on distribution metrics. The results and discussion are in App. G.

## 5. Discussion

### 5.1. Ablation Study on LLM-based Tournament

**Setup.** We quantify the gains from LLM-based tournaments in a more controlled setting. We set $K = 10$ and study the performance of Top-$k$ FMG as $k$ increases from 1 to $K$. As a baseline, we compare with the "1-$k$" FMG, which is the HRG inferred by $\bigcup_{r=0}^{k-1} P(G_{C_r})$. Intuitively, Top-$k$ FMG selects the top $k$ ranked runs per molecule, while 1-$k$ FMG selects the first $k$. In both cases, the final grammar pools together $N \cdot k$ specialized grammars from individual runs.

**Results.** We visualize Top-$k$ FMG and 1-$k$ FMG on small datasets in Fig. 4. As the number of rule-set candidates $k$ increases, we find a sharp trade-off in class membership versus coverage: at low $k$ (esp. $k = 1$), membership reaches nearly 100%, but it steadily declines as sub-optimal rules are added. Meanwhile, increasing $k$ improves both diversity and RS – pooling simple rules enables generation of more varied, easily synthesized molecules – but at the cost of motif inclusion. Notably, there's no significant differ-

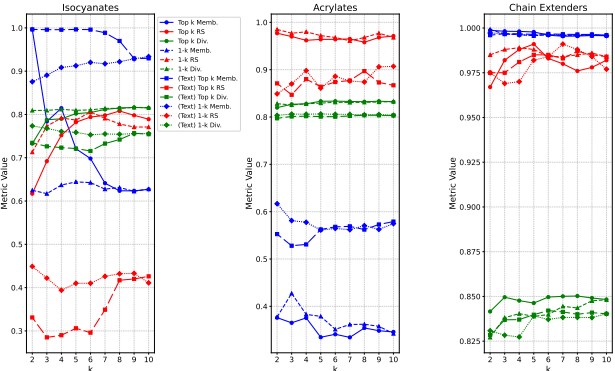

*Figure 4.* We vary $k$ from 1-10 for FMG and 2-10 for FMG-Text (see Sec. 5.2). Full FMG results in App. E. Full comparison between FMG and FMG-Text in App. F.3.

ence between Top-$k$ versus 1-$k$ FMG for diversity and RS, suggesting tournament rankings reflect understanding of class-specific motifs more and is neutral to broader generative criteria. Results on real-world datasets and further analysis are in App. E.

### 5.2. Ablation Study on Image vs Text Input Format

We also add results in Tables 1 & 2 from a controlled study where we modify FMG to use a text-based encoding (FMG-Text) instead of molecular images to test our hypothesis that images are superior to text as the input format.

**Setup.** Each FMG input is a grid of cells showing (1) a molecule, (2) a molecule with one substructure, or (3) a molecule with two substructures highlighted in different colors. While (1) can be replaced by SMILES, (2) and (3) require visual emphasis of substructures, which text-based formats struggle to express. We use a verbose but reliable approach: number all atoms and list motif atoms. This improved parsing and allowed us to fully re-implement FMG with text inputs (FMG-Text) by swapping each visual cell for its text-based counterpart. We replicated the Sec. 5.3 study for FMG-Text too, and the aggregate results on small datasets are shown in Fig. 4 (Text).

**Results.** FMG-Text is a competitive alternative when images cannot be used. Barring FMG, FMG-Text leads in 13/24 columns of Tables 1 & 2. FMG with image inputs still performs best overall – especially on diversity and synthesizability, which are crucial for practical molecule generation. We identify a tradeoff for Isocyanates & Acrylates that is not present for Chain Extenders. Text inputs can make small motif identification (membership) easier and more robust to larger $k$. Image inputs better support global substructure reasoning, which drives higher synthesizability. Chain Extenders are more linear, so the choice of text or image representation may be perceived equally by FMG. Results

on real-world datasets and additional background on why FMG defaults to images over text are in App. F.

### 5.3. Ablation Study on Heuristic vs MMFM Modules

*Table 4.* For each module, we swap MMFM for a heuristic (-module). We report the "1-$k$" FMG (FMG Union) and avg. across $k$ runs (FMG Avg) on the 3 small datasets. We choose $k = 5$.

| Method | Novelty | | | Div. | | | RS | | | Memb. | | |
|---|---|---|---|---|---|---|---|---|---|---|---|---|
| FMG Avg | 99.96±0.01 | 99.86±0 | 99.94±0 | 0.79±0.01 | 0.83±0 | 0.81±0.02 | 44.3±3.4 | 87.4±1.5 | 91.9±3.8 | 60.14±13.63 | 35.48±4.02 | 28.30±13.25 |
| FMG Union | 99.96 | 99.87 | 99.94 | 0.81 | 0.83 | 0.84 | 78.7 | 97.2 | 98.8 | 64.42 | 37.88 | 22.07 |
| FMG (-merge) Avg | 99.95±0.00 | 99.88±0 | 99.94±0 | 0.74±0.01 | 0.83±0 | 0.85±0.00 | 32.6±5.7 | 91.0±2.0 | 97.4±0.8 | 95.75±4.16 | 16.61±0.78 | 15.48±1.11 |
| FMG (-merge) Union | 99.95 | 99.88 | 99.94 | 0.76 | 0.83 | 0.85 | 39.7 | 90.3 | 96.4 | 93.74 | 16.40 | 14.44 |
| FMG (-edge) Avg | 99.96±0.00 | 99.87±0 | 99.95±0 | 0.76±0.01 | 0.82±0 | 0.77±0.00 | 57.9±6.5 | 93.5±1.8 | 99.9±5.4 | 45.81±15.34 | 37.44±5.32 | 38.56±10.58 |
| FMG (-edge) Union | 99.95 | 99.87 | 99.95 | 0.81 | 0.83 | 0.84 | 66.8 | 92.7 | 98.4 | 58.57 | 33.83 | 16.23 |
| FMG (-root) Avg | 99.96±0.01 | 99.88±0 | 99.94±0 | 0.79±0.03 | 0.85±0 | 0.83±0.02 | 49.1±7.0 | 89.5±2.6 | 91.9±10.9 | 52.17±12.13 | 22.90±2.53 | 14.23±6.39 |
| FMG (-root) Union | 99.97 | 99.86 | 99.94 | 0.82 | 0.85 | 0.86 | 54.9 | 87.0 | 96.2 | 47.01 | 22.18 | 14.84 |

**Setup.** We separately ablate each MMFM-assisted module to investigate how crucial each module is for FMG. We ablate the merge module by directly passing $G_C^{(T_1)}$ to Step 3.3.4, the spanning tree module by the maximal spanning tree (MST) heuristic (Tarjan & Yannakakis, 1984), and the root module by picking a root clique at random. Ablating any LLM module breaks the overall design story, so we can no longer rank top $k$ in earnest, hence we omit Top-$k$ FMG.

**Results.** In Table 4, we see ablating any MMFM component has negative implications for the results, albeit in different ways and differently across datasets. Ablating Merge discourages the class-defining motifs for acrylates and chain extenders to be formed during decomposition, meaning they are less likely captured in the rules (with an exception for isocyanates, whose defining motif (N=C=O) has only 2 bonds and must already be in a clique). For isocyanates, RS drops significantly: it's known an amine (R-NH2) reacts with phosgene (COCl2) to produce the isocyanate, so without MMFM's knowledge, the intermediate may not be formed, producing rules less amenable to synthetic considerations. Ablating MMFM-guided spanning tree construction has milder consequences. Diversity, RS, and membership are only slightly worse. The MST heuristic is well-motivated, but its rule-based selection is less adaptable to domain-specific constraints like chemical reactivity, since it models interaction strength solely on the basis of neighborhood overlap. Meanwhile, an MMFM is more flexible to capture these constraints, selectively breaking the heuristics when context necessitates it.

## 6. Conclusion

We present a MMFM-guided grammar induction framework for molecular generation. FMG with GPT-4o as the base model already demonstrates expert-approved reasoning abilities by applying its chemistry knowledge, image comprehension, and in-context abilities within our guided tree decomposition algorithm. We use innovative prompting and feedback mechanisms to ensure the workflow is interpretable, user-friendly, customizable, and (dare we say) *foundational* for future molecular design workflows.

## Acknowledgements

Michael and Gang completed internships at the MIT-IBM Watson AI Lab. Gang is supported by the IBM Fellowship.

## Impact Statement

This paper presents work whose goal is to advance molecular discovery workflows. The application of our method can have consequences for real-world discovery workflows. There are no ethical aspects which we foresee and feel must be discussed here.

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

# A. All About Prompts

The prompts consumed by the MMFM is instantiated from a generic template. Each generic template has both static substitutions, e.g. hand-written descriptions of a specific domain, and dynamic substitutions, e.g. motifs from the grammar induction. We denote static substitutions within [] and dynamic substitutions within <>. As a simple example, the template "Design a [] molecule containing a <>." [] could be a class (e.g. photovoltaic) and <> could be a specific motif, populated at runtime. In Table 5, we provide the generic templates, example of a static substitution (we use PTC as the running example for static substitutions), and example of a dynamic substitution for how we include chain-of-thought. In Table 6, we provide templates used for FMG inference, using a similar format as the first table.

# B. Baseline Settings

For VAEs, we define (T), (FT) and (I) settings as follows:

- **Training (T)**: We train a VAE from scratch.
- **FT (FT)**: We leverage any pretrained checkpoints released by the authors, where possible. We do a 80-20 train-val split of the dataset and finetune until the validation loss converges. We select the checkpoint with the lowest validation loss for sampling.
- **Inference (I)**: For our Small Datasets, there are as few as 11 samples, making (FT) extremely difficult. We instead adapt pretrained checkpoints to sample in the posterior distribution of the dataset. Specifically, we fit a Gaussian distribution to the encoded latent vectors from our dataset. We then sample latent vectors from the fitted Gaussian distribution to decode into molecules.

1. **JT-VAE**: We train with default hyperparameters until convergence, following the instructions on public repository. We sample from a Gaussian prior.
2. **HierVAE**: We follow the preprocessing steps to obtain a vocabulary from the full dataset. We then train with default hyperparameters. As recommended, we train an unsupervised language model on the full dataset until convergence, then sample from a Gaussian prior. Note that a train-test split is not possible due to the vocabulary differing between splits. For the (+expert) setting, we use the expert motifs curated by Sun et al. (2024).
3. **MoLeR (I)**: We load the publicly released pretrained checkpoint and do posterior interpolation to sample from our data distribution. Specifically, we fit a Gaussian distribution by taking the mean to be the average of the latent codes from encoding the samples in our dataset and estimating the full covariance matrix. We use sklearn.mixture.GaussianMixture's implementation.
4. **MoLeR (FT)**: We finetune the pretrained checkpoint on our real-world datasets. We do the recommended 80-20 train-valid split, where metadata between splits are synced. We finetune until validation loss converges. We use all the default hyperparameters.
5. **MolGPT**: We use a batch size of 32 to accomodate our smaller datasets. Since the vocabulary size is different with the pretrained models, we are unable to finetune. We use the same train, test split as MoLeR and select the checkpoint with the best validation loss.
6. **GPT4-ICL**: We use OpenAI API for the GPT-4o model. An example prompt for acrylates is provided:

> Here are some acrylates.
> <examples>
> Remember the defining acrylate group is C=CC(=O)O, which consists of a carbon-carbon double bond and a carboxylate ester. Generate 100 more acrylates. Output one SMILES per line, beginning with the first line.

<examples> is replaced with all the SMILES strings in the small dataset, or a random subset of all the strings for the real-world datasets.

7. **Chemical Language Models (MolT5, Text+Chem T5)**: We implement the pretrained checkpoints "molt5-large-caption2smiles" and "multitask-text-and-chemistry-t5-base-augm" for MolT5 and Text+Chem T5, respectively. We prompt the models for molecular generation. We set the maximum generation length to 512. For each generation, we randomly sample five molecules from the training set to construct the prompt. The template is similar to the one employed in GPT4-ICL.

*Table 5.* Reference table of all tree decomposition task prompts used. We number substitutions and show the substituted text from a sample run. See text for how they are used (Section 3.5).

| Prompt | Example [substitutions] | Example <substitutions> |
|---|---|---|
| I want you to think like a chemist performing a detailed analysis of the chemical composition of [1] through its constituent motifs. I will highlight for you some of the distinctive fragments of a molecule. They are numbered from 0 and individually highlighted in RED. Focus ONLY on the substructure highlighted in red within each cell. Here is the descriptions for each substructure provided by an expert: <1> I want you to tell me if any two of them should be combined together to form a more meaningful substructure. [2] Your task is to highlight the primary functional groups of the molecule. Output a single pair of numbers if you think those two fragments should be combined, and a brief explanation why. If no such pairs exist, don't output anything. | a toxic compound

This molecule belongs to a collection of molecules characterized by distinct functional groups known for their carcinogenic properties or liver toxicity. These groups comprise a rich variety of elements such as halides, alkylating agents, epoxides, and furan rings. | Motif 0: A carbonyl group (C=O) attached to a carbon chain.
Motif 1: A nitrile group ($C \equiv N$) attached to a tertiary carbon.
Motif 2: A di-substituted carbon chain with two adjacent nitrile groups (N=C-C=C-N).
...
Motif 23: Another benzene ring structure. |
| I want you to think like a chemist performing a detailed analysis of the chemical composition of [1] through its constituent motifs. I will highlight for you some of the distinctive substructures of [1]. They are numbered from 0. Here are the textual descriptions of each motif: <1> I want you to pick only ONE of these as the root motif most essential to its chemical profile. It should be the single most important motif the rest of [1] was built around. [2], so your selected root motif should contain those group(s). If there are multiple such motifs, or one doesn't clearly stand out, just pick one of them. Give your answer as a single number. Explain your reasoning carefully. | a toxic compound

This molecule belongs to a collection of molecules characterized by distinct functional groups known for their carcinogenic properties or liver toxicity. These groups comprise a rich variety of elements, most notably halides | Motif 0: A carbonyl group (C=O) attached to a carbon chain.
Motif 1: A nitrile group ($C \equiv N$) attached to a tertiary carbon.
Motif 2: A di-substituted carbon chain with two adjacent nitrile groups (N=C-C=C-N).
...
Motif 23: Another benzene ring structure. |
| I want you to think like a chemist performing a detailed analysis of the chemical composition of [1] through its constituent motifs. This requires two steps: 1) analyzing the individual motifs and 2) analyzing the pairwise interactions of motifs. The first step is already done. The second step is where I need your help. I will highlight for you different motif interactions within the same molecule. These interactions are numbered one-by-one, beginning with 0. Here are the textual descriptions of each motif interaction pair. <1> I want you to tell me which one of these is MOST important, and which one of these is LEAST important. Output one number identifying the MOST important, and give a brief explanation. Output one number identifying the LEAST important, and a brief explanation why. | a toxic compound | Interaction 0 features a benzene and an isobutyraldehyde.
Interaction 1 features an isobutyraldehyde and a chloroacetylene.
Interaction 2 features a chloroacetylene and a benzene. |

*Table 6.* Reference table of prompts used to create and compare design stories (Section 3.5).

| Prompt | Example [substitutions] | Example <substitutions> |
|---|---|---|
| I want you to think like a chemist performing a detailed analysis of the chemical composition of [1] through its constituent motifs. I will highlight for you a specific pair of motifs, the positive motif in red and the negative motif in green. I want you to explain to me the importance of this positive-negative interaction through the perspective of designing [1]. Explain how the addition of the negative (green) motif to the positive (red) motif is justified in the context of designing [2]. Keep your response to a single paragraph in prose. | a toxic compound 

 This molecule belongs to a collection of molecules characterized by distinct functional groups known for their carcinogenic properties or liver toxicity. These groups comprise a rich variety of elements such as halides, alkylating agents, epoxides, and furan rings. | Motif 0: A carbonyl group (C=O) attached to a carbon chain. Motif 1: A nitrile group ($C \equiv N$) attached to a tertiary carbon. Motif 2: A di-substituted carbon chain with two adjacent nitrile groups (N=C-C=C-N). ... Motif 23: Another benzene ring structure. |
| I want you to weave the following chronological log entries of a chemist into a high-level story of how the chemist came up with a new molecule. Summarize each of the expert's individual numbered entries, keeping the rationales intact but removing any redundant, ambiguous or unnecessary information. Remove the ambiguous mentions of positive and negative motifs. Your final output should be the same numbered list of steps, but each step is just a few sentences at most. | a toxic compound 

 This molecule belongs to a collection of molecules characterized by distinct functional groups known for their carcinogenic properties or liver toxicity. These groups comprise a rich variety of elements such as halides, alkylating agents, epoxides, and furan rings. | Motif 0: A carbonyl group (C=O) attached to a carbon chain. Motif 1: A nitrile group ($C \equiv N$) attached to a tertiary carbon. Motif 2: A di-substituted carbon chain with two adjacent nitrile groups (N=C-C=C-N). ... Motif 23: Another benzene ring structure. |
| I want you to compare two competing analyses for designing the molecule <1>. Which analysis demonstrates deeper understanding of [1]? Pay attention to the following points: - Which analysis better highlights [2]? - Which analysis has better insights regarding the interactions amongst those motifs? Output only the answer (A or B) and nothing else. Now, here are the analyses: | toxic compounds implicated with carcinogenicity for male and female rats 

 the defining halide motif(s) of the toxic compound's chemical class | CC(=O)C(N=Nc1ccc(-c2ccc(N=NC(C(C)=O)C(=O)Nc3ccccc3)c(Cl)c2)cc1Cl)C(=O)Nc1ccccc1 |

## C. Additional Discussion of Evaluation Criteria

We provide additional points of consideration when interpreting the results in Sec. 4.

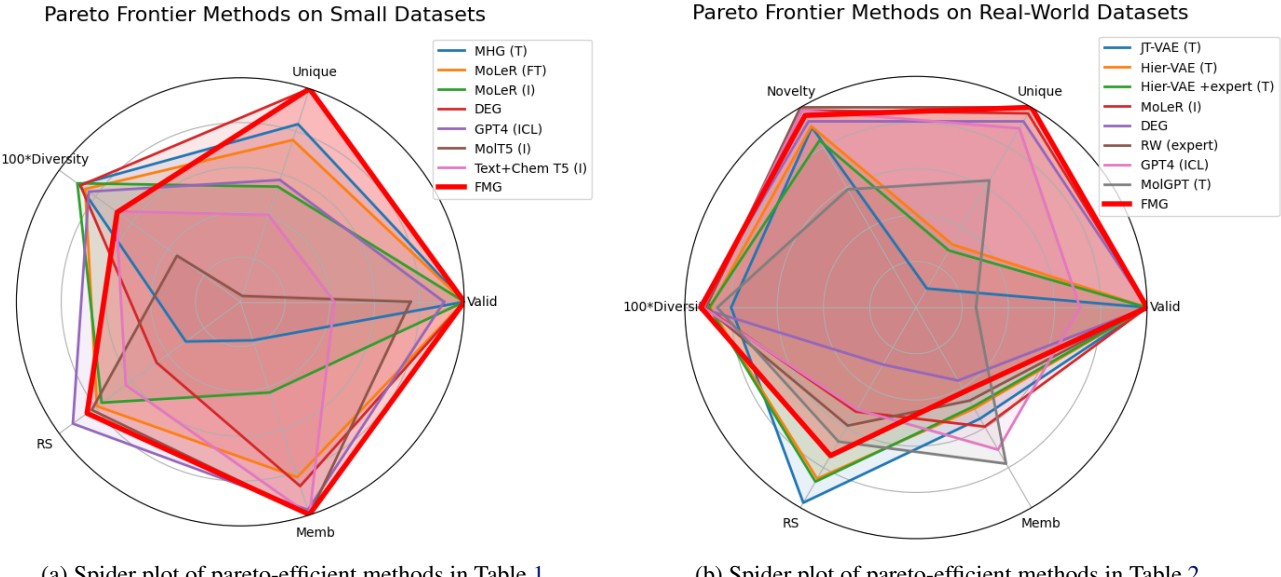

(a) Spider plot of pareto-efficient methods in Table 1.

(b) Spider plot of pareto-efficient methods in Table 2.

*Figure 5.* Visualization of results across all 5 evaluation metrics.

**Holistic Assessment.** Evaluating generative models is challenging and requires a holistic consideration of different, competing metrics. A method which scores high on one metric (e.g. synthesizability) but does catastrophically bad on another one (e.g. uniqueness) is not practical. To facilitate a more holistic validation, we provide spider plots for Tables 1 & 2 in Figures 5a & 5b. On small datasets, no method other than FMG reaches near 100% unique, valid & memb. simultaneously. For instance, GPT4 (ICL) appears to score higher on diversity & RS, but struggles to generate valid and unique samples. On real-world datasets, FMG generates the most unique, novel, valid & diverse samples, and methods that score higher on RS and Memb. have serious shortcomings.

**Ranking Criteria.** In Tab. 1, the two T5 (I) methods are excluded from ranking (as mentioned in the caption) due to the difficulty in obtaining sufficient valid, unique samples, meaning their seemingly higher RS/Memb. scores don't have sufficient sample support. The same is written in the caption of Tab. 2 for the VAE (T) methods. When simple fixes don't work (e.g., tuning sampling temperature), we make a note of it in the caption and exclude it from the rankings.

**Robustness.** We don't report error bars because robustness across samples (the purpose of error bars) can simply be absorbed into the number of samples we generate. Generating a sufficiently large sample size can ensure greater statistical significance of the results, especially when 4 of our metrics (valid, novel, RS, memb.) are defined at the individual sample level and the other 2 metrics (diversity, uniqueness) evaluate coverage. For small molecules (Tab. 1), we generate 10000 samples; for large molecules (Tab. 2), we generate 1000 samples due to inference being more expensive for some of the large model baselines. In both cases, we see a fixed sample size of 1000 or 10000 already pushes multiple baselines to the limits, so generating more than what they're capable of may further complicate a holistic assessment of the methods.

## D. Expert Evaluation of LLM Judge

### D.1. Setup

For a given molecule, the expert compares two discrepant decompositions on the basis of soundness of reasoning. We then prompt our LLM judge with chain-of-thought reasoning (Wei et al., 2022). We ask the LLM to summarize the key points and decide which decomp. is better, solely based on the explanations. For Chain Extenders, the expert compared all 11 molecules and provided a rationale for their decision for each molecule. For the 4 other datasets, we randomly select 5 molecules each, and ask the expert to directly provide the decision without writing a rationale due to time constraints. Each

pair was evaluated twice with flipped order to eliminate bias. We run the LLM judge 3 times per molecule, for a total of 15 data points per dataset.

## D.2. Example

> **Expert**
>
> "Analysis A demonstrates a solid understanding of motifs like amide, urea, imidazole, and carbonyl groups—key to chain extenders. It justifies motif 1 as the root based on its role in peptide bond formation. Analysis B neglects important groups and lacks a strong rationale for selecting motif 0 as root."
>
> **LLM Judge**
>
> "'
>
> **Defining Motifs:**
>
> Analysis A better identifies functional groups like amides/carbonyls that define chain extender properties. It correctly selects motif 1 for its role in polymer backbones.
>
> **Functional Group Explanation:**
>
> Analysis A explains the contributions of groups to mechanical and processing properties. Analysis B simplifies the motifs and lacks depth in chemical reasoning.
>
> **Decision:**
>
> Analysis A is favored for better understanding and detailed explanation of groups defining chain extenders.
> "'"

## D.3. Results

For Chain Extenders, the LLM judge agrees with the expert 10, 9, 8 times out of 11 when the better decomposition is shown last, and 8, 8, 7 times when it is shown first. For the 4 other datasets, we see the results in Table 7. The total comparisons the LLM judge agrees with the expert is $77/108 = 71\%$. With the null hypothesis being every comparison is an independent coin flip, this result yields a p-value of $1.1e - 5$, which is statistically significant.

## D.4. Analysis

The LLM judge can identify better decompositions, validating our judging protocol and the premise behind our self-improving feedback loop described in Sec. 3.5. These results reinforce the usefulness of FMG's interpretability – its outputs can support downstream decision-making and design critique, even enabling the LLM to act as a self-checking agent in an expert-in-the-loop pipeline.

*Table 7.* We tally LLM judge decisions against the human decision (Gold) across 3 repeated calls per molecule. For 6 molecules (columns with answer in [brackets]), the expert found both designs equally reasonable, so we exclude those from the total.

|  | Isocyanates | | Acrylates | | HOPV | | PTC | | Total |
|---|---|---|---|---|---|---|---|---|---|
| Gold | $A$ | $B$ | $A$ | $B$ | $A$ | $B$ | $A$ | $B$ | |
| 1 | $BAAB[B]$ | $BBBB[B]$ | $AABAB$ | $BBABB$ | $BBAAA$ | $BBBBA$ | $[B]BABA$ | $[A]BBBA$ | 25/36 |
| 2 | $BAAB[B]$ | $BBBA[A]$ | $AAAAA$ | $BBBBA$ | $BBABA$ | $BBBBA$ | $[B]BABA$ | $[B]BBAB$ | 25/36 |
| 3 | $BAAB[B]$ | $ABBA[B]$ | $AAAAB$ | $BBBBB$ | $BAAAB$ | $BBBBB$ | $[B]BABA$ | $[A]ABBB$ | 26/36 |
| Score | 6/12 | 9/12 | 12/15 | 13/15 | 9/15 | 13/15 | 6/12 | 9/12 | $77/108 = 71\%$ |

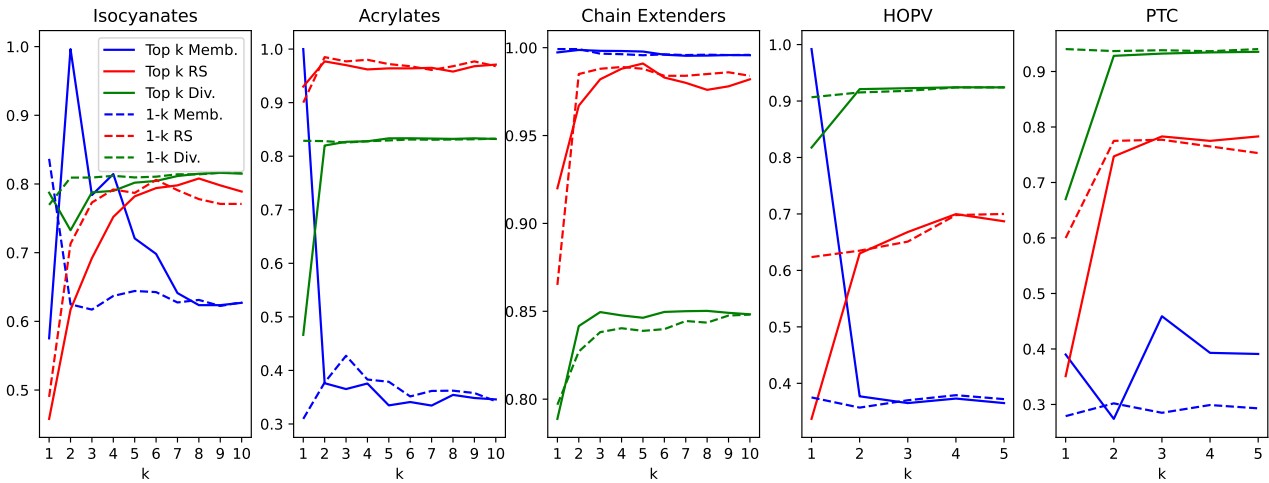

*Figure 6.* We vary $k$ from 1-10 (small dataset) and 1-5 (real-world dataset) following the same settings as the main results.

## E. Ablation: Ensemble Over Seeds

**Results.** We observe in Fig. 6 sharp tradeoffs in the generation metrics as k increases. It is easy to achieve near 100% membership for low values of k. This is because one of the points of comparison when evaluating two discrepant design stories being, "Which analysis better highlights the defining motif(s) of the acrylates chemical class?" We can deduce that 1) for each molecule, running for sufficient number of seeds always produces some decomposition that embeds the chemical class's defining motif within one of the rules, and 2) FMG is capable of ranking decompositions containing that property higher than those that do not. As a corollary, membership drops as k increases, as rules from sub-optimal decompositions are added to the grammar. In contrast, domain-specificity has some *intrinsic* tradeoff with synthesizability. Isocyanates are known to be tricky to synthesize due to unwanted side reactions. Choosing decompositions with design stories demonstrating a thorough understanding of the domain is more likely to overcomplicate the grammar from a synthesizability perspective.

We also note some general trends as k increases. Diversity and RS seem to improve as more rule sets are combined. This is likely because a larger collection of "simple" rules, pooled from alternative decompositions, enables more simple molecules to be generated, albeit at the cost of membership. Interestingly, there are no major differences between Top k and 1-k for RS and diversity, suggesting the learning procedure targets mainly class-specific considerations, remaining neutral to more general considerations.

## F. Ablation: Images vs Text

### F.1. FMG-Text Choice of Text Encoding

**Attempt 1: Tagged SMILES.** Our first attempt to use a text-only input format was to add tags (e.g. $<>$) into SMILES to denote motif boundaries. For example:

- Original SMILES: C=CC(=O)OC1CC2CC1C2
- Motif: C=CC(=O)O$<$C$>$1$<$CC$>$2C$<$C$>$1$<$C$>$2

GPT-4o failed basic comprehension tests. Though it understands that numbers denote ring closures, it became confused about which numbers were relevant to the motif's ring. As a result, it interpreted the tagged string via straightforward character concatenation – e.g., as CCCCC, a straight-chain alkane – rather than recognizing the cyclopentane motif. We attempted to add ring numbers within the tags to clarify the motif's ring structure, but this only obfuscated the global context and further degraded the model's understanding.

**Attempt 2: Atom-number encoding.** We then arrived at our more verbose: number all atoms and list motif atoms. For example:

- Original SMILES: [C:1]=[C:2][C:3](=[O:4])[O:5][C:6]1[C:7][C:8]2[C:9][C:10]1[C:11]2

- (optional) Motif 1: 6,7,8,10,11
- (optional) Motif 2: 8,9,10,11

We made sure everything worked as expected on a few case studies. Then, we ran our full evaluation protocol.

## F.2. FMG-Text Best Of Ensemble Results

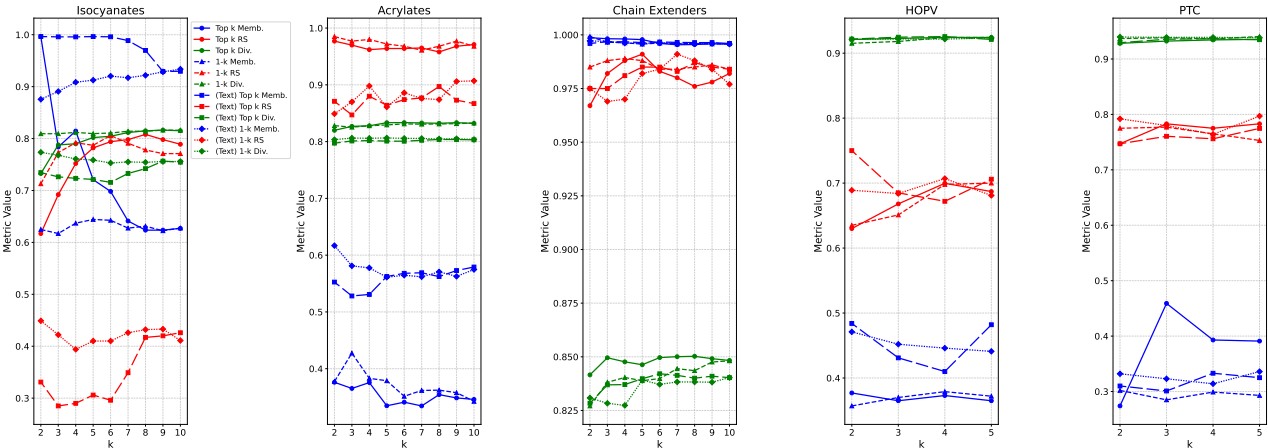

*Figure 7.* We vary $k$ from 2-10 (small dataset) and 2-5 (real-world dataset) to compare FMG-Text with FMG.

The full results of varying $k$ for FMG-Text, in comparison to the results in App. E, is shown in Fig. 7.

## F.3. Motivation for Images Over Text

In addition to the quantitative edge FMG has over FMG-Text, we also provide additional points of consideration for choosing images as the format of representation over text.

1. **Literature review of LLM's chemistry comprehension abilities.** Foundation models like GPT-4o and Gemini have shown multi-modal comprehension ability in aligning natural language descriptions with corresponding images for advanced reasoning, but there has been less work on aligning formal languages like SMILES with corresponding images. This may be because LLMs find it easier to obtain semantics from natural descriptions rather than SMILES. For instance, studies like that done by White et al. (2023) (Section D) and Guo et al. (2023) find LLMs perform great on tasks that require reasoning from the natural description of molecules, but struggle when fed SMILES/SELFIES (e.g. 0% SMILES-to-name success in the first study).

2. **Technical formulation is in terms of hypergraphs.** Our underlying formulation builds off the history of hyperedge-replacement grammars, which operate at the hyperedge (or substructure) level instead of the atom level. SMILES/-SELFIES syntax doesn't easily support substructure-level annotations and extending it to do so not only is out of the scope of this work but also changes the data used for LLM pre-training. Meanwhile, rdkit has a library of functions for highlighting, color-coding, and rendering substructures. SMILES/SELFIES is also incompatible with a particular detail in our formulation, which treats bonds (not atoms) as the fundamental units. Our design choice ensures that each clique's atoms (edges) are disjoint, which is essential for our grammar induction and molecule reconstruction. Because SMILES/SELFIES encodes atoms explicitly but treat bonds implicitly, there's no straightforward way to annotate bonds or define disjoint cliques. Enforcing compatibility would require taking the bond-atom dual of (1) our grammar formulation or (2) the SMILES/SELFIES syntax. This mismatch, discussed in the preliminaries, further motivates our use of images, where we can easily highlight constituent bonds.

3. **Interpretability of the grammar learning process.** We thought long and hard about how the complex, hierarchical structured representation of hypergraphs can be fed into LLMs. After initial conversations with chemists, we came to the conclusion that highlighting substructures is the most visually meaningful way to consume the information. We, in parallel with a few others like Wang et al. (2025), identified an opportunity for combining the latest multi-modal understanding and cheminformatics rendering tools. We hope these points sufficiently motivate the visual representation input, and a summarized discussion will be added to the main text.

# G. Results on a Small Subset of MOSES

*Table 8.* We trained FMG on a 1k subset (0.05%) of the refined ZINC dataset used by the MOSES benchmark (Polykovskiy et al., 2020). We generated 30k samples and computed standard -TestSF metrics using the released splits. Baseline numbers are copied from the MOSES leaderboard. IntDiv2 is omitted due to redundancy with IntDiv.

| Model | Valid($\uparrow$) | Unique@1k($\uparrow$) | Unique@10k($\uparrow$) | FCD-TestSF($\downarrow$) | SNN-TestSF($\uparrow$) | Scaf-TestSF($\uparrow$) | IntDiv($\uparrow$) | Novelty($\uparrow$) |
|---|---|---|---|---|---|---|---|---|
| Train | 1.00 | 1.00 | 1.00 | 0.48 | 0.59 | 0.00 | 0.86 | 1.00 |
| HMM | 0.08±0.03 | 0.62±0.12 | 0.57±0.14 | 25.43±2.56 | 0.38±0.01 | 0.05±0.02 | 0.85±0.04 | **1.00±0.00** |
| NGram | 0.24±0.00 | 0.97±0.01 | 0.92±0.00 | 6.23±0.10 | 0.50±0.00 | 0.10±0.01 | 0.87±0.00 | 0.97±0.00 |
| Combinatorial | **1.00±0.00** | **1.00±0.00** | 0.99±0.00 | 4.51±0.03 | 0.44±0.00 | 0.09±0.00 | 0.87±0.00 | 0.99±0.00 |
| CharRNN | 0.97±0.03 | **1.00±0.00** | **1.00±0.00** | **0.52±0.04** | 0.56±0.01 | 0.11±0.01 | 0.86±0.00 | 0.84±0.05 |
| AAE | 0.94±0.03 | **1.00±0.00** | **1.00±0.00** | 1.06±0.24 | 0.57±0.00 | 0.08±0.01 | 0.86±0.00 | 0.79±0.03 |
| VAE | 0.98±0.00 | **1.00±0.00** | **1.00±0.00** | 0.57±0.03 | **0.58±0.00** | 0.06±0.01 | 0.86±0.00 | 0.69±0.01 |
| JTN-VAE | **1.00±0.00** | **1.00±0.00** | **1.00±0.00** | 0.94±0.05 | 0.52±0.01 | 0.10±0.01 | 0.86±0.00 | 0.91±0.01 |
| LatentGAN | 0.90±0.00 | **1.00±0.00** | **1.00±0.00** | 0.83±0.01 | 0.51±0.00 | 0.11±0.01 | 0.86±0.00 | 0.95±0.00 |
| FMG (0.05%) | **1.00±0.00** | **1.00±0.00** | **1.00±0.00** | 26.30±0.41 | 0.29±0.00 | **0.12±0.00** | **0.90±0.00** | **1.00±0.00** |

**Setup.** The sole purpose of this study is completeness. We want to understand FMG's behavior on the popular MOSES benchmark used for large-scale representation learning. We trained FMG on three 1k subsets (just 0.05%) of the 1.9 million molecules in the refined ZINC dataset used by MOSES (Polykovskiy et al., 2020). This reflects an extremely data-scarce setting, so direct comparisons should not be made; we only include the leaderboard for broader context.

**Results.** We can see from Table 8 that FMG excels on all five unconditional generation metrics reported in our main paper. That said, we acknowledge its distributional match is very weak, as expected from a model trained on only 0.05% of the data. However, its high validity, novelty, diversity, and uniqueness demonstrate FMG's potential as a backbone model for further optimization (as done by Guo et al. (2022b)).

**Discussion.** We want to reiterate the pressing issues in real-world molecular discovery that FMG was designed to solve. We believe there lies a critical gap between large-scale representation learning benchmarks and real-world molecular design settings. Real-world settings often come with only a handful of training examples, because chemical classes feature more specific criteria and there are fewer experts to validate the data. Our main results in Tables 1 & 2 demonstrate FMG outperforming methods which are state-of-the-art on large-scale benchmarks. We see their performances drop significantly due to failing constraints like synthesizability and class membership. Maintaining diversity also becomes more challenging with less data. When tackling domain-specific settings, e.g. synthetically accessible chain extenders with amine groups, expert knowledge plays a critical role – but integrating it via annotations traditionally required costly manual labor (Sun et al., 2024). FMG seeks to automate this labor with MMFMs, constructing rigorous hierarchical languages that capture explicit/implicit constraints through step-by-step algorithmic reasoning. The result is an interpretable, knowledge-driven workflow that excels at step-by-step reasoning for domain-specific molecular design.

# H. Case Studies, Expert Analyses, and Design Story

In the Appendix, we do turn-by-turn walkthroughs of representative molecules from every dataset. We show logs of every task-related (prompt, response) pair used in our algorithm. We invite a real chemist to comment on the algorithm's logs for each dataset and comment on a) the difficulty of the prompt and b) GPT's answers. We preface each dataset (subsection) with a brief overview of the domain and rationale for choosing the specific molecule. For the interested reader, we also include a sample design story in Section H.2.1 used in learning the FMG. As the unrefined stories are heavy in jargon and quite elaborative, we only include one example. We choose a HOPV molecule as they feature the richest stories. We include the full text under HOPV's Section H.2.

In Section H.7, we leave with concluding remarks on the overall performance of GPT-4o, highlighting LLM agents' strong performance in substructure extraction and limits in harder tasks requiring expert intuition.

## H.1. Case Study: Predictive Toxicology Challenge (PTC)

First, we establish some formatting conventions that we use for PTC and HOPV:

• We log the task prompt, the corresponding response and the expert's comments for each call to GPT-4o, subdivided by

header cells labeling what's in the following cell.
- We omit utility prompts like extracting integer answers and instead focus on the core selection tasks done by GPT.
- Dynamic substitutions are quoted in "'...'".
- Pre-written background specific to the domain is *italicized*.

For PTC, since it often contain halides and quaternary ammonium groups, which assist in transferring reactants between different phases in a reaction, the selected molecule picked not only have multiple symmetrical halide component but are also comprehensive enough for GPT to studying the rest of functional motif's relative impact to the entire molecule, making it an ideal candidate for studying phase-transfer catalysis in synthetic chemistry and investigating toxicity, given that some PTCs are known carcinogens.

*Figure 8.* Input molecule from PTC

## Prompt

I want you to think like a chemist performing a detailed analysis of the chemical composition of a toxic compound through its constituent motifs. I will highlight for you "'24'" of the substructures of a molecule. They are numbered one-by-one from Motif 0 to Motif "'23'", inclusive. I want you to explain, concisely, what each numbered motif is. Make sure to start from Motif 0 and go in order of the numbering. MAKE SURE you describe EVERY MOTIF!

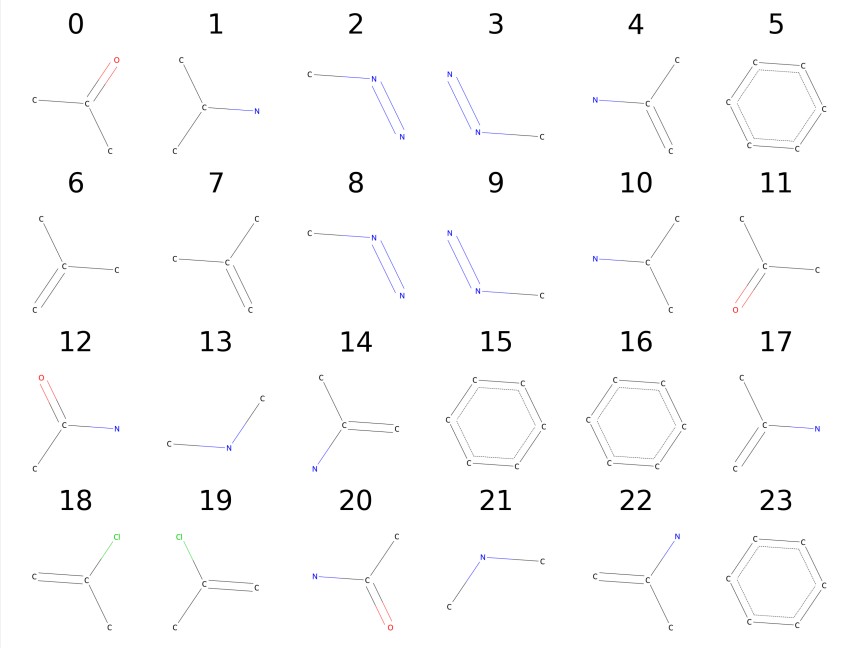

## Response

Certainly! Here is a concise description of each motif:

**Motif 0:** A carbonyl group (C=O) attached to a carbon chain.

**Motif 1:** A nitrile group ($C \equiv N$) attached to a tertiary carbon.

**Motif 2:** A di-substituted carbon chain with two adjacent nitrile groups (N=C-C=C-N).

**Motif 3:** Similar to Motif 2 but with an additional carbon on one end.

**Motif 4:** An imine group (C=N) connected to a vinylic carbon chain (C=C).

**Motif 5:** Benzene ring (C6H6) - a six-membered aromatic carbon ring.

**Motif 6:** A singly unsaturated carbon chain.

**Motif 7:** A doubly unsaturated carbon chain.

**Motif 8:** Di-substituted carbon chain with a nitrile group (N=C) on one end and another nitrile ($N \equiv C$) in the middle.

**Motif 9:** Similar to Motif 8 but with different placement of carbons.

**Motif 10:** An amino group (NH2) attached to a C=N group (imino).

**Motif 11:** A ketone (C=O) with adjacent carbons.

**Motif 12:** A cyano-ketone ($C \equiv N$ and C=O) attached to adjacent carbons.

**Motif 13:** An alkyl cyanide ($R - C \equiv N$) where R is an alkyl chain.

**Motif 14:** An amino group (NH2) connected to a nitrile ($C \equiv N$).

**Motif 15:** A benzene ring (C6H6) - slightly different representation of Motif 5.

**Motif 16:** Another representation of a benzene ring (C6H6).

**Motif 17:** A nitrile group ($C \equiv N$) attached to a doubly unsaturated carbon chain.

**Motif 18:** An alkyl halide (carbon chain with a chlorine, C-Cl).

**Motif 19:** Similar to Motif 18 with a different number of carbons.

**Motif 20:** A doubly canned ketone (C=O) group attached to an imine (C=N) group.

**Motif 21:** An imine group (C=N) attached to an ethyl group (C2H5).
**Motif 22:** A doubly bonded carbon chain similar to an alkene.
**Motif 23:** Another benzene ring structure.
This list consists of a variety of functional groups and structural motifs commonly found in organic chemistry, ranging from aromatic rings to carbon-nitrogen triple bonds, and functional groups such as ketones and amines.

### Comments by Expert

The response thoroughly and accurately identified and described each of the 24 motifs, spanning a range of functional groups and structural components. These include highly reactive entities such as carbonyls, nitriles, and imines, as well as stable aromatic systems like benzene rings. Each motif was correctly categorized with its chemical identity and described in terms of its structure and connectivity, reflecting a solid understanding of organic chemistry principles.

### Prompt

I want you to think like a chemist performing a detailed analysis of the chemical composition of a toxic compound through its constituent motifs. I will highlight for you some of the distinctive fragments of a molecule. They are numbered from 0 and individually highlighted in GREEN. Focus ONLY on the substructure highlighted in green within each cell. Here is the descriptions for each substructure provided by an expert:
"'Certainly! Here is a concise description of each motif:
**Motif 0:** A carbonyl group (C=O) attached to a carbon chain.
**Motif 1:** A nitrile group ($C \equiv N$) attached to a tertiary carbon.
**Motif 2:** A di-substituted carbon chain with two adjacent nitrile groups (N=C-C=C-N).
...
**Motif 23:** Another benzene ring structure.
This list consists of a variety of functional groups and structural motifs commonly found in organic chemistry, ranging from aromatic rings to carbon-nitrogen triple bonds, and functional groups such as ketones and amines.'"
I want you to tell me if any two of them should be combined together to form a more meaningful substructure. *This molecule belongs to a collection of molecules characterized by distinct functional groups known for their carcinogenic properties or liver toxicity. These groups comprise a rich variety of elements such as halides, alkylating agents, epoxides, and furan rings.* Your task is to highlight the primary functional groups of the molecule. Output a single pair of numbers if you think those two fragments should be combined, and a brief explanation why. If no such pairs exist, don't output anything.

### Response

**Pair: 8 and 9**

**Explanation:** Motifs 8 and 9 are both di-substituted carbon chains with nitrile groups in different positions. Combining these two fragments could provide a more comprehensive understanding of the arrangement and connectivity of nitrile groups along the carbon chain, which is critical in analyzing the structure and reactivity of the molecule, especially in the context of toxic compounds where such functional groups can have significant implications for toxicity.

**Comments by Expert**

Medium, PTC molecules are more complex, so it requires considering combinations of more than one pair of motifs. This is because molecules like this often contain interconnected functional groups that work synergistically to influence their chemical properties, for example, liver toxicity or cancerogenetics.

**Prompt**

I want you to think like a chemist performing a detailed analysis of the chemical composition of a *toxic compound* through its constituent motifs. I will highlight for you "'23'" of the substructures of a molecule. They are numbered one-by-one from Motif 0 to Motif "'22'", inclusive. I want you to explain, concisely, what each numbered motif is. Make sure to start from Motif 0 and go in order of the numbering. MAKE SURE you describe EVERY MOTIF!

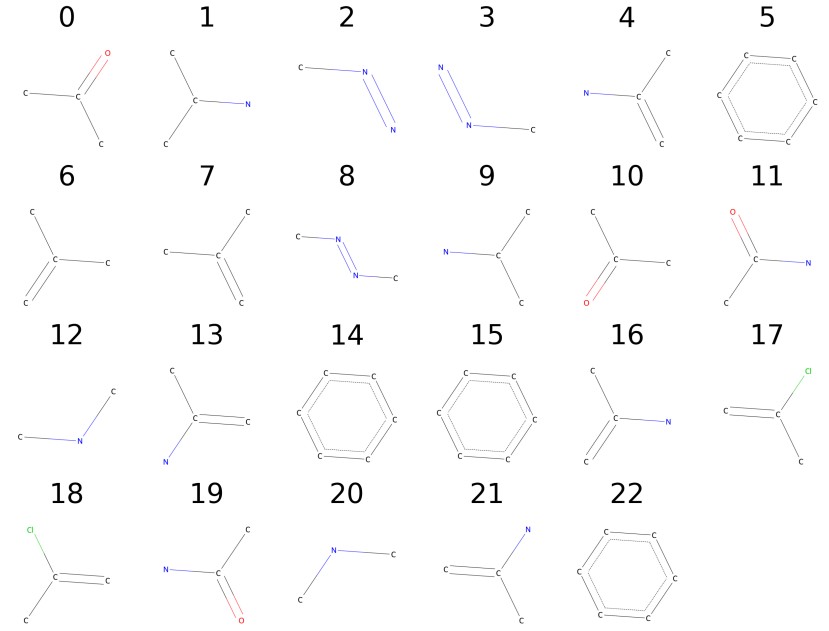

**Response**

Certainly! Here are the chemical motifs numbered from 0 to 22:
**Motif 0**: Acetone (propan-2-one) - A simple ketone with the formula $(CH_3)_2CO$.
**Motif 1**: Acetonitrile - A nitrile compound where the CN group is attached to a methyl group.
**Motif 2**: Fumaroid nitrile - A dinitrile with both nitrile groups on opposite sides.
**Motif 3**: Cis-maleonitrile - A dinitrile with both nitrile groups on the same side.
**Motif 4**: Acrylonitrile - A nitrile compound with a vinyl group attached to the CN.
**Motif 5**: Benzene - An aromatic hydrocarbon with a six-membered carbon ring and alternating double bonds (represented here with a hexagon).
**Motif 6**: Propene (propylene) - An alkene with the chemical formula $C_3H_6$ containing a double bond between two carbon atoms.
**Motif 7**: 2-Butene - An alkene which has a double bond between the second and third carbons of a four-carbon chain.
**Motif 8**: Butanedinitrile - A nitrile with a CN group at each end of a butane backbone.
**Motif 9**: Acrylonitrile derivative - Similar to acrylonitrile with an additional methyl group.

**Motif 10**: Methacrolein - An aldehyde with a carbon-carbon double bond.

**Motif 11**: Ethyl cyanide (propanenitrile) - A nitrile with a three-carbon chain.

**Motif 12**: Propan-1-amine - An amine with a three-carbon chain.

**Motif 13**: Tert-butyl - A bulky, branched alkyl group with three methyl groups attached to a single carbon atom.

**Motif 14**: Benzene (repetition of Motif 5) - Another illustration of benzene.

**Motif 15**: Benzene (repetition of Motif 5 and 14) - Yet another illustration of benzene.

**Motif 16**: Isobutyraldehyde - An aldehyde with a branched three-carbon chain.

**Motif 17**: Chloroacetylene - An alkyne with chlorine attached to a carbon-carbon triple bond.

**Motif 18**: 1,1-Dichloroethane - An alkane with two chlorine atoms attached to the same carbon atom.

**Motif 19**: 2-Methylpropanoylnitrile - A nitrile with a methyl group attached to a carbon atom that is part of the nitrile structure.

**Motif 20**: Benzene (repetition of Motifs 5, 14, and 15) - Another repetition of benzene.

**Motif 21**: Acetonitrile again (as in Motif 1) - Another nitrile example.

**Motif 22**: Benzene - final repetition of the benzene structure.

This detailed description covers the unique structural motifs presented in the molecule, from acetone to benzene, and other functional groups like nitriles and alkenes.

### Prompt

I want you to think like a chemist performing a detailed analysis of the chemical composition of a *toxic compound* through its constituent motifs. I will highlight for you some of the distinctive fragments of a molecule. They are numbered from 0 and individually highlighted in GREEN. Focus ONLY on the substructure highlighted in green within each cell. Here is the descriptions for each substructure provided by an expert:

"'Certainly! Here are the chemical motifs numbered from 0 to 22:

**Motif 0**: Acetone (propan-2-one) - A simple ketone with the formula $(CH_3)_2CO$.

**Motif 1**: Acetonitrile - A nitrile compound where the CN group is attached to a methyl group.

... **Motif 22**: Benzene - final repetition of the benzene structure.

This detailed description covers the unique structural motifs presented in the molecule, from acetone to benzene, and other functional groups like nitriles and alkenes.'"

I want you to tell me if any two of them should be combined together to form a more meaningful substructure. *This molecule belongs to a collection of molecules characterized by distinct functional groups known for their carcinogenic properties or liver toxicity. These groups comprise a rich variety of elements such as halides, alkylating agents, epoxides, and furan rings.* Your task is to highlight the primary functional groups of the molecule. Output a single pair of numbers if you think those two fragments should be combined, and a brief explanation why. If no such pairs exist, don't output anything.

**Response**

**Pair: 2, 3**

**Explanation:** Motif 2 and Motif 3 both depict dinitriles, with variations in their positioning (fumaroid vs. cis-maleonitrile, respectively). Combining these motifs could result in a more complete representation of a molecule with multiple nitrile groups, which are significant regarding potential toxic and carcinogenic properties. This merged substructure can serve as a more insightful representation of the functional group within the overall toxic compound.

**Comments by Expert**

This is partially correct, while combining Motifs 2 and 3 is valid due to their shared nitrile functionality, since Motifs 17 and 18 contain halide groups, they can also be combined. Halide-containing motifs often play a significant role in the reactivity and toxicity of a compound, especially in mechanisms involving alkylating agents or electrophilic substitution reactions.

**Prompt**

I want you to think like a chemist performing a detailed analysis of the chemical composition of a *toxic compound* through its constituent motifs. I will highlight for you some of the distinctive substructures of a *toxic compound*. They are numbered from 0. Here are the textual descriptions of each motif:

"'Certainly! Here are the chemical motifs numbered from 0 to 22:

**Motif 0**: Acetone (propan-2-one) - A simple ketone with the formula $(CH_3)_2CO$.

**Motif 1**: Acetonitrile - A nitrile compound where the CN group is attached to a methyl group.

...

**Motif 22**: Benzene - final repetition of the benzene structure.

This detailed description covers the unique structural motifs presented in the molecule, from acetone to benzene, and other functional groups like nitriles and alkenes."'

I want you to pick only ONE of these as the root motif most essential to its chemical profile. It should be the single most important motif the rest of the *toxic compound* was built around. *This molecule belongs to a collection of molecules characterized by distinct functional groups known for their carcinogenic properties or liver toxicity. These groups comprise a rich variety of elements, most notably halides, so your selected root motif MUST contain a halide group.* If there are multiple such motifs, or one doesn't clearly stand out, just pick one of them. Give your answer as a single number. Explain your reasoning carefully.

**Comments by Expert**

Medium level difficulty task for the language model or even for chemistry experts, pick only one of the most important motif can cause somewhat different thoughts between language models and chemistry experts, this is in particular significant when multiple motifs carry significant relevance, such as halide-containing groups. Chemists

might emphasize functional groups based on different perspectives including reactivity, toxicity mechanisms, or known biological interactions.

**Response**

Upon reviewing the chemical motifs, the selection criteria specify that the root motif should contain a halide group, which is characteristic of the molecule's toxic properties.

Examining through the motifs, certain ones include halides:
- **Motif 0**: Acetone does not have a halide.
- **Motif 1**: Acetonitrile does not have a halide.
- **Motif 2**: Fumaroid nitrile does not have a halide.
- **Motif 3**: Cis-maleonitrile does not have a halide.
- **Motif 4**: Acrylonitrile does not have a halide.
- **Motif 5**: Benzene does not have a halide.
- **Motif 6**: Propene does not have a halide.
- **Motif 7**: 2-Butene does not have a halide.
- **Motif 8**: Butanedinitrile does not have a halide.
- **Motif 9**: Acrylonitrile derivative does not have a halide.
- **Motif 10**: Methacrolein does not have a halide.
- **Motif 11**: Ethyl cyanide does not have a halide.
- **Motif 12**: Propan-1-amine does not have a halide.
- **Motif 13**: Tert-butyl does not have a halide.
- **Motif 14**: Benzene repetition does not have a halide.
- **Motif 15**: Benzene repetition does not have a halide.
- **Motif 16**: Isobutyraldehyde does not have a halide.
- **Motif 17**: Chloroacetylene **contains a halide**.
- **Motif 18**: 1,1-Dichloroethane **contains a halide**.
- **Motif 19**: 2-Methylpropanoylnitrile does not have a halide.
- **Motif 20**: Benzene repetition does not have a halide.
- **Motif 21**: Acetonitrile repetition does not have a halide.
- **Motif 22**: Benzene repetition does not have a halide.

From the motifs listed, **Motif 17 (Chloroacetylene)** and **Motif 18 (1,1-Dichloroethane)** contain halide groups. Both of these could potentially serve as the root due to their inclusion of a chlorine atom, which is a halide. Given the need to select one motif, **Motif 18** (1,1-Dichloroethane) stands out slightly more due to the presence of two chlorine atoms, which could contribute to heightened toxicity and distinct chemical properties of the toxic compound.

Thus, the most essential root motif most likely to be built around is **Motif 18**.

**Comments by Expert**

GPT describe both motif 17 and 18, as they both contains the halide function groups. GPT choose Motif 18 as the most essential root motif, which is correct. this is because 1,1-Dichloroethane obtain higher significancy due to the presence of two chlorine atoms, which enhances its toxic potential and chemical reactivity compared to the single halide in Chloroacetylene.

**Prompt**

I want you to perform a simple post-processing step of the following response:

"'Certainly! Here are the chemical motifs numbered from 0 to 22:

**Motif 0**: Acetone (propan-2-one) - A simple ketone with the formula $(CH_3)_2CO$.

**Motif 1**: Acetonitrile - A nitrile compound where the CN group is attached to a methyl group.

...

**Motif 22**: Benzene - final repetition of the benzene structure.

This detailed description covers the unique structural motifs presented in the molecule, from acetone to benzene, and other functional groups like nitriles and alkenes.'" The input is a response from another language agent. It describes motifs numbered from Motif 0 to "'Motif 22'", inclusive! I want you to rephrase each motif description by

filling in X within the following sentence template: This motif is X Be sure to condense the description and output a single PHRASE such that the sentence template is grammatically correct. Don't capitalize the first letter, since your answer should just be a phrase. Output your rephrasing for each motif on a SEPARATE line, using only a new line for delimiting different motifs. Don't output anything else. MAKE SURE you do it for EVERY MOTIF!

**Comments by Expert**

The task is straightforward and requires minimal complexity, as it mainly involves rephrasing the descriptions of chemical motifs into a predefined format while retaining their original meaning. This type of prompt is simple to execute for anyone with fundamental chemistry knowledge.

**Response**

an acetone (propan-2-one)
an acetonitrile
a fumaroid nitrile
a cis-maleonitrile
an acrylonitrile
a benzene
a propene (propylene)
a 2-butene
a butanedinitrile
an acrylonitrile derivative
a methacrolein
an ethyl cyanide (propanenitrile)
a propan-1-amine
a tert-butyl
a benzene
a benzene
an isobutyraldehyde
a chloroacetylene
a 1,1-dichloroethane
a 2-methylpropanoylnitrile
a benzene
an acetonitrile
a benzene

**Comments by Expert**

It provides an accurate and concise summary of all the functional groups, ensuring that each motif is clearly identified while maintaining the integrity of the chemical descriptions. This demonstrates a good understanding of the understanding for diversity of structural-functional relationship within the molecule.

**Prompt**

I want you to think like a chemist performing a detailed analysis of the chemical composition of a molecule through its constituent motifs. This requires two steps: 1) analyzing the individual motifs and 2) analyzing the pairwise interactions of motifs. The first step is already done. The second step is where I need your help. I will highlight for you different motif interactions within the same molecule. These interactions are numbered one-by-one, beginning with 0. Here are the textual descriptions of each motif interaction pair.
Interaction "'0'" features "'a benzene'" and "'an isobutyraldehyde'".
Interaction "'1'" features "'an isobutyraldehyde'" and "'a chloroacetylene'".
Interaction "'2'" features "'a chloroacetylene'" and "'a benzene'".
I want you to tell me which one of these is MOST important, and which one of these is LEAST important. Output one number identifying the MOST important, and give a brief explanation. Output one number identifying the LEAST important, and a brief explanation why.

**Comments by Expert**

This is a hard task, as it requires abundant chemistry empirical experiences in organic synthesis, particularly in understanding how different motifs influence molecular stability, reactivity, and interactions. Factors such as the electronic effects of the functional groups, steric hindrance, resonance stability, and the roles of motifs in each synthetic reaction or reaction mechanisms can vary, which all contributes to the complexity of motif interactions.

**Response**

To determine which interaction is the most and least important, it's useful to consider the chemical properties and potential intermolecular forces involved in these interactions.
### Interaction Details: 1. **Interaction 0**: Benzene and Isobutyraldehyde
2. **Interaction 1**: Isobutyraldehyde and Chloroacetylene
3. **Interaction 2**: Chloroacetylene and Benzene
### Analysis: - **Benzene**: Benzene rings often participate in $\pi - \pi$ stacking interactions, increasing molecular stability and electronic distribution. - **Isobutyraldehyde**: This functional group has a polar aldehyde component that can form hydrogen bonds and dipole-dipole interactions. - **Chloroacetylene**: This highly reactive group contains both an acidic hydrogen (from the acetylene) and a chloro substituent which may influence its reactivity and interaction propensity.
### Pairwise Interactions: - **Interaction 0 (Benzene and Isobutyraldehyde)**: This interaction may involve weak $\pi - \pi$ interactions from benzene and dipole-dipole attractions from the aldehyde group.
- **Interaction 1 (Isobutyraldehyde and Chloroacetylene)**: The aldehyde offers dipole-dipole interactions while the chloroacetylene provides potential for bonding interactions based on its triple bond and electronegative chloro group.
- **Interaction 2 (Chloroacetylene and Benzene)**: This interaction might involve $\pi - \pi*$ interactions from benzene and possible dipole-induced interactions from chloroacetylene.
### Most Important: **Interaction 1 (Isobutyraldehyde and Chloroacetylene)** Explanation: This interaction involves strong dipole-dipole interactions from the polar aldehyde group and the highly reactive chloroacetylene. The triple bond and the chlorine atom enhance the electrophilic character, making this interaction highly influential in the molecule's reactivity and stability.
### Least Important: **Interaction 0 (Benzene and Isobutyraldehyde)** Explanation: While benzene can participate in $\pi - \pi$ interactions, these non-covalent interactions are generally weaker compared to strong dipole-dipole or potential covalent interactions found in other pairs. The electrostatic attraction between benzene's $\pi-$ electrons and the aldehyde may not contribute as significantly to the molecule's overall properties.
### Summary: - **Most Important: Interaction 1** - **Least Important: Interaction 0**

**Comments by Expert**

This is fully correct, interaction 1 ( benzene and a 1,1-dichloroethane ) should be most important and 0 (acrylonitrile and a benzene) should be the least in this case. This assessment aligns with the chemical reasoning that Interaction 1 involves significant dipole-dipole interactions due to the presence of halides and a benzene ring, making it crucial to the molecule's overall properties. In contrast, Interaction 0 is less important, as the $\pi - \pi$ interactions between benzene and acrylonitrile are comparatively weaker, contributing less to molecular stability or reactivity. This distinction reflects a deep understanding of the underlying chemical principles governing the interactions.

**Prompt**

I want you to think like a chemist performing a detailed analysis of the chemical composition of a molecule through its constituent motifs. This requires two steps: 1) analyzing the individual motifs and 2) analyzing the pairwise interactions of motifs. The first step is already done. The second step is where I need your help. I will highlight for you different motif interactions within the same molecule. These interactions are numbered one-by-one, beginning

with 0. Here are the textual descriptions of each motif interaction pair.

Interaction "'0'" features "'an acrylonitrile'" and "'a benzene'".

Interaction "'1'" features "'a benzene'" and a "'1,1-dichloroethane'".

Interaction "'2'" features "'a 1,1-dichloroethane'" and "'an acrylonitrile'".

I want you to tell me which one of these is MOST important, and which one of these is LEAST important. Output one number identifying the MOST important, and give a brief explanation. Output one number identifying the LEAST important, and a brief explanation why.

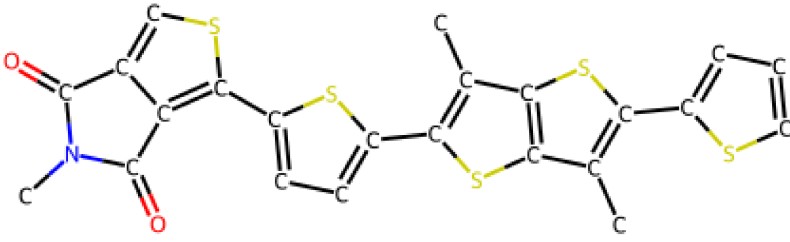

**Response**

**Most Important: Interaction 1**

Explanation: Interaction 1 features a benzene and a 1,1-dichloroethane. Benzene is a highly stable aromatic compound, and its interactions are often significant due to its resonance stability and the potential for $\pi - \pi$ stacking interactions. 1,1-Dichloroethane can interact through dipole-dipole interactions due to the presence of electronegative chlorine atoms. The combination of these two motifs could provide strong intermolecular or intramolecular interactions, impacting the overall stability and properties of the molecule significantly.

**Least Important: Interaction 0**

Explanation: Interaction 0 features an acrylonitrile and a benzene. Acrylonitrile, while reactive due to its nitrile group, may not interact as strongly with benzene compared to the involvement of more polar or reactive groups like the dichloroethane. The interaction with benzene here would primarily be through weaker van der Waals forces and does not have the significant dipole-dipole interactions present in Interaction 1.

## H.2. Case Study: Harvard Organic Photovoltaic Dataset (HOPV)

For HOPV, the selected molecules include several of the most significant functional motifs that in contributing to molecules' photovoltaic properties. These functional groups include the classical thiolphene unit, thiazole ring, pyrimidinedione ring and dithieno[3,2-b:2',3'-d]thiophene (DTT), it is a really classical, comprehensive and representative HOPV molecule.

*Figure 9.* Input molecule from HOPV

**Prompt**

I want you to think like a chemist performing a detailed analysis of the chemical composition of a *photovoltaic molecule* through its constituent motifs. I will highlight for you 11 of the substructures of a molecule. They are numbered one-by-one from Motif 0 to Motif "'10'", inclusive. I want you to explain, concisely, what each numbered motif is. Make sure to start from Motif 0 and go in order of the numbering. MAKE SURE you describe EVERY MOTIF!

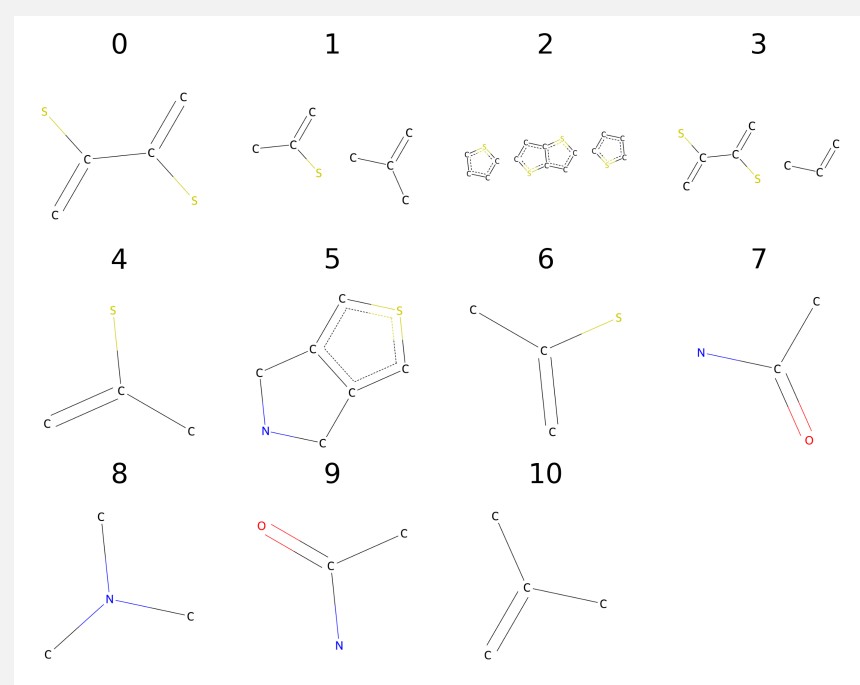

**Response**

Here's the detailed analysis of each motif in the given photovoltaic molecule, broken down:

1. **Motif 0**: - **Structure**: It is a 1,3-dithiol-2-one structural motif. It features a ketone (carbonyl group) with a five-membered ring containing two sulfur atoms adjacent to the ketone group.

2. **Motif 1**: - **Structure**: This motif consists of two distinct substructures: 1. A five-membered aromatic ring resembling thiophene, composed of a sulfur atom (yellow) and four carbon atoms. 2. A simple three-carbon chain with a double bond, ending in a ketone group (carbonyl group).

3. **Motif 2**: - **Structure**: This motif features a set of three different heterocyclic aromatic rings: 1. A thiophene ring (five-membered ring with sulfur). 2. A furan ring (five-membered ring with an oxygen atom). 3. A pyrrole ring (five-membered ring with a nitrogen atom).

4. **Motif 3**: - **Structure**: This is similar to Motif 0 but includes an additional carbonyl group (C=O) branching off the central carbon in the five-membered ring structure.

5. **Motif 4**: - **Structure**: This motif resembles a stylized thioacetone structure. It consists of a three-carbon chain with a sulfur atom connected to the central carbon and two other carbon atoms attached via single bonds.

6. **Motif 5**: - **Structure**: This motif features an indole-like structure. It is a bicyclic system with a benzene ring fused to a five-membered ring containing a nitrogen atom and a sulfur atom, resembling benzothiophene.

7. **Motif 6**: - **Structure**: This is a thioketo structure. It consists of a three-carbon chain with a sulfur atom double-bonded to the central carbon and a carbon attached via a double bond.

8. **Motif 7**: - **Structure**: This motif represents a nitrile ketone structure. It is a three-carbon chain where one carbon is triple-bonded to nitrogen (nitrile group) and double-bonded to oxygen (carbonyl group).

9. **Motif 8**: - **Structure**: This structure is a simple nitrile. It consists of a three-carbon chain with a nitrogen atom triple-bonded to the central carbon.

10. **Motif 9**: - **Structure**: This motif resembles acetyl cyanide. It is a three-carbon chain with a ketone group (carbonyl group) and a cyanide group (triple-bonded nitrogen).

11. **Motif 10**: - **Structure**: This is an isoprene structure. It features a four-carbon chain with two double bonds present, making up a simple conjugated diene.

These motifs collectively illustrate the diversity of functional groups and substructures possibly present in the complex photovoltaic molecule. Each motif brings distinct electronic and structural properties to the table, contributing to the molecule's overall function in photovoltaic applications.

**Comments by Expert**

This is an easy prompt with clear instructions to follow. GPT have successfully identified and accurately described the names and functionalities of all 11 motifs, demonstrating a clear understanding of the molecular substructures and their relevance in the context of photovoltaic applications.

**Prompt**

I want you to think like a chemist performing a detailed analysis of the chemical composition of a molecule through its constituent motifs. I will highlight for you some of the distinctive fragments of a molecule. They are numbered from 0 and individually highlighted in GREEN. Focus ONLY on the substructure highlighted in green within each cell. Here is the descriptions for each substructure provided by an expert:

'''Here's the detailed analysis of each motif in the given photovoltaic molecule, broken down:

1. **Motif 0**: - **Structure**: It is a 1,3-dithiol-2-one structural motif. It features a ketone (carbonyl group) with a five-membered ring containing two sulfur atoms adjacent to the ketone group.

2. **Motif 1**: - **Structure**: This motif consists of two distinct substructures: 1. A five-membered aromatic ring resembling thiophene, composed of a sulfur atom (yellow) and four carbon atoms. 2. A simple three-carbon chain with a double bond, ending in a ketone group (carbonyl group). ... '''

I want you to tell me if any two of them should be combined together to form a more meaningful substructure. This molecule belongs to a collection of molecules used to aid the design of organic solar cells. These molecules comprise of functional groups and complex ring structures that contribute to optimized photovoltaic efficiency, utilizing factors like Pi-orbital electron delocalization or electrondonating groups/structures increasing steric hindrance. Your task is to highlight the primary functional groups of the molecule. Output a single pair of numbers if you think those two fragments should be combined, and a brief explanation why. If no such pairs exist, don't output anything.

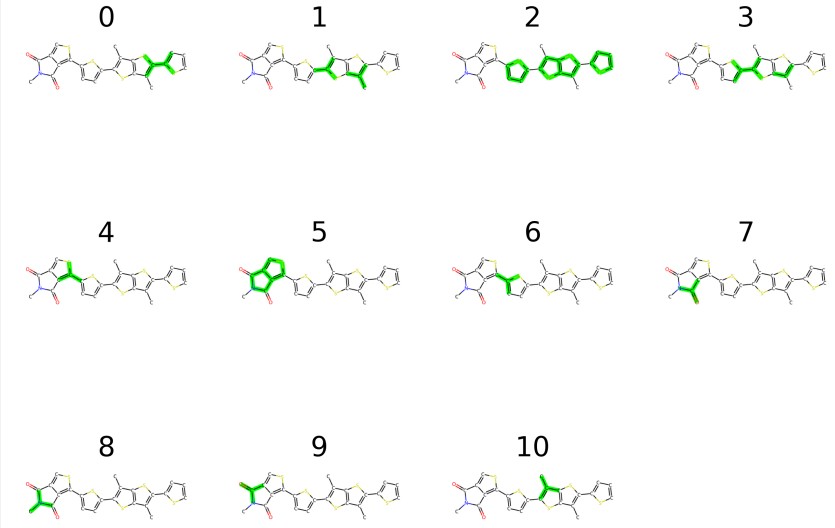

**Response**

No two fragments should be combined together to form a more meaningful substructure from the given motifs. Each motif represents distinct functional groups or structural elements that can individually contribute to the photovoltaic properties of the molecule.

**Comments by Expert**

Easy prompt, useful and specific instructions for GPT such as hint to utilize factors like electron delocalization, etc. GPT's answer: No two fragments should be combined together to form a more meaningful substructure from the given motifs. Each motif represents distinct functional groups or structural elements that can individually contribute to the photovoltaic properties of the molecule. This is correct, motif 1 provide the most distinct function to the molecule's optoelectronic properties, however, none of them form no meaningful substructure. They were injected before due to the ease of synthesis.

**Prompt**

I want you to think like a chemist performing a detailed analysis of the chemical composition of a *photovoltaic molecule* through its constituent motifs. I will highlight for you some of the distinctive substructures of a *photovoltaic molecule*. They are numbered from 0. Here are the textual descriptions of each motif:

"'Here's the detailed analysis of each motif in the given *photovoltaic molecule*, broken down:

1. **Motif 0**: - **Structure**: It is a 1,3-dithiol-2-one structural motif. It features a ketone (carbonyl group) with a five-membered ring containing two sulfur atoms adjacent to the ketone group.

2. **Motif 1**: - **Structure**: This motif consists of two distinct substructures: 1. A five-membered aromatic ring resembling thiophene, composed of a sulfur atom (yellow) and four carbon atoms. 2. A simple three-carbon chain with a double bond, ending in a ketone group (carbonyl group).

... "'

I want you to pick only ONE of these as the root motif most essential to its chemical profile. It should be the single most important motif the rest of the molecule was built around. *This molecule belongs to a collection of molecules used to aid the design of organic solar cells. These molecules comprise of functional groups and complex ring structures that contribute to optimized photovoltaic efficiency, utilizing thiophene-based groups for enhanced Pi-orbital electron delocalization or electrondonating groups/structures increasing steric hindrance. Your selected root motif MUST contain the thiophene group.* If there are multiple such motifs, or one doesn't clearly stand out, just pick one of them. Give your answer as a single number. Explain your reasoning carefully.

**Response**

The root motif most essential to the chemical profile of the photovoltaic molecule, which contains the thiophene group and contributes significantly to the molecule's function in photovoltaic applications, is **Motif 1**.

**Reasoning:** - **Thiophene Presence**: Motif 1 contains a distinct thiophene unit, which is critical for enhanced $\pi$-orbital electron delocalization. Thiophenes are well-known for their role in organic electronics and photovoltaics due to their ability to facilitate charge transport through $\pi$-conjugation. - **Substructure Complexity**: Motif 1 consists of both a thiophene ring and a three-carbon chain with a ketone group. This complexity aids in creating regions of electron-rich and electron-deficient areas within the molecule, which is beneficial for the formation of charge-separated states and efficient photovoltaic processes. - **Integration into Larger Structure**: Motif 1 provides a suitable backbone around which additional functional groups and motifs can be attached, enhancing the overall photovoltaic performance by balancing electronic and structural properties.

Therefore, based on the importance of the thiophene group and the structural features that support photovoltaic efficiency, Motif 1 is chosen as the root motif.

**Comments by Expert**

correct, GPT answered motif 1, which highlights a distinct thiolphene unit and its significancy in photovoltaic applications. Thiophene is a core structural component commonly found in conducting polymers and widely utilized in photovoltaic applications due to its role in facilitating $\pi$-orbital electron delocalization and charge transport.

**Prompt**

I want you to perform a simple post-processing step of the following response: "'Here's the detailed analysis of each motif in the given photovoltaic molecule, broken down:

1. **Motif 0**: - **Structure**: It is a 1,3-dithiol-2-one structural motif. It features a ketone (carbonyl group) with a five-membered ring containing two sulfur atoms adjacent to the ketone group.

2. **Motif 1**: - **Structure**: This motif consists of two distinct substructures: 1. A five-membered aromatic ring resembling thiophene, composed of a sulfur atom (yellow) and four carbon atoms. 2. A simple three-carbon chain with a double bond, ending in a ketone group (carbonyl group).

..."' The input is a response from another language agent. It describes motifs numbered from Motif 0 to Motif "'10'", inclusive! I want you to rephrase each motif description by filling in X within the following sentence template: This motif is X Be sure to condense the description and output a single PHRASE such that the sentence template is grammatically correct. Don't capitalize the first letter, since your answer should just be a phrase. Output your rephrasing for each motif on a SEPARATE line, using only a new line for delimiting different motifs. Don't output anything else. MAKE SURE you do it for EVERY MOTIF!

### Response

1. a 1,3-dithiol-2-one structural motif with a ketone and a five-membered ring containing two sulfur atoms
2. two distinct substructures: a thiophene-like five-membered aromatic ring and a three-carbon chain ending in a ketone group
3. three different heterocyclic aromatic rings: thiophene, furan, and pyrrole
4. similar to Motif 0 but with an additional carbonyl group branching off the central carbon
5. a stylized thioacetone structure with a three-carbon chain and a sulfur atom connected to the central carbon
6. an indole-like structure with a benzene ring fused to a five-membered ring containing nitrogen and sulfur
7. a thioketo structure with a three-carbon chain and a sulfur atom double-bonded to the central carbon
8. a nitrile ketone structure with a three-carbon chain, a nitrile group, and a carbonyl group
9. a simple nitrile with a three-carbon chain and a nitrogen atom triple-bonded to the central carbon
10. an acetyl cyanide-like structure with a three-carbon chain, a ketone group, and a cyanide group
11. an isoprene structure with a four-carbon chain and two double bonds making up a conjugated diene

### Comments by Expert

The answer from GPT is correct, it provides concise explanations for each motif. This indicates a clear understanding of the structural and functional elements of the photovoltaic molecule.

### Prompt

I want you to think like a chemist performing a detailed analysis of the chemical composition of a *photovoltaic molecule* through its constituent motifs. This requires two steps: 1) analyzing the individual motifs and 2) analyzing the pairwise interactions of motifs. The first step is already done. The second step is where I need your help. I will highlight for you different motif interactions within the same molecule. These interactions are numbered one-by-one, beginning with 0. Here are the textual descriptions of each motif interaction pair.

Interaction 0 features "'1. a 1,3-dithiol-2-one structural motif with a ketone and a five-membered ring containing two sulfur atoms'" and "'2. two distinct substructures: a thiophene-like five-membered aromatic ring and a three-carbon chain ending in a ketone group.'" Interaction 1 features "'2. two distinct substructures: a thiophene-like five-membered aromatic ring and a three-carbon chain ending in a ketone group'" and "'3. three different heterocyclic aromatic rings: thiophene, furan, and pyrrole.'" Interaction 2 features "'3. three different heterocyclic aromatic rings: thiophene, furan, and pyrrole and 1. a 1,3-dithiol-2-one structural motif with a ketone'" and a "'five-membered ring containing two sulfur atoms'".

I want you to tell me which one of these is MOST important, and which one of these is LEAST important. Output one number identifying the MOST important, and give a brief explanation. Output one number identifying the LEAST important, and a brief explanation why.

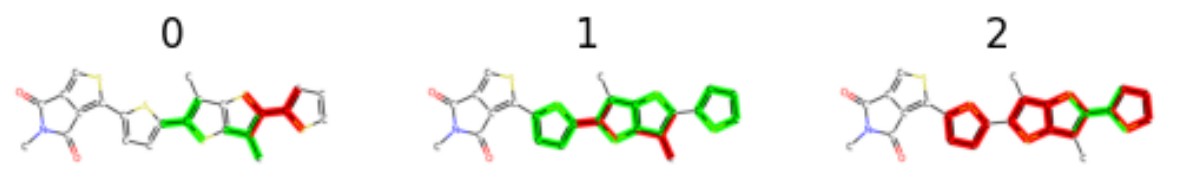

### Response

Given the importance of the motifs and their interactions in photovoltaic molecules, let's consider the roles played by each interaction.

**Most Important Interaction (Number 1): 2** **Reasoning:** Interaction 2 features three heterocyclic aromatic rings: thiophene, furan, and pyrrole, interacting with a 1,3-dithiol-2-one structural motif with a ketone and a five-membered ring containing two sulfur atoms. This interaction involves a larger number of functional groups with a variety of heteroatoms, contributing significantly to the electronic properties and potential charge transfer efficiency. The presence of multiple aromatic rings could enhance electron delocalization, improving the photovoltaic properties of the molecule.

**Least Important Interaction (Number 0): 1** **Reasoning:** Interaction 1 involves a 1,3-dithiol-2-one structural motif interacting with two substructures: a thiophene-like five-membered aromatic ring and a three-carbon chain

ending in a ketone group. While the 1,3-dithiol-2-one motif is crucial, the three-carbon chain ending in a ketone group is relatively straightforward and less complex compared to the multiple heterocyclic aromatic systems presented in interactions 1 and 2. The lack of additional aromatic heterocycles in this interaction makes it less critical in the context of electronic interactions within the molecule.

Thus, the most important interaction is **2**, and the least important interaction is **1**.

**Comments by Expert**

Correct, interaction 2 is the most important because it features three heterocyclic aromatic rings—thiophene, furan, and pyrrole—interacting with a 1,3-dithiol-2-one structural motif containing a ketone and a sulfur-rich five-membered ring. This combination significantly contributes to electronic delocalization and charge transfer efficiency, which are critical for photovoltaic functionality. Interaction 1 is the least important due to its simpler structure, involving a three-carbon chain ending in a ketone group and lacking the additional aromatic cycles. This simplicity results in a lower contribution to the electronic interactions.

### H.2.1. EXAMPLE DESIGN STORY

We provide an example design story for CN1C(=O)c2c3c(c(C(F)(F)F)c(-c4c5C(=O)[Si](C)(C)C(=O)c5cn4C)c4C(=O)N(C)C(O)c(c43)c(C(F)(F)F)c2)C1=O. The story begins by rationalizing the choice of a root motif, in the context of designing organic photovoltaics, then proceeds to explain each positive-negative connection in the parse tree from the perspective of structure-property relationships and design. Here is the story:

1. For the monomer described, intended for organic photovoltaic applications, the most essential motif would be the one that best supports charge transport and absorption of light. In organic photovoltaics, aromatic and conjugated systems are crucial because they enable extended $\pi$-conjugation, which improves the absorption of solar radiation and facilitates the transport of charge carriers.

Among the motifs provided, the **Pyrimidine ring (number 2)** is a strong candidate due to its aromatic nature and the presence of nitrogen atoms. The nitrogen atoms in a pyrimidine ring can participate in hydrogen bonding and other electroactive interactions, making this structure very important for the electronic properties of the polymer.

### Therefore, the root motif is **number 2**.

#### Reasoning:

- **Conjugation and Aromaticity**: The pyrimidine ring is aromatic, providing a structure that allows for delocalization of $\pi$-electrons, which is beneficial for light absorption and charge transport. - **Nitrogen Atoms**: The nitrogen atoms can engage in various intermolecular interactions which are important for altering the electronic properties and improving the efficiency in photovoltaic applications. - **Versatility**: Pyrimidine rings are known to be versatile components in organic electronics, capable of facilitating charge carrier mobility and stability.

Thus, the Pyrimidine ring stands out as the single most important motif essential to the chemical profile of the monomer for its usefulness in organic photovoltaic applications. 2. The positive (red) and negative (green) motifs in this polymer design play a crucial role in enhancing its electronic properties, making it a viable candidate for applications such as organic solar cells. The red motif, likely a conjugated system with extended pi-electron delocalization, serves as the electron-donating or transporting component. The green motif, an electron-withdrawing group, is strategically positioned to create a strong dipole moment and facilitate charge separation. This donor-acceptor interplay is justified as it maximizes the polymer's ability to harness photogenerated excitons and promotes efficient charge transfer. The integration of the negative motif within the red motif structure lowers the overall bandgap, which is beneficial for the absorption of a broader spectrum of sunlight, thus increasing the efficiency of solar energy conversion. Furthermore, the inherent stability and solubility imparted by this design ensure that the polymer can be processed in various thin-film fabrication techniques, making it highly adaptable for use in organic photovoltaics. 3. In the design of polymer materials for applications such as organic solar cells, the positive (red) and negative (green) motifs play crucial roles in optimizing the photophysical and electronic properties of the polymer. The red motif, an aromatic structure with electron-donating groups, enhances the polymer's ability to absorb sunlight and facilitate exciton generation. The green motif, an adjacent electron-withdrawing unit, increases the polymer's electron affinity, aiding in the efficient separation and transport of charge carriers upon photoexcitation. This juxtaposition of donor and acceptor segments within the same molecular framework creates a built-in push-pull system, improving charge separation efficiency and minimizing recombination losses. Such finely tuned electronic interactions are essential for achieving high power

conversion efficiencies in organic solar cells, as they directly impact the material's photovoltaic performance. Thus, the careful integration of these motifs is justified as it provides a strategic means to balance and enhance the photovoltaic and charge transport properties needed for high-performance organic solar materials. 4. The incorporation of the negative (green) motif into the positive (red) motif within this polymer design is pivotal for enhancing its functionality in specific applications like organic solar cells. The positive motif, marked in red, likely represents an electron-rich region, while the negative motif, highlighted in green, signifies an electron-deficient segment. This strategic juxtaposition creates a donor-acceptor (D-A) interaction, which is essential for effective charge separation and transport. This configuration helps facilitate the alternating push-pull effect of electrons, which is crucial in improving the photophysical properties of the polymer, such as light absorption and charge carrier mobility. In organic solar cells, such D-A conjugated systems enhance the absorption spectrum, increase the efficiency of exciton dissociation, and promote the transport of electrons and holes, thereby improving the overall power conversion efficiency. Therefore, the careful design and synthesis of these alternating motifs are justified as they provide a molecular architecture conducive to high-performance organic photovoltaic materials. 5. The positive (red) and negative (green) motifs in the polymer depicted play a crucial role in determining its electronic properties and overall efficacy in applications like organic solar cells. The positive motif likely represents an electron-rich segment, while the negative motif represents an electron-deficient segment. This complementary interaction promotes intramolecular charge transfer, which is essential for enhancing the polymer's ability to absorb light and generate electron-hole pairs efficiently. The strategic incorporation of the green motif within the red framework ensures a balanced electronic structure, thereby optimizing the separation of charge carriers and minimizing recombination losses. This facilitates effective charge transport, leading to improved photovoltaic performance. Furthermore, this donor-acceptor architecture can be tuned to adjust the absorption spectrum, enabling better alignment with the solar spectrum. Hence, the addition of the negative (green) motif to the positive (red) motif is a deliberate design choice to enhance the polymer's semiconducting properties, making it highly effective for applications in organic solar cells. 6. In the design of polymers for organic solar cells, the strategic incorporation of the negative (green) motif into the positive (red) motif is crucial for optimizing the electronic properties and enhancing performance. The positive motif, depicted in red, likely represents an electron-donating unit, while the negative motif, shown in green, is indicative of an electron-withdrawing unit. When these two motifs are combined within the polymer backbone, they create an alternating donor-acceptor structure that facilitates charge separation and transport by lowering the bandgap and increasing the polymer's ability to absorb sunlight. This push-pull interaction between the electron-rich and electron-deficient segments contributes to the creation of effective pathways for exciton dissociation and charge carrier mobility. Consequently, this interaction plays a fundamental role in balancing and optimizing the absorption spectrum, charge transport, and overall efficiency of the organic solar cells, making the addition of the green motif justified and beneficial for enhancing the performance of such devices. 7. In the design of polymers for organic solar cells, the interaction between the positive (red) and negative (green) motifs is of paramount importance for optimizing the electronic properties of the material. The positive motif, marked in red, likely represents an electron-donating unit, while the negative motif, marked in green, represents an electron-accepting unit. This push-pull mechanism, also referred to as a donor-acceptor interaction, is crucial for improving charge separation and transport within the polymer. By incorporating both electron-donating and electron-withdrawing motifs, the resulting polymer exhibits a narrowed bandgap, enhancing its ability to absorb a broader spectrum of sunlight. The dipole moments generated at the interface between these motifs also facilitate exciton dissociation, improving the efficiency of charge generation. Additionally, the conjugated backbone formed by the alternating positive and negative motifs creates a pathway for electron mobility, which is essential for the polymer's performance as an active layer in organic solar cells. Thus, the deliberate design incorporating these specific motifs is justified by its direct contribution to improved light absorption, charge separation, and overall photovoltaic efficiency. 8. The design of the polymer shown, which includes alternating positive (red) and negative (green) motifs, is likely aimed at enhancing the material's electronic properties, useful for applications such as organic solar cells. The positive motif suggests regions with electron-withdrawing characteristics while the negative motif indicates electron-donating properties. This juxtaposition creates a donor-acceptor interaction framework that is crucial for tuning the polymer's bandgap. In organic solar cells, such a tailored bandgap is vital for effective light absorption and charge separation. The electron-rich (negative) motif can facilitate enhanced electron mobility while the electron-deficient (positive) motif can improve hole transport efficiency. This duality ensures the polymer can effectively separate and transport charge carriers generated upon light absorption, thereby improving the overall efficiency of the solar cells. The inclusion of side chains and functional groups further modulates solubility, processability, and morphological stability, making the polymer system adaptable for practical device fabrication. 9. In the context of designing polymers for applications such as organic solar cells, the integration of a negative (green) motif with a positive (red) motif is crucial for optimizing the electronic properties and ensuring effective performance. The red motif in the polymer structure typically represents electron-donating units, which enhance the electron density and facilitate charge transport. The green motif, on the other hand, represents electron-withdrawing units, which help to lower the polymer's

overall energy levels and improve electron acceptance. This positive-negative interaction is fundamental because it results in a defined energy band gap that is essential for efficient absorption of sunlight and conversion into electrical energy. Additionally, the interplay between these motifs enhances the polymer's ability to form a well-ordered, crystalline structure, further improving charge mobility and overall efficiency. Their balanced interaction improves the stability, solubility, and mechanical properties of the polymer, making it suitable for real-world applications in organic electronics and photovoltaics. 10. In designing a polymer for applications like organic solar cells, the interaction between the positive (red) and negative (green) motifs plays a critical role in modulating the polymer's electronic properties. The positive red motif, in this case, forms the backbone of the polymer's structure, contributing to the overall electronic conduction. By introducing the negative green motif, which likely contains electron-withdrawing groups such as fluorine atoms, the electron density distribution within the polymer is altered, enhancing its ability to facilitate charge transfer processes. This complementary interaction can enhance the polymer's capacity for efficient charge separation and transport, crucial for the effective conversion of light to electricity in organic solar cells. Furthermore, the specific electron-donating and electron-accepting properties of these motifs can be fine-tuned to optimize the polymer's absorption spectrum and energy levels, making the material more effective in capturing sunlight and transforming it into usable energy. Thus, the integration of these motifs is justified through a detailed understanding of their synergistic effects on the polymer's photovoltaic performance. 11. The polymer depicted showcases a strategic combination of a positive red motif and a negative green motif, designed to enhance its properties for applications like organic solar cells. The red motif, rich in electronegative carbonyl (C=O) and fluorine (F) groups, likely contributes to the electron-accepting characteristics necessary for efficient charge separation and transfer, critical in photovoltaic functions. Meanwhile, the green motif, embedded centrally, presumably acts as an electron-donating unit due to its conjugated ring structure with additional electron-withdrawing carbonyl groups enhancing the polymer's ability to create a stable, low-energy LUMO level. The interplay between these motifs through $\pi$-conjugation and potential intramolecular charge transfer enhances the polymer's electronic properties, such as its photochemical stability, light absorption, and charge mobility. This synergistic interaction justifies the inclusion of the negative motif to optimize the polymer's efficiency in harvesting solar energy, delivering performance enhancements critical for next-generation organic solar cells. 12. In evaluating the design of the polymer for potential applications such as organic solar cells, the interaction between the positive (red) and negative (green) motifs is crucial. The strategic incorporation of the negative motif, marked by its electron-withdrawing groups (such as the carbonyl groups), is critical in creating an internal charge transfer within the polymer matrix. This enhances electron mobility by facilitating a donor-acceptor interaction where the red motif acts as the electron donor and the green motif functions as the electron acceptor. Such a configuration supports efficient separation and transport of charge carriers. Consequently, this charge transfer interaction reduces recombination losses and increases the overall efficiency of the polymer in solar energy conversion. Additionally, the spatial orientation and electronic properties of the combined motifs influence the polymer's absorption spectrum and photophysical properties, optimizing light absorption. Thus, the deliberate addition of the negative motif to the polymer structure is justified by its substantial impact on enhancing electrical conductivity, charge separation, and optical properties, which are vital for the efficacy and performance of organic solar cells. 13. In the context of designing a polymer for applications such as organic solar cells, the incorporation of positive and negative motifs plays a pivotal role in tailoring the material's electronic properties. The positive (red) motif may be indicative of an electron-donating moiety, which can enhance the charge carrier mobility within the polymer. Conversely, the negative (green) motif likely represents an electron-withdrawing group, crucial for stabilizing the electron density and improving the polymer's electron affinity. The interaction between these motifs creates a push-pull effect in the polymer backbone, which is vital for optimizing the absorption of a broad spectrum of light and facilitating efficient charge separation and transport. This push-pull mechanism is particularly advantageous in organic solar cells, as it contributes to higher power conversion efficiency by maximizing the generation and transport of free charge carriers when the polymer is exposed to sunlight. The strategic placement and balance of these motifs enable fine-tuning of the polymer's HOMO-LUMO (highest occupied molecular orbital-lowest unoccupied molecular orbital) gap, ensuring it is suitable for effective photovoltaic performance. 14. In the given polymer, the design incorporates a positive (red) and a negative (green) motif, which plays a critical role in its chemical and physical properties. The positive motif, marked in red, is likely an electron-donor segment that provides a source of electrons through conjugated systems or electron-rich groups. The negative motif, highlighted in green, could incorporate electron-withdrawing groups that facilitate electron acceptance, making it an electron-acceptor segment. This complementary interaction between donor and acceptor segments within the polymer is fundamental for optimizing the charge-transfer processes, which are essential in applications such as organic solar cells. When light excites the polymer, the positive (red) motif can donate electrons that are efficiently transferred to the negative (green) motif. This charge separation is crucial for generating electrical current in organic photovoltaics. Additionally, this positive-negative interaction can enhance the polymer's stability, morphology, and overall electronic properties, making it a viable candidate for high-performance organic solar cells. 15. In designing polymers for applications like organic solar cells, the interaction

between the positive (red) and negative (green) motifs plays a critical role in optimizing the polymer's electronic properties and overall performance. The red motif, likely possessing electron-donating characteristics, enhances the polymer's ability to transport holes, making it an efficient donor material. On the other hand, the green motif, characterized by electron-withdrawing properties, contributes to electron transport and serves as an acceptor material. The juxtaposition of these contrasting electronic features within the same polymer backbone facilitates effective charge separation and transport, crucial for the efficiency of organic photovoltaic devices. Moreover, the fine-tuning of these donor-acceptor interactions influences the polymer's bandgap and energy levels, which can be tailored to maximize light absorption in the solar spectrum, thus enhancing the photocurrent generation in solar cell applications. This delicate balance and interaction between the positive and negative motifs thereby justify their integration, significantly contributing to the polymer's optoelectronic properties and making it a viable candidate for high-performance organic solar cells. 16. In designing polymers for applications such as organic solar cells, the interplay between electron-donating (red) and electron-withdrawing (green) motifs is pivotal for optimizing the electronic properties and stability of the material. The red motif serves as an electron-donating unit, facilitating the creation of a high-energy orbital system essential for effective light absorption and exciton generation. Conversely, the green motif acts as an electron-accepting unit, which helps in stabilizing the generated excitons and improving charge separation efficiency. This complementary interaction between the electron-rich and electron-deficient segments creates a balanced distribution of electronic density, thereby fine-tuning the energy levels and improving charge transport abilities. Additionally, this donor-acceptor synergy enhances the structural rigidity and thermal stability of the polymer, making it more robust under operational conditions. The judicious incorporation of the negative (green) motif to the positive (red) motif is thus essential in engineering polymers with the desirable electronic and physical characteristics suited for high-performance organic solar cells. 17. In the context of designing a polymer for applications such as organic solar cells, the interaction between the positive (red) and negative (green) motifs is crucial for tuning the polymer's optoelectronic properties. The negative motif in green, characterized by electron-withdrawing groups, enhances the polymer's electron affinity, improving its ability to accept electrons. This leads to a lower energy band gap, which is favorable for absorbing a broader spectrum of sunlight. Additionally, the complementary positioning of these motifs enhances charge separation and transport efficiencies within the polymer matrix. The electron-rich positive motif in red can act as a donor, facilitating effective charge transfer processes when paired with the electron-deficient negative motif. By strategically incorporating these motifs into the polymer framework, we can optimize the material's photovoltaic performance, achieving better light absorption, higher charge carrier mobility, and ultimately improved efficiency in converting solar energy to electrical energy in organic solar cells. 18. In the context of designing polymers for applications such as organic solar cells, the positive (red) and negative (green) motifs are strategically integrated to optimize electronic and structural properties. The red motif, being a conjugated aromatic structure, offers high electron density and good charge transport characteristics due to its delocalized $\pi$-electrons, which is crucial in facilitating efficient charge mobility. Conversely, the green motif, characterized by its electron-withdrawing functional groups (e.g., carbonyl and fluorine atoms), introduces electron deficiency into the polymer chain. This electron-withdrawing nature helps to lower the polymer's LUMO (Lowest Unoccupied Molecular Orbital) energy level, aiding in the enhancement of electron acceptor properties. The presence of both motifs creates a donor-acceptor (D-A) interaction within the polymer, optimizing the solar cell's intrinsic properties such as bandgap tuning, absorption spectrum, and charge separation efficiency. This deliberate juxtaposition of electron-rich and electron-deficient motifs forms a polymer network that is highly suitable for converting sunlight into electrical energy with maximized efficiency, making this motif combination crucial for advanced organic photovoltaic applications. 19. In the design of a polymer for applications such as organic solar cells, the incorporation of complementary electronic motifs is crucial for optimizing charge transfer and enhancing device efficiency. The positive motif highlighted in red and the negative motif in green represent electron-donating and electron-withdrawing segments, respectively. This donor-acceptor architecture facilitates effective intermolecular charge transfer dynamics, which is essential for efficient exciton dissociation and charge transport within the polymer matrix. The electron-donating capacity of the red motif, typically featuring conjugated systems and groups that can delocalize electrons, complements the electron-deficiency of the green motif, often imbued with electronegative groups or atoms like fluorine and carbonyl groups. This interaction not only promotes optimal energy level alignment between the highest occupied molecular orbital (HOMO) and the lowest unoccupied molecular orbital (LUMO) but also enhances the polymer's photophysical properties by broadening its absorption spectrum. Such a synergistic design is beneficial for increasing the power conversion efficiency of organic solar cells by maximizing light absorption and facilitating efficient charge separation and mobility. 20. In the context of designing polymers for applications such as organic solar cells, the interplay between electron-rich (positive, red) and electron-deficient (negative, green) motifs is of paramount importance. The red motif, characterized by its extended $\pi$-conjugation, acts as an electron-donating unit, facilitating efficient charge transport. Meanwhile, the green motif introduces electron-withdrawing functionalities, thereby reducing the polymer's highest occupied molecular orbital (HOMO) energy levels while increasing its lowest unoccupied molecular orbital (LUMO)

energy levels. This complementary pairing creates a unique donor-acceptor interface within the polymer structure, enhancing charge separation and thereby improving the photovoltaic performance. Furthermore, the electron-withdrawing groups can stabilize the resulting negative charges, reducing charge recombination rates. This careful juxtaposition of motifs thereby optimizes light absorption, enhances charge carrier mobility, and ultimately leads to improved energy conversion efficiencies in organic solar cell applications. 21. In designing polymers for applications such as organic solar cells, the strategic incorporation of both positive (red) and negative (green) motifs is essential to optimize the material's properties. The red motif, a polyaromatic segment, serves as an electron donor, while the green motif, with silicon and carbonyl groups, acts as an electron acceptor due to its electron-withdrawing nature. This donor-acceptor interaction enhances charge separation and charge carrier mobility within the polymer, crucial for efficient photovoltaic performance. The juxtaposition of these motifs can lead to a reduction in the polymer's band gap, increased light absorption, and improved exciton dissociation, thereby enhancing the efficiency of organic solar cells. Additionally, the specific arrangement of the motifs affects the crystallinity and morphological stability of the polymer, which are pivotal for device performance and longevity. This synergistic design illustrates how the confluence of electron-rich and electron-deficient units can be exploited to tailor the electronic and physical properties of polymers, rendering them suitable for cutting-edge applications in organic electronics.

### H.3. Case Study: Isocyanates

Isocyanates are highly reactive organic compounds characterized by the presence of one or more isocyanate groups (-N=C=O). Their high reactivity makes them essential in the production of polyurethanes, this reaction forms urethane linkages, which are the backbone of polyurethane materials. Polyurethanes that are synthesized from isocyanates are really versatile, with a wide range of applications from flexible foams and elastomers to rigid insulation materials. The molecular structure of the isocyanate can be tailored, with aliphatic or aromatic variations influencing the final properties of the polymer, such as its mechanical strength, flexibility, and resistance to environmental factors like UV radiation. However, due to their high reactivity, isocyanates are associated with potential health hazards, including irritation and sensitization. The safety concerns attached to this compounds made it requires strict industrial safety protocols during handling and processing.

For isocyanate, features -N=C=O and highly reactive, especially in forming polyurethane linkages. The selected molecule contains two distinct -N=C=O groups serving as core functional groups to form isocyanate and highly represent the desired reactivity in this dataset of the molecules.

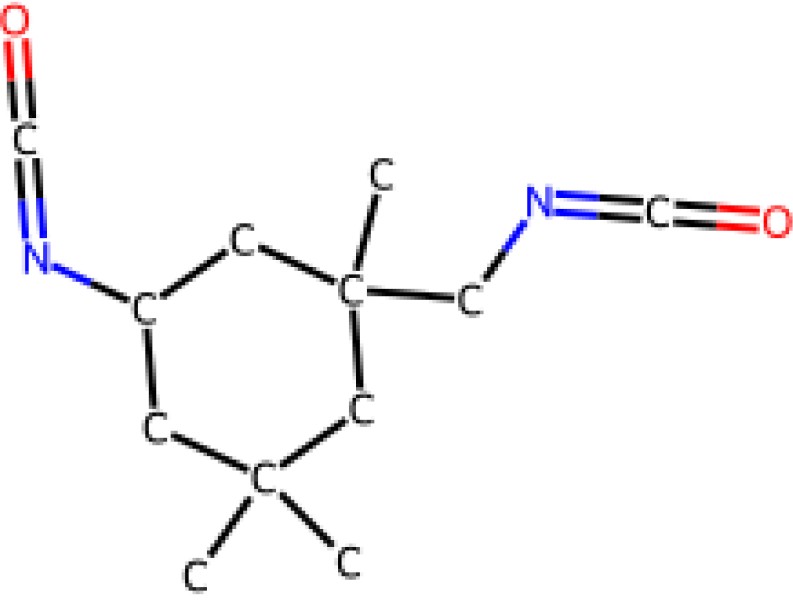

*Figure 10.* Input molecule from Isocyanates

## Prompt

I want you to think like a chemist performing a detailed analysis of the chemical composition of an isocyanates through its constituent motifs. I will highlight for you 9 of the substructures of a molecule. They are numbered one-by-one from Motif 0 to Motif 8, inclusive. I want you to explain, concisely, what each numbered motif is. Make sure to start from Motif 0 and go in order of the numbering. MAKE SURE you describe EVERY MOTIF!

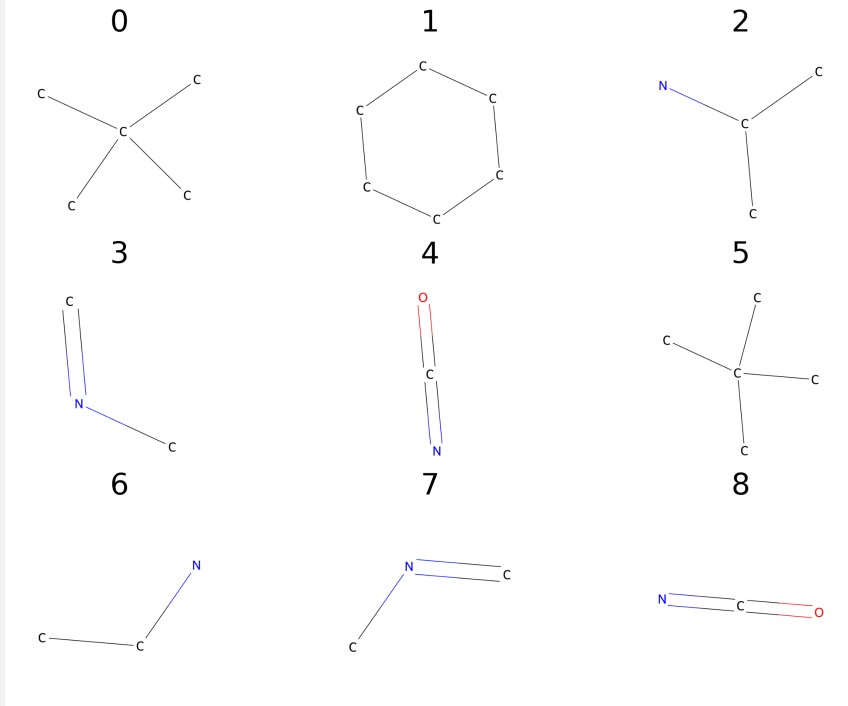

## Response

Sure, let's analyze each motif in order:

%%% Motif 0 This structure represents a tert-butyl group. It consists of a central carbon atom bonded to three methyl groups (each a CH3 group).

%%% Motif 1 This structure is a cyclohexane ring. It consists of six carbon atoms arranged in a hexagonal ring, with each carbon atom bonded to two other carbons and two hydrogen atoms (not shown).

%%% Motif 2 This structure is an ethylamine group. It consists of an amine group (NH2) attached to an ethyl group (CH3-CH2-).

%%% Motif 3 This structure is an ethylideneamine motif. It consists of a carbon-nitrogen (C=N) double bond, with the nitrogen also bonded to an alkyl group (CH3) and the carbon to another carbon.

%%% Motif 4 This structure features an O=C=N group, indicating the presence of an isocyanate group bonded to a carbonyl group. This is a key motif in many urea and carbamate derivatives.

%%% Motif 5 This structure represents a neopentyl group. It consists of a central carbon atom bonded to four other carbon atoms: one in a methyl group (CH3) and three in methylene groups (CH2).

%%% Motif 6 This structure shows a nitrile group (C≡N) attached to an ethyl group (CH3-CH2-). Nitriles are commonly found in organic compounds.

%%% Motif 7 This structure is a cyanoacetamide group. It consists of a nitrile group (C≡N) adjacent to a carbonyl group (C=O).

%%% Motif 8 This structure is an isocyanate group (N=C=O). It consists of a nitrogen atom double-bonded to a carbon atom, which is also double-bonded to an oxygen atom.

These motifs may represent fragments of larger molecules, often seen in various industrial and chemical synthesis processes.

This is an easy and straightforward prompt, with clear guidelines and restrictions that ensure each motif is individually analyzed and described. GPT's answer: response accurately identified and clearly explained the functionality of all motifs from 0 to 8, providing precise and concise descriptions of each substructure. The answer effectively captures the unique chemical features of the motifs, demonstrating a comprehensive understanding of the molecular composition.

**Prompt**

I want you to think like a chemist performing a detailed analysis of the chemical composition of an isocyanates through its constituent motifs. I will highlight for you some of the distinctive fragments of a molecule. They are numbered from 0 and individually highlighted in RED. Focus ONLY on the substructure highlighted in red within each cell. Here is the descriptions for each substructure provided by an expert:

Sure, let's analyze each motif in order:

%%% Motif 0 This structure represents a tert-butyl group. It consists of a central carbon atom bonded to three methyl groups (each a CH3 group).

%%% Motif 1 This structure is a cyclohexane ring. It consists of six carbon atoms arranged in a hexagonal ring, with each carbon atom bonded to two other carbons and two hydrogen atoms (not shown).

%%% Motif 2 This structure is an ethylamine group. It consists of an amine group (NH2) attached to an ethyl group (CH3-CH2-).

%%% Motif 3 This structure is an ethylideneamine motif. It consists of a carbon-nitrogen (C=N) double bond, with the nitrogen also bonded to an alkyl group (CH3) and the carbon to another carbon.

%%% Motif 4 This structure features an O=C=N group, indicating the presence of an isocyanate group bonded to a carbonyl group. This is a key motif in many urea and carbamate derivatives.

%%% Motif 5 This structure represents a neopentyl group. It consists of a central carbon atom bonded to four other carbon atoms: one in a methyl group (CH3) and three in methylene groups (CH2).

%%% Motif 6 This structure shows a nitrile group (C≡N) attached to an ethyl group (CH3-CH2-). Nitriles are commonly found in organic compounds.

%%% Motif 7 This structure is a cyanoacetamide group. It consists of a nitrile group (C≡N) adjacent to a carbonyl group (C=O).

%%% Motif 8 This structure is an isocyanate group (N=C=O). It consists of a nitrogen atom double-bonded to a carbon atom, which is also double-bonded to an oxygen atom.

These motifs may represent fragments of larger molecules, often seen in various industrial and chemical synthesis processes.

I want you to tell me if any two of them should be combined together to form a more meaningful substructure. Remember the defining isocyanate group is -N=C=O. Your task is to highlight the primary functional groups of the molecule. Output a single pair of numbers if you think those two fragments should be combined, and a brief explanation why. If no such pairs exist, don't output anything.

**Response**

%%% Motif 4 This structure features an O=C=N group, representing the presence of an isocyanate group bonded to a carbonyl group, which is a critical motif in many urea and carbamate derivatives.

%%% Motif 8 This structure features an isocyanate group (N=C=O), consisting of a nitrogen atom double-bonded to a carbon atom, which is also double-bonded to an oxygen atom.

Combining **Motif 4** and **Motif 8** forms the primary functional group: a carbonyl group directly attached to an isocyanate group, which is critical for identifying the chemical properties and reactivity of isocyanates.

%%% **Conclusion:** **4, 8** may be combined into a functional group consisting of a carbonyl bonded directly to an isocyanate, making the structure more meaningful in the context of isocyanate chemistry.

**Comments by Expert**

This prompt is easy and straightforward, it simply asks GPT to identify if any two groups should be combined to form a more meaningful isocyanate structure. GPT's answer is 4 and 8. GPT correctly recognized Motifs 4 and 8 as isocyanate-related groups but incorrectly suggested that they should be combined, as both motifs already represent

functional isocyanate groups individually. This demonstrates a slight misunderstanding of the requirement to propose combinations only when the resulting structure provides new or enhanced functionality, which is unnecessary in this case since the motifs are already complete and meaningful in their current forms.

## Prompt

I want you to think like a chemist performing a detailed analysis of the chemical composition of an isocyanates through its constituent motifs. I will highlight for you some of the distinctive substructures of an isocyanates. They are numbered from 0. Here are the textual descriptions of each motif:

Sure, let's analyze each motif in order:

%%% Motif 0 This structure represents a tert-butyl group. It consists of a central carbon atom bonded to three methyl groups (each a CH3 group).

%%% Motif 1 This structure is a cyclohexane ring. It consists of six carbon atoms arranged in a hexagonal ring, with each carbon atom bonded to two other carbons and two hydrogen atoms (not shown).

%%% Motif 2 This structure is an ethylamine group. It consists of an amine group (NH2) attached to an ethyl group (CH3-CH2-).

%%% Motif 3 This structure is an ethylideneamine motif. It consists of a carbon-nitrogen (C=N) double bond, with the nitrogen also bonded to an alkyl group (CH3) and the carbon to another carbon.

%%% Motif 4 This structure features an O=C=N group, indicating the presence of an isocyanate group bonded to a carbonyl group. This is a key motif in many urea and carbamate derivatives.

%%% Motif 5 This structure represents a neopentyl group. It consists of a central carbon atom bonded to four other carbon atoms: one in a methyl group (CH3) and three in methylene groups (CH2).

%%% Motif 6 This structure shows a nitrile group (C≡N) attached to an ethyl group (CH3-CH2-). Nitriles are commonly found in organic compounds.

%%% Motif 7 This structure is a cyanoacetamide group. It consists of a nitrile group (C≡N) adjacent to a carbonyl group (C=O).

%%% Motif 8 This structure is an isocyanate group (N=C=O). It consists of a nitrogen atom double-bonded to a carbon atom, which is also double-bonded to an oxygen atom.

These motifs may represent fragments of larger molecules, often seen in various industrial and chemical synthesis processes.

I want you to pick only ONE of these as the root motif most essential to its chemical profile. It should be the single most important motif the rest of the isocyanate was built around. The defining isocyanate group is -N=C=O, so your selected root motif MUST contain the group. If there are multiple such motifs, or one doesn't clearly stand out, just pick one of them. Give your answer as a single number. Explain your reasoning carefully.

## Response

%%% Answer: 8
%%% Reasoning:
In the given motifs, we are looking for the motif that contains the isocyanate group (-N=C=O) at its core, as this functional group is essential to the chemical profile of isocyanates.

Among the given motifs, Motif 8 includes the isocyanate group distinctly:

- Motif 8: It features the isocyanate group (N=C=O) prominently and without any additional structural complexity directly attached to it. This makes it the simplest and most straightforward representation of the essential isocyanate entity.

This feature, the critical and defining part of an isocyanate, means Motif 8 is the most crucial to the chemical profile of the substance. The rest of the molecule is built around this group, making it the root structure.

Other motifs either have additional complex groups attached to the isocyanate group or other functional groups that could potentially classify the molecules differently. Hence, Motif 8 is chosen as it aligns precisely with the core chemical characteristic we are focusing on.

## Comments by Expert

The prompt is of medium level complexity, since picking the most important functional group involves empirical experience in organic chemistry, and the answer required one individual selection can bring discrepancies sometimes between language models and chemistry expert. GPT's answer: correct, GPT answer motif 8 since it features the

distinct isocyanate group, which is essential to the chemical profile of isocyanates. The explanation provided is thorough, highlighting that Motif 8 represents the simplest and most direct form of the isocyanate group without additional structural complexity. GPT even correctly noted that other motifs include either additional groups attached to the isocyanate or other functional groups that could alter the molecule's classification, further justifying the choice of Motif 8. This demonstrates a clear understanding of the core chemical characteristics required for the task and therefore delivers a perfect answer to the question.

### H.4. Case Study: Acrylates

Acrylates refer to a broad class of chemical compounds that are derivatives of acrylic acid and its related esters. They can undergo polymerization rapidly through a free- radical mechanism, making them particularly useful in applications requiring fast curing processes, such as in adhesives, coatings, sealants, and 3D printing materials. Acrylates possess a double bond, which is highly reactive in the presence of free radicals, allowing them to form long polymer chains with varying degrees of crosslinking. This flexibility in polymer structure endow acrylates with unique properties such as optical clarity, UV resistance, and flexibility, depending on the specific formulation. In industries like automotive, aerospace, and biomedical devices, acrylate-based materials are featured for their robust performance in tough environments. Additionally, modified acrylates such as methacrylates are also featured for multifunctions and with greater control over polymers' properties including rigidity, toughness, and adhesion.

For acrylate, the molecule I selected have typical ester/acrylate functional groups, the rest of the part containing both carbon chains and aromatic structures. The molecule is not over complicated acrylate example but have all functional groups for GPT to learn and it's very representative. This molecule obtains vinyl end groups which are highly reactive in polymerization reactions, making it a good candidate for studying acrylate reactivity as well.

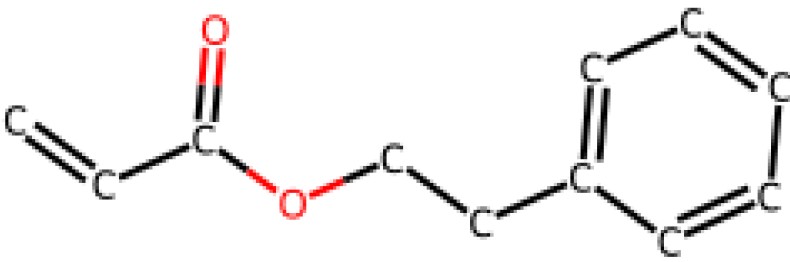

*Figure 11.* Input molecule from Acrylates

**Prompt**

I want you to think like a chemist performing a detailed analysis of the chemical composition of an acrylates through its constituent motifs. I will highlight for you 7 of the substructures of a molecule. They are numbered one-by-one from Motif 0 to Motif 6, inclusive. I want you to explain, concisely, what each numbered motif is. Make sure to start from Motif 0 and go in order of the numbering. MAKE SURE you describe EVERY MOTIF!

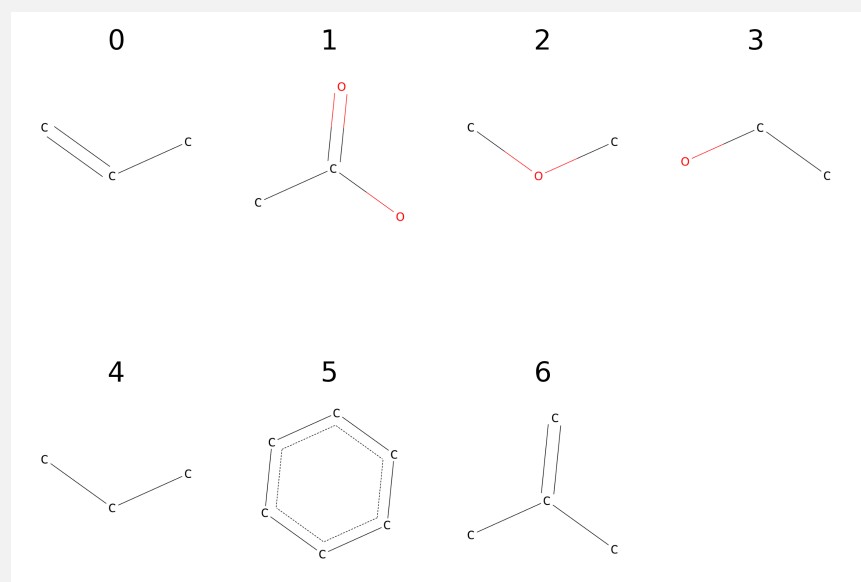

Sure, here's a concise description of each numbered motif as per the image provided:

**Motif 0**: This motif is a vinyl group, which consists of two carbon atoms connected by a double bond (C=C), with each carbon atom having one additional connection for a total of three.

**Motif 1**: This is an acrylate group, also known as a propenoate ester. It consists of a carbonyl group (C=O) attached to a carbon atom, which also has an oxygen atom bonded to another carbon atom (an ester linkage).

**Motif 2**: This motif is an ethyl ether group, specifically an ether where an oxygen atom connects two ethyl (CH3-CH2) groups.

**Motif 3**: This is a methoxy group, where an oxygen atom is attached to a single methyl group (CH3).

**Motif 4**: This is a simple propyl group, consisting of three carbon atoms in a chain (CH3-CH2-CH2).

**Motif 5**: This is a benzene ring, also known as an aromatic ring, which is made up of six carbon atoms arranged in a hexagonal ring with alternating double bonds between carbon atoms.

**Motif 6**: This is an isopropyl group, a branched chain comprising three carbon atoms where the central carbon is connected to two methyl groups (CH3-CH-CH3).

Each motif represents a fundamental building block that can contribute to the structure and properties of an acrylate molecule.

This is an easy and straightforward prompt, with clear instructions and restrictions that make it easy for GPT to follow. By specifying the total number of motifs and requiring a sequential explanation, the task provides a logical structure that minimizes the potential for errors.

GPT's response is entirely accurate, offering correct and concise explanations for all motifs. Each motif's functionality is clearly described, reflecting a thorough understanding of their roles within the acrylate molecule. This approach ensures clarity and completeness, effectively addressing the prompt's requirements.

I want you to think like a chemist performing a detailed analysis of the chemical composition of an acrylate through its constituent motifs. I will highlight for you some of the distinctive fragments of a molecule. They are numbered from 0 and individually highlighted in GREEN. Focus ONLY on the substructure highlighted in green within each cell. Here is the descriptions for each substructure provided by an expert:

Sure, here's a concise description of each numbered motif as per the image provided:

**Motif 0**: This motif is a vinyl group, which consists of two carbon atoms connected by a double bond (C=C), with each carbon atom having one additional connection for a total of three.

**Motif 1**: This is an acrylate group, also known as a propenoate ester. It consists of a carbonyl group (C=O)

attached to a carbon atom, which also has an oxygen atom bonded to another carbon atom (an ester linkage).
**Motif 2**: This motif is an ethyl ether group, specifically an ether where an oxygen atom connects two ethyl (CH3-CH2) groups.
**Motif 3**: This is a methoxy group, where an oxygen atom is attached to a single methyl group (CH3).
**Motif 4**: This is a simple propyl group, consisting of three carbon atoms in a chain (CH3-CH2-CH2).
**Motif 5**: This is a benzene ring, also known as an aromatic ring, which is made up of six carbon atoms arranged in a hexagonal ring with alternating double bonds between carbon atoms.
**Motif 6**: This is an isopropyl group, a branched chain comprising three carbon atoms where the central carbon is connected to two methyl groups (CH3-CH-CH3).
Each motif represents a fundamental building block that can contribute to the structure and properties of an acrylate molecule.
I want you to tell me if any two of them should be combined together to form a more meaningful substructure. Remember the defining acrylate group is C=CC(=O)O, which consists of a carbon-carbon double bond and a carboxylate ester. Your task is to construct this defining acrylate group by combining the alkene or vinyl group with the ester group. If the acrylate group is already present, your task is to highlight any other noteworthy functional groups. Output a single pair of numbers if you think those two fragments should be combined, and a brief explanation why. If no such pairs exist, don't output anything.

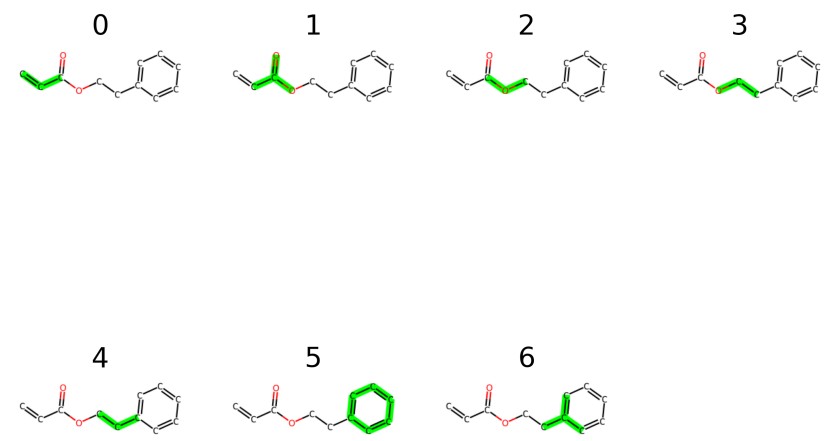

**Response**

1 0. The reason for this particular combination is to form the defining acrylate group, C=CC(=O)O. By combining Motif 1 (the acrylate/propenoate ester featuring a carbonyl carbon bonded to an oxygen atom linked by an ester bond to another carbon) with Motif 0 (the vinyl group, an alkene with two carbon atoms connected by a double bond), we achieve the desired acrylate structure containing a carbon-carbon double bond and a carboxylate ester.

**Comments by Expert**

Correct, 0 and 1 should be combined together, as they appropriately form the acrylate group. This combination integrates the vinyl group's carbon-carbon double bond (Motif 0) with the ester functionality (Motif 1), resulting in the characteristic structure of acrylates. The answer demonstrates a clear understanding of the fundamental chemical features and their assembly into the essential acrylate group.

**Prompt**

I want you to think like a chemist performing a detailed analysis of the chemical composition of an acrylates through its constituent motifs. I will highlight for you 6 of the substructures of a molecule. They are numbered one-by-one from Motif 0 to Motif 5, inclusive. I want you to explain, concisely, what each numbered motif is. Make sure to start from Motif 0 and go in order of the numbering. MAKE SURE you describe EVERY MOTIF!

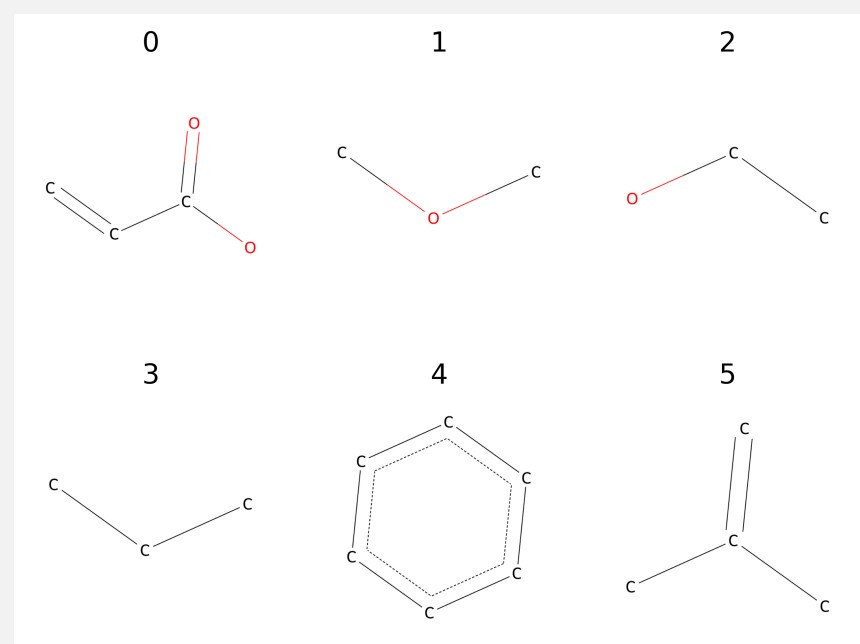

Certainly, here is a concise characterization of each motif:

**Motif 0:** This is an ester group with an adjacent carbon-carbon double bond (C=C). The structure can be described as a vinyl ester derived from acrylic acid. Specifically, it contains a carbonyl group (C=O) adjacent to an oxygen single-bonded (O—C) to a vinyl group (—CH=CH2).

**Motif 1:** This is an ether group, characterized by an oxygen atom linked to two alkyl groups, which are represented here as carbon atoms (C—O—C). This motif shows the simplest ether functional group.

**Motif 2:** This is another ether group, but oriented differently in comparison to Motif 1. Here, an oxygen atom is bonded to two carbons in a linear fashion again forming the general formula of an ether (R—O—R').

**Motif 3:** This is an alkane chain with three carbons, indicating a simple linear hydrocarbon with single bonds between the carbon atoms (C—C—C). This motif is a fragment of a longer alkane chain or a part of more complex hydrocarbons.

**Motif 4:** This motif is a benzene ring, signified by its hexagonal structure with alternating double bonds. Benzene is an aromatic hydrocarbon with significant stability due to its conjugated pi electron system.

**Motif 5:** This is an isopropenyl group, described as an alkene in which the central carbon is double-bonded to one carbon and single-bonded to two other carbons (other than hydrogen). This motif presents a branching point in the structure with a double bond.

These motifs collectively depict a variety of functional groups and structural elements commonly found in the chemistry of acrylates and related organic compounds.

**Comments by Expert**

The answer is Correct and provides correct descriptions for all motifs from 0 to 5. Each motif is clearly identified and concisely characterized, reflecting a solid understanding of their functional and structural significance. The analysis effectively captures the diversity of functional groups, including esters, ethers, alkanes, aromatics, and alkenes, aligning with the prompt's requirements.

**Prompt**

I want you to think like a chemist performing a detailed analysis of the chemical composition of an acrylate through its constituent motifs. I will highlight for you some of the distinctive fragments of a molecule. They are numbered from 0 and individually highlighted in GREEN. Focus ONLY on the substructure highlighted in green within each cell. Here is the descriptions for each substructure provided by an expert:

Certainly, here is a concise characterization of each motif:

**Motif 0:** This is an ester group with an adjacent carbon-carbon double bond (C=C). The structure can be described as a vinyl ester derived from acrylic acid. Specifically, it contains a carbonyl group (C=O) adjacent to an oxygen single-bonded (O—C) to a vinyl group (—CH=CH2).

**Motif 1:** This is an ether group, characterized by an oxygen atom linked to two alkyl groups, which are represented here as carbon atoms (C—O—C). This motif shows the simplest ether functional group.

**Motif 2:** This is another ether group, but oriented differently in comparison to Motif 1. Here, an oxygen atom is bonded to two carbons in a linear fashion again forming the general formula of an ether (R—O—R').

**Motif 3:** This is an alkane chain with three carbons, indicating a simple linear hydrocarbon with single bonds between the carbon atoms (C—C—C). This motif is a fragment of a longer alkane chain or a part of more complex hydrocarbons.

**Motif 4:** This motif is a benzene ring, signified by its hexagonal structure with alternating double bonds. Benzene is an aromatic hydrocarbon with significant stability due to its conjugated pi electron system.

**Motif 5:** This is an isopropenyl group, described as an alkene in which the central carbon is double-bonded to one carbon and single-bonded to two other carbons (other than hydrogen). This motif presents a branching point in the structure with a double bond.

These motifs collectively depict a variety of functional groups and structural elements commonly found in the chemistry of acrylates and related organic compounds.

I want you to tell me if any two of them should be combined together to form a more meaningful substructure. Remember the defining acrylate group is C=CC(=O)O, which consists of a carbon-carbon double bond and a carboxylate ester. Your task is to construct this defining acrylate group by combining the alkene or vinyl group with the ester group. If the acrylate group is already present, your task is to highlight any other noteworthy functional groups. Output a single pair of numbers if you think those two fragments should be combined, and a brief explanation why. If no such pairs exist, don't output anything.

### Response

For acrylate chemistry, the defining motif is C=CC(=O)O, which consists of a carbon-carbon double bond and a carboxylate ester group. Let's analyze the motifs:

**Motif 0:** - Contains the carbonyl group (C=O) and ester (O-R) parts, along with a carbon-carbon double bond, indicating it already represents an acrylate substructure.

**Motifs 1, 2, 3, 4, and 5:** - None of these motifs independently contain the defining features of an acrylate group. Motifs 1, 2, and 5 are ether and isopropenyl groups, while 3 is an alkane chain and 4 is a benzene ring.

**Result:** The acrylate group is already present in Motif 0, already encompassing both the ester and the vinyl group. There is no need to combine any other motifs to form the acrylate group, as it is already fully represented. No pairs to combine, as the defining acrylate motif is already contained within Motif 0.

### Comments by Expert

This response from GPT is fully correct. It correctly identifies Motif 0 as already the defining acrylate group, which includes the essential carbon-carbon double bond and carboxylate ester group. No additional combinations are necessary, the response demonstrates a clear understanding of the task requirements and provides a precise explanation of why no other motifs need to be combined.

## H.5. Case Study: Chain Extenders

Chain extenders are small, difunctional molecules used in polymer chemistry to extend and link polymer chains, thereby increasing the molecular weight and enhancing the physical properties of the resulting polymer. The bonding process triggered by chain extenders lead to materials with improved mechanical properties, such as higher tensile strength, flexibility, and impact resistance. Chain extenders are particularly important in applications requiring tough yet flexible materials, such as elastomers, foams, and coatings. In polyurethane systems, for example, chain extenders typically have two reactive groups, such as hydroxyl or amine groups, which react with isocyanates or other functional groups to create strong, covalent bonds between polymer segments. The choice of chain extenders can hugely impact the final material properties; for example, the use of diamines versus diols in polyurethanes can significantly influence polymers' elasticity and thermal stability. chain extenders also play a role in optimizing the processing conditions and curing times of the polymers, making them really important component in polymer synthesis.

For molecular extenders, the molecule is being chosen for its level of complexity, it provided a nice symmetrical structure contains most of the significant functional motifs that are critical in molecular backbones expansion, polymerization and molecular backbone's flexibility.

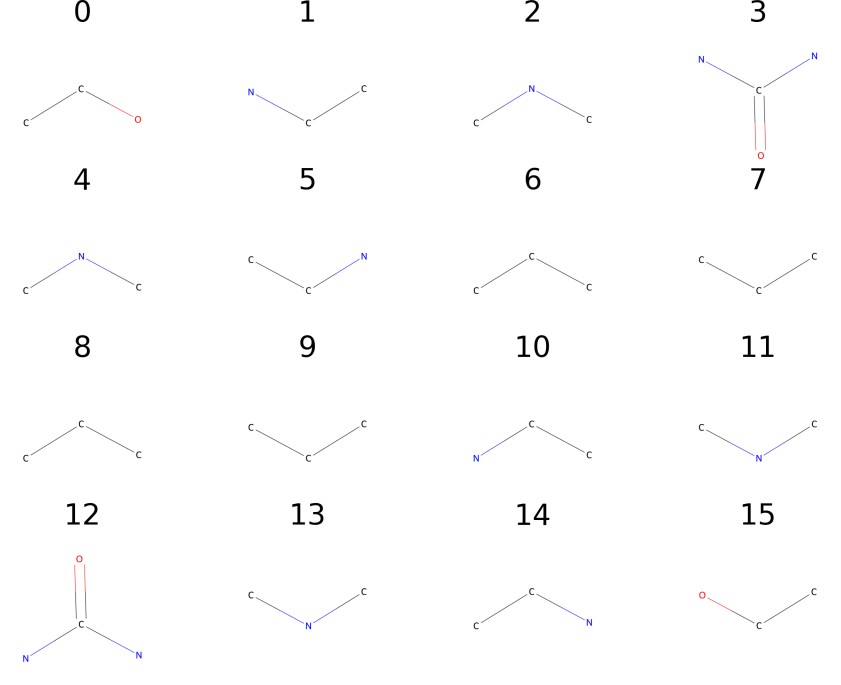

*Figure 12.* Input molecule from Chain Extenders

simple three-carbon aliphatic chain (C-C-C). 11. **Motif 10**: This is another linear aliphatic fragment, similar to Motif 8 and 9 (C-C-C). 12. **Motif 11**: This is a linear di-N-C-C fragment featuring two secondary amines. 13. **Motif 12**: This is a cyanoformamide (N-C=O) motif similar to Motif 3 but with two nitrogen atoms. 14. **Motif 13**: This is a linear N-C-C motif featuring a single secondary amine. 15. **Motif 14**: This is another linear N-C-C motif featuring a single secondary amine, similar to Motif 13. 16. **Motif 15**: This structure is an ester functional group, similar to Motif 0 but arranged differently within the chain (C-O-C=O).

**Comments by Expert**

Rate of prompt: This is an easy and straightforward prompt, requiring only the identification and explanation of each motif individually. The clear labeling of the motifs and their corresponding numbers ensures that the task is well-structured and unambiguous. Additionally, the explicit emphasis on describing every motif prevent GPT from grouping similar structures into generalized explanations, thereby maintaining the specificity and accuracy of the analysis. GPT's answer: fully correct, GPT describe each motif accurately and in detailed explanations. GPT effectively distinguishing between subtle variations in structure.

**Prompt**

I want you to think like a chemist performing a detailed analysis of the chemical composition of a chain extender through its constituent motifs. I will highlight for you some of the distinctive fragments of a molecule. They are numbered from 0 and individually highlighted in RED. Focus ONLY on the substructure highlighted in red within each cell. Here is the descriptions for each substructure provided by an expert:

Sure, I'll provide a concise description of each motif:

1. **Motif 0**: This structure is an ester functional group, specifically an alkyl chain with an ester linkage (C-O-C=O). 2. **Motif 1**: This is an imine (Schiff base) structure, featuring a double bond between carbon and nitrogen (C=N). 3. **Motif 2**: This is a secondary amine, consisting of a carbon-nitrogen single bond (C-N-C). 4. **Motif 3**: This is a nitrile group with an adjacent carbonyl, forming a structure known as a cyanoformamide (N-C=O). 5. **Motif 4**: This is a secondary amine with two different R groups (N-C-C). 6. **Motif 5**: This is again a secondary amine but with a different arrangement compared to Motif 2 (C-C-N). 7. **Motif 6**: This is a linear aliphatic chain fragment (C-C-C). 8. **Motif 7**: This is a linear N-C-C-N fragment featuring two secondary amines. 9. **Motif 8**: This is a simple three-carbon aliphatic chain (C-C-C). 10. **Motif 9**: This is another simple three-carbon aliphatic chain (C-C-C). 11. **Motif 10**: This is another linear aliphatic fragment, similar to Motif 8 and 9 (C-C-C). 12. **Motif 11**: This is a linear di-N-C-C fragment featuring two secondary amines. 13. **Motif 12**: This is a cyanoformamide (N-C=O) motif similar to Motif 3 but with two nitrogen atoms. 14. **Motif 13**: This is a linear N-C-C motif featuring a single secondary amine. 15. **Motif 14**: This is another linear N-C-C motif featuring a single secondary amine, similar to Motif 13. 16. **Motif 15**: This structure is an ester functional group, similar to Motif 0 but arranged differently within the chain (C-O-C=O).

I want you to tell me if any two of them should be combined together to form a more meaningful substructure. Remember that chain extenders often contain repeating units like ether, ester, amide, or urethane groups, and the goal is to extend the molecular backbone. Your task is to highlight the primary functional groups of the molecule. Output a single pair of numbers if you think those two fragments should be combined, and a brief explanation why. If no such pairs exist, don't output anything.

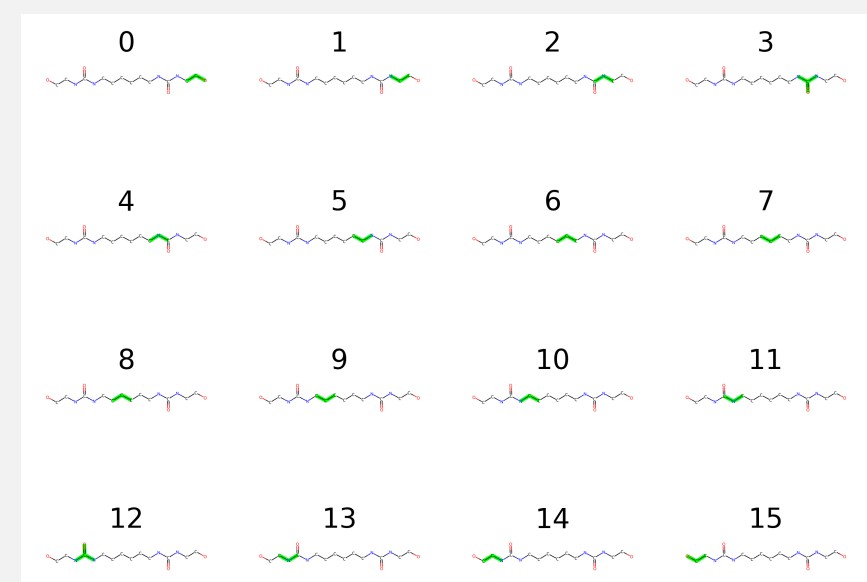

### Response

**Pair to Combine: Motif 2 and Motif 4**

**Explanation:**

- Both motifs feature secondary amines. - When combined, they could form a segment of a polyamine chain which is a common segment in certain polymers, potentially contributing to the chain extension and backbone of the resulting polymeric material.

### Comments by Expert

The prompt is easy and straightforward, requiring an evaluation of whether any pair of motifs should be combined, with detailed guidance on the types of functional groups commonly found in chain extenders, such as ether, ester, amide, or urethane groups. GPT's answer: GPT correctly identified Motif 2 and Motif 4 as suitable for combination, noting that both contain secondary amines and align with the goal of extending molecular backbones. Additionally, the combined structure not only satisfies the functional requirements but also provides enhanced stability, which is crucial in the context of polymeric material design for chain extenders.

### Prompt

I want you to think like a chemist performing a detailed analysis of the chemical composition of a chain extender through its constituent motifs. I will highlight for you some of the distinctive substructures of a chain extender. They are numbered from 0. Here are the textual descriptions of each motif:

Sure, I'll provide a concise description of each motif:

1. **Motif 0**: This structure is an ester functional group, specifically an alkyl chain with an ester linkage (C-O-C=O). 2. **Motif 1**: This is an imine (Schiff base) structure, featuring a double bond between carbon and nitrogen (C=N). 3. **Motif 2**: This is a secondary amine, consisting of a carbon-nitrogen single bond (C-N-C). 4. **Motif 3**: This is a nitrile group with an adjacent carbonyl, forming a structure known as a cyanoformamide (N-C=O). 5. **Motif 4**: This is a secondary amine with two different R groups (N-C-C). 6. **Motif 5**: This is again a secondary amine but with a different arrangement compared to Motif 2 (C-C-N). 7. **Motif 6**: This is a linear aliphatic chain fragment (C-C-C). 8. **Motif 7**: This is a linear N-C-C-N fragment featuring two secondary amines. 9. **Motif 8**: This is a simple three-carbon aliphatic chain (C-C-C). 10. **Motif 9**: This is another simple three-carbon aliphatic chain (C-C-C). 11. **Motif 10**: This is another linear aliphatic fragment, similar to Motif 8 and 9 (C-C-C). 12. **Motif 11**: This is a linear di-N-C-C fragment featuring two secondary amines. 13. **Motif 12**: This is a cyanoformamide (N-C=O) motif similar to Motif 3 but with two nitrogen atoms. 14. **Motif 13**: This is a linear N-C-C motif featuring a single secondary amine. 15. **Motif 14**: This is another linear N-C-C motif featuring a single secondary amine, similar to Motif 13. 16. **Motif 15**: This structure is an ester functional group, similar to Motif 0 but arranged differently within the chain (C-O-C=O).

I want you to pick only ONE of these as the root motif most essential to its chemical profile. It should be the single most important motif the rest of the chain extender was built around. Remember that chain extenders often contain repeating units like ether, ester, amide, or urethane groups, which help extend the molecular backbone, so your selected root motif MUST contain one of these repeating units. If there are multiple such motifs, or one doesn't clearly stand out, just pick one of them. Give your answer as a single number. Explain your reasoning carefully.

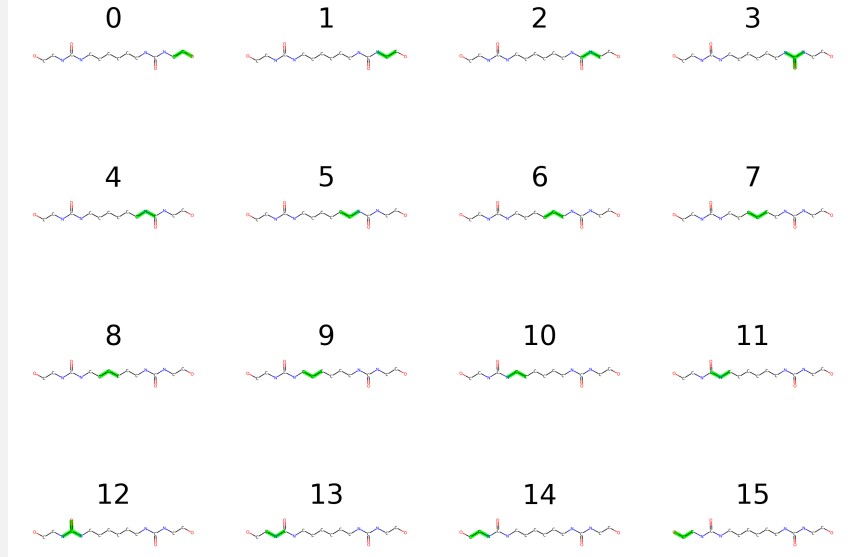

### Response

After examining the given motifs, the root motif should possess a fundamental repeating unit pivotal for building the chain extender's molecular backbone. Esters, amides, and related functional groups are significant in forming backbones due to their capability to extend polymer chains. Among the given motifs, ester groups are notable candidates for this role.

Motif 0 and Motif 15 are both ester functional groups (C-O-C=O), which are commonly used as chain extenders due to their ability to form strong and flexible polymer chains. These motifs provide the necessary repetition for polymer chain growth and flexibility.

Since Motif 0 and Motif 15 are both relevant and appear similar, either could be considered. For simplicity and based on initial numbering, I will select **Motif 0**. This choice ensures the selection of a distinct ester functional group crucial to the overall chemical profile of the chain extender.

Therefore, the single most essential motif for the chemical profile of the chain extender is:

**0**

### Comments by Expert

The prompt is of medium complexity, this prompt let GPT pick the most important motif, which can sometimes result in discrepancies between the answers from language models and chemistry experts, especially when multiple motifs hold comparable levels of significance in the molecule's functionality. GPT's answer: GPT answered motifs 0 and 15, and mentioned that the reason is because they contain the ester functional group that are necessary for polymer chain growth and the flexibility of the chain. This is correct. By choosing Motif 0 based on its position and relevance, GPT adhered to the prompt's instruction to select one root motif. But from the perspective of a chemistry expert, motifs 3 and 12 are also important structure in chain extend molecules. which feature cyanoformamide structures, are also critical to the functionality of chain extenders. These motifs may contribute additional chemical versatility, such as enhancing stability or enabling specific interactions within the polymer backbone. Despite this, the prompt's requirement to select only one motif led GPT to prioritize ester groups, a valid choice given their well-established importance in polymer chemistry. In general, GPT's answer is correct, while a broader analysis might highlight the significancy of other motifs like 3 and 12, the chosen answer reflects a reasonable and informed prioritization of ester groups as key contributors to the molecular design of chain extenders.

## H.6. Quantitative Evaluation of Case Studies

*Table 9.* Tallying the turn-by-turn expert feedback across all case studies in App. H.1-H.5.

| Dataset | | Easy | Medium | Hard |
|---|---|---|---|---|
| Small Dataset | Correct | 6 | 3 | 0 |
| | Partial | 0 | 1 | 0 |
| Real-World Dataset | Correct | 5 | 2 | 2 |
| | Partial | 0 | 1 | 0 |

We tally all turn-by-turn comments by the expert and categorize each MMFM response as either correct, partially correct, or wrong. In Table 9, we see there were no instances where the expert thought GPT4's explanation was flat out wrong. In all but two instances, the expert completely agrees with GPT4's explanation.

## H.7. Concluding Thoughts by Expert

The tasks ranged from simple identification tasks to more complex evaluations of motif interactions and significance. The agent generally performed well, delivering accurate, detailed, and logical responses. Its performance varied depending on the prompt's complexity and the nuanced requirements of the chemistry domain.

For simple prompts, requiring basic identification or description of motifs, the agent excelled due to clear instructions, providing precise answers aligned with chemical reasoning and minimal errors. For mid-level tasks, such as combining functional groups or selecting significant motifs (e.g., case studies H.5, H.1), the agent performed well but occasionally lacked nuance in addressing chemical subtleties. These tasks demanded more empirical knowledge, such as understanding toxicity or stability, which increased their complexity. In high-complexity tasks, like evaluating interactions or reactivity (e.g., case study H.2), the agent demonstrated strong reasoning but sometimes missed broader implications or alternative perspectives due to limited empirical insights.

The agent consistently delivered accurate and structured answers and adhered to task requirements. However, limitations emerged in tasks needing experiential knowledge or where multiple valid interpretations existed, as it prioritized one perspective without exploring alternatives unless prompted. Overall, the agent performs exceptionally well in structured chemical analysis, offering clear and precise explanations, but requires refinement for tasks involving deeper interpretative depth or expert intuition.

