# OpenReview forum: "Foundation Molecular Grammar: Multi-Modal Foundation Models Induce Interpretable Molecular Graph Languages"
_ICML.cc/2025/Conference — ICML 2025 poster_

### Official Review · Reviewer_PGBN · 2025-03-12

**Overall Recommendation:** 2

**Summary:**

This paper introduces Foundation Molecular Grammar (FMG). This approach uses a multimodal large language model to identify meaningful substructures in molecular graphs, generating an interpretable grammar for molecule generation. By rendering molecules as images and prompting the model with specialized prompts, FMG enforces chemically valid decomposition steps and captures motifs without heavy expert annotation.

**Claims And Evidence:**

While the paper repeatedly highlights the novel idea of using a multi-modal foundation model (MMFM) to induce a Foundation Molecular Grammar(FMG), it is still not evident how this method solves a pressing issue in molecular discovery workflows.

**Essential References Not Discussed:**

NA

**Experimental Designs Or Analyses:**

The authors emphasize interpretability as a major advantage of using FMG. However, the paper only provides limited examples in Table 3 of the interpretability in action—mostly brief snippets. A deeper discussion or demonstration (e.g., a domain-expert walkthrough showing how the FMG clarifies or improves the design process) would reinforce the claim that this approach is uniquely transparent or insightful.

**Methods And Evaluation Criteria:**

The paper evaluates the proposed method on small, specialized monomer datasets plus two real-world datasets (HOPV, PTC). However, the field of molecular generation typically uses larger, more standard benchmarks (e.g., ZINC250k, MOSES) to compare performance comprehensively. Without these broader benchmarks or additional state-of-the-art methods, the generality and competitiveness of the proposed approach remain unclear.

**Other Comments Or Suggestions:**

The notation in tables needs to be well-defined. For example, what does each column mean in the table?

**Other Strengths And Weaknesses:**

NA.

**Questions For Authors:**

Please refer to previous sections.

**Relation To Broader Scientific Literature:**

Using MMFM in graph generative tasks is novel and sounds promising.

**Theoretical Claims:**

There are no theoretical proofs in the paper.

---

> ### Author Rebuttal · Authors · 2025-04-01
>
> *The authors emphasize interpretability as a major advantage of using FMG. However, the paper only provides limited examples.... A deeper discussion or demonstration (e.g., a domain-expert walkthrough...) would reinforce the claim that this approach is uniquely transparent or insightful.*
>
> * The examples in Table 3 are actually drawn from one of the five full example domain-expert walkthroughs in Appendix D, where the expert critiques FMG’s reasoning step-by-step. Additionally, in Appendix D.6, we include concluding thoughts delivered by the expert, highlighting LLM agents’ strong performance in substructure extraction and limits in harder tasks needing expert intuition. The modular nature of the decomposing process is key to FMG’s interpretability, and the case studies highlight how FMG makes this clearer.
>
> **Additional Case Study:** As both you and V4pu suggested, we ran another case study where an expert contrasted discrepant decompositions by their rationale. Due to the short rebuttal window, we only finished this for 11 chain extenders but are actively working on other datasets. The setup is: the expert chooses a (good, bad) pair of decompositions for each molecule and summarizes the rationale. We then ask our LLM-as-a-judge to summarize the key points and decide solely based on the explanations, which decomp. is better. In most cases, the LLM judge identifies the good decomposition, showing how LLM explanations are useful in closing its own design loop.
>
> For example, the expert wrote:
>
> “Analysis A demonstrates a solid understanding of motifs like amide, urea, imidazole, and carbonyl groups—key to chain extenders. It justifies motif 1 as the root based on its role in peptide bond formation. Analysis B neglects important groups and lacks a strong rationale for selecting motif 0 as root.”
>
> GPT similarly judged:
>
> **Defining Motifs:**
>
> Analysis A better identifies functional groups like amides/carbonyls that define chain extender properties. It correctly selects motif 1 for its role in polymer backbones.
>
> **Functional Group Explanation:**
>
> Analysis A explains the contributions of groups to mechanical and processing properties. Analysis B simplifies the motifs and lacks depth in chemical reasoning.
>
> **Decision:**
>
> Analysis A is favored for better understanding and detailed explanation of groups defining chain extenders.
>
> We repeated the 11-molecule test 5 times and found LLM agreement with the expert on (10, 7, 8, 8, 9) runs.
>
> This case study mirrors expert round-table discussions, where competing rationales must be contrasted to make progress. In App. C, we show selecting higher-ranked decompositions judged by the LLM improves class membership; other design goals could benefit similarly.
>
> We hope this presents further quantitative evidence for the usefulness of the explanations.
>
> *While the paper repeatedly highlights the novel idea of using a multi-modal foundation model (MMFM) to induce a Foundation Molecular Grammar (FMG), it is still not evident how this method solves a pressing issue in molecular discovery workflows.... the field of molecular generation typically uses larger, more standard benchmarks to compare performance comprehensively. Without these broader benchmarks or additional state-of-the-art methods, the generality and competitiveness of the proposed approach remain unclear.*
>
> * Thanks for these two comments, which go hand-in-hand. We believe the real issue in molecular discovery is the lack of small, domain-specific benchmarks supporting interpretability and expert input. Standard benchmarks focus on large-scale representation learning, and less so on automating interpretable, expert-knowledge-guided design. On the other hand, domain-specific designs in our experience often come with only a handful of training examples, both because the domain is narrow and because the property values of interest are costly to obtain.
>
> * To evaluate FMG’s generality and competitiveness, we do compare with pretrained and transfer learning models in Tables 1 and 2. We see their performance drops significantly due to failing constraints like synthesizability, class membership while retaining broad coverage. When tackling synthetically accessible chain extenders with amine groups, we believe expert knowledge plays a critical role—but integrating it via annotations traditionally required costly manual labor. FMG seeks to automate this labor, doing the hierarchical decompositions while respecting explicit/implicit constraints, while staying interpretable for expert validation.
>
> *The notation in tables needs to be well-defined. For example, what does each column mean in the table?*
>
> * They are standard metrics used by prior work (e.g. [1,2]). We’ll clarify these notations in the revision.
>
> [1] Data-efficient graph grammar learning for molecular generation. ICLR 2022.
>
> [2] Representing Molecules as Random Walks Over Interpretable Grammars. ICML 2024.

---

> > ### Comment · Reviewer_PGBN · 2025-04-08
> >
> > I appreciate the authors’ efforts in addressing my concerns; however, they remain only partly resolved. Specifically:
> > 1. Interpretability: Demonstrating interpretability solely through a handful of case studies is insufficient. More extensive experiments are needed to establish the method’s interpretability convincingly.
> > 2. Dataset and Metrics: I continue to believe the study should include larger, commonly used datasets like ZINC250k and incorporate additional evaluation metrics (e.g., FCD, Scaf, NSPDK). These would more thoroughly validate the method and situate its performance relative to established benchmarks.

---

> > > ### Author Response · Authors · 2025-04-09
> > >
> > > **More Benchmarking on Interpretability.**
> > >
> > > We appreciate your continued engagement and agree evaluating interpretability rigorously is essential. As it is inherently qualitative, we believe expert-guided assessments offer the most meaningful validation. Our goal is to assist domain experts in decomposing and designing molecules more effectively—making their judgment the ground truth for FMG.
> > >
> > > To go beyond qualitative discussion, we’ve now completed our quantitative benchmark on the 4 remaining datasets. For each, we randomly selected 15 molecules, generated two candidate decompositions, and asked an expert to select the one with the more chemically sound reasoning. We then prompted our MMFM to make the same choice. Each pair was evaluated twice with flipped order to eliminate bias.
> > >
> > > ||Isocy.||Acry.||HOPV||PTC||Total|
> > > |-|-|-|-|-|-|-|-|-|-|
> > > |Gold|A|B|A|B|A|B|A|B||
> > > |Score|$6/12$|$9/12$|$12/15$|$13/15$|$9/15$|$13/15$|$6/12$|$9/12$|$77/108~=71$%|
> > >
> > > In 6 additional cases, the expert found both designs equally reasonable, which we excluded. Against a random baseline, the MMFM’s agreement rate yields $p=1.1e-5$, indicating statistically significant alignment with expert judgment.
> > >
> > > These results reinforce our hypothesis that FMG’s hierarchical, interpretable structure can support downstream decision-making and design critique—even enabling the LLM to act as a self-checking agent in an expert-in-the-loop pipeline.
> > >
> > > **Additional Evaluation Metrics.**
> > >
> > > In response to your suggestion, we added FCD, Scaf, and NSPDK metrics for all major baselines (ICL, VAE, CLMs), shown below.
> > >
> > > |Method|Valid (Avg.)|FCD(↓)|||Scaf(↑)|||NSPDK(↓)|||
> > > |-|-|-|-|-|-|-|-|-|-|-|
> > > |MoLeR (I)|100%|35.63|17.34|26.32|0.0|0.0|0.05|0.19|0.12|0.12|
> > > |GPT4 (ICL)|91%|18.23|7.87|19.33|0.05|0.58|0.55|0.10|0.03|0.08|
> > > |MolT5 (I)|76%|19.88|16.28|40.91|0.0|0.0|0.0|0.51|0.71|0.31|
> > > |Text+ChemT5 (I)|42%|13.78|14.70|24.65|0.08|0.64|0.95|0.27|0.11|0.13|
> > > |FMG|100%|17.67|9.63|22.46|0.0|0.15|0.0|0.13|0.08|0.11|
> > >
> > > |Method|Valid (Avg.)|FCD(↓)|||Scaf(↑)|||NSPDK(↓)|||
> > > |-|-|-|-|-|-|-|-|-|-|-|
> > > |MoLeR (I)|100%|35.63|17.34|26.32|0.0|0.0|0.05|0.19|0.12|0.12|
> > > |GPT4 (ICL)|91%|18.23|7.87|19.33|0.05|0.58|0.55|0.10|0.03|0.08|
> > > |MolT5 (I)|76%|19.88|16.28|40.91|0.0|0.0|0.0|0.51|0.71|0.31|
> > > |Text+ChemT5 (I)|42%|13.78|14.70|24.65|0.08|0.64|0.95|0.27|0.11|0.13|
> > > |FMG|100%|17.67|9.63|22.46|0.0|0.15|0.0|0.13|0.08|0.11|
> > >
> > > FMG consistently outperforms MolT5 (I) and MoLeR (I). While ICL and ChemT5 show advantages on certain metrics, their low uniqueness (ICL) & validity (ChemT5) raise questions about their practical reliability—echoing our point to V4pu on holistic usefulness.
> > >
> > > **Performance on larger benchmarks (ZINC, MOSES).**
> > >
> > > While FMG is tailored for expert-in-the-loop, domain-specific design, we agree it’s valuable to evaluate its behavior on broader benchmarks.
> > >
> > > Given limited time, we trained FMG on a 1k subset (0.05%) of ZINC250k and evaluated using the MOSES benchmark. We generated 30k samples and computed standard -TestSF metrics. (Details: no data splitting for grammar induction; held-out test set for evaluation; IntDiv2 omitted due to redundancy with IntDiv.)
> > >
> > > Results (numbers from MOSES leaderboard):
> > >
> > > |Model|Valid(↑)|Unique@1k(↑)|Unique@10k(↑)|FCD-TestSF(↓)|SNN-TestSF(↑)|Scaf-TestSF(↑)|IntDiv(↑)|Novelty(↑)|
> > > |--------------|--------------|--------------|----------------|----------------|----------------|----------------|--------------|----------------|
> > > |Train|1.00|1.00|1.00|0.48|0.59|0.00|0.86|1.00|
> > > |HMM|0.08±0.03|0.62±0.12|0.57±0.14|25.43±2.56|0.38±0.01|0.05±0.02|0.85±0.04|1.00±0.00|
> > > ...3 rows omitted...
> > > |CharRNN|0.97±0.03|1.00±0.00|1.00±0.00|**0.52±0.04**|0.56±0.01|0.11±0.01|0.86±0.00|0.84±0.05|
> > > |VAE|0.98±0.00|1.00±0.00|1.00±0.00|0.57±0.03|**0.58±0.00**|0.06±0.01|0.86±0.00|0.69±0.01|
> > > |JTN-VAE|1.00±0.00|1.00±0.00|1.00±0.00|0.94±0.05|0.52±0.01|0.10±0.01|0.86±0.00|0.91±0.01|
> > > |LatentGAN|0.90±0.00|1.00±0.00|1.00±0.00|0.83±0.01|0.51±0.00|0.11±0.01|0.86±0.00|0.95±0.00|
> > > |FMG (0.05%)|**1.00±0.00**|**1.00±0.00**|**1.00±0.00**|26.30±0.41|0.29±0.00|**0.12±0.00**|**0.90±0.00**|**1.00±0.00**|
> > >
> > > FMG leads on all five unconditional generation metrics reported in our main paper. That said, we acknowledge its distributional match is weaker, as expected from a model trained on just *0.05%* of the data. However, its high validity, novelty, diversity, and uniqueness demonstrate FMG's potential as a compositional grammar backbone—suitable for downstream optimization (e.g., via MC-REINFORCE as done in DEG).
> > >
> > > **Closing.** We will incorporate these extended analyses—on interpretability, standard benchmarks, and broader metrics—into the revised manuscript, and continue to expand FMG’s benchmarking suite.
> > >
> > > We hope this update shows that FMG, while targeting a distinct use case, holds up to broader scrutiny and provides a novel, interpretable framework for molecule design. We would be truly grateful if you would consider raising your score and helping to advocate for this contribution.
> > >
> > > Warm regards,
> > >
> > > FMG authors

---

### Official Review · Reviewer_V4pu · 2025-03-12

**Overall Recommendation:** 3

**Summary:**

In this paper, the authors show that one can incorporate the “graph grammar” of molecules into a multimodal language model. Essentially, the method, called FMG, (i) takes a molecular graph as an input, (ii) extracts “features” of such a graph, (iii) represent them with images, (iv) ask a multimodal (vision-language) model to infer a grammar that governs the molecular graph via chain-of-thoughts reasoning and preference learning. The grammar can then be used for property prediction or to generate new molecular graphs. This way, the generative model can generate new molecules that are more “interpretable” since the process of generating such molecules is documented in the chain-of-thought traces.

**Claims And Evidence:**

The claims are reasonable and the method itself is well-justified, albeit quite complex, involving many building blocks. I would like to see an ablation study on each building block.

**Essential References Not Discussed:**

N/A

**Experimental Designs Or Analyses:**

The reported numbers (diversity, synthesizability, validity, and membership) don’t seem to be significant compared to the (much simpler) baselines. Moreover, the presentations of the results are problematic: (i) lower scores are often bolded (e.g. in Tab. 1), and (ii) no error-bars are reported. I am thus wondering if the claims are actually validated by the experiments.

Moreover, the main point of the method: interpretability, lacks validation. It is unclear to me whether the explanations outputted by the model are actually useful. I would strongly suggest an additional study to quantify this.

**Methods And Evaluation Criteria:**

The methods and the evaluation criteria do make sense.

**Other Comments Or Suggestions:**

See above.

**Other Strengths And Weaknesses:**

N/A

**Questions For Authors:**

Please address the issues raised above.

**Relation To Broader Scientific Literature:**

The idea of enforcing interpretability in a molecular generation is a great idea. Indeed, one of the current limitations of molecular generative models is the interpretability and synthesizability (or lack thereof) of the novel, generated molecules. So, this paper’s motivation is well-positioned within the literature.

**Theoretical Claims:**

N/A.

---

> ### Author Rebuttal · Authors · 2025-04-01
>
> *The claims are reasonable and the method itself is well-justified, albeit quite complex, involving many building blocks. I would like to see an ablation study on each building block.*
> * Thanks for acknowledging the reasonableness of our method! We do have an ablation study on each building block in Section 5.1 (Table 4). In the section, we isolate the importance of the MMFM’s role at each stage of the algorithm. We study the effect of ablating the MMFM with a known heuristic, along with a brief rationale. We also have an ablation of the FMG self-optimization in Sec 5.2 (App. C).
>
> *The reported numbers don’t seem to be significant compared to the (much simpler) baselines… presentation of results are problematic: (i) lower scores are often bolded (e.g. in Tab. 1), and (ii) no error-bars are reported.*
> * Thank you for pointing out these observations. Evaluating generative models is challenging and requires a holistic consideration of different, competing metrics. A method which scores high on one metric (e.g. synthesizability) but does catastrophically bad on another one (e.g. uniqueness) is not practical. When simple fixes don’t work (e.g., tuning sampling temperature), we make a note of it in the caption and exclude it from the rankings.
> * Regarding (i), lower scores. In Tab. 1, the two T5 (I) methods are excluded from ranking (as mentioned in the caption) due to the difficulty in obtaining sufficient valid, unique samples, meaning their seemingly higher RS/Memb. scores don’t have sufficient sample support. The same is written in the caption of Tab. 2 for the VAE (T) methods.
> * Regarding (ii), no error bars. In this case, robustness from multiple runs (the purpose of error bars) can simply be absorbed into the number of samples we generate. Generating a sufficiently large sample size can ensure greater statistical significance of the results, especially when 4 of our metrics (valid, novel, RS, memb.) are defined at the individual sample level and the other 2 metrics (diversity, uniqueness) evaluate coverage. For small molecules (Tab. 1), we generate 10000 samples; for large molecules (Tab. 2) we generate 1000 samples due to inference being more expensive for some of the large model baselines. In both cases, we see a fixed sample size of 1000 or 10000 already pushes multiple baselines to the limits, so generating more than what they’re capable of may further unbalance the comparisons (see (i)).
> * To facilitate a more holistic validation, we made the following spider plots:
> https://ibb.co/k62nj3c8
> https://ibb.co/7JK24Y82
> * Only the methods that lie on the Pareto frontier across all metrics are plotted.
>   * On small datasets, no method other than FMG reaches near 100% unique, valid & memb. simultaneously. For instance, GPT4 (ICL) appears to score higher on diversity & RS, but struggles to generate valid and unique samples.
>   * On real-world datasets, FMG generates the most unique, novel, valid & diverse samples, and methods that score higher on RS and Memb. have serious shortcomings.
> * We will add the new plots to the paper to mitigate the challenge of evaluating different generative models by using competing metrics.
>
> *...the main point of the method: interpretability, lacks validation. It is unclear to me whether the explanations outputted by the model are actually useful. I would strongly suggest an additional study to quantify this.*
> * The explanation outputs are critical for the evaluation, interpretation and self-improvement of FMG.
> * Evaluation: The GPT-generated explanations are assessed by human experts. We actually have 5 in-depth step-by-step walkthroughs in App. D, with 1 example molecule chosen per dataset (domain). Each step in each case study is scrutinized by an expert (see “Comments by Expert” highlighted in light purple).
> * For a quantitative assessment, we tallied the GPT4’s turn-by-turn answers across the 5 case studies by whether the expert completely agrees or not. We had the expert classify the difficulty of the task in each turn.
>
> |Dataset||Easy|Medium|Hard|
> |-|-|:-:|:-:|:-:|
> |Small Dataset|Correct|6|3|0|
> ||Partial|0|1|0|
> |Real-World Dataset|Correct|5|2|2|
> ||Partial|0|1|0|
> * There were no instances where the expert thought GPT4’s explanation was flat out wrong. In all but two instances, the expert completely agrees with GPT4’s explanation.
> * Interpretation: Additionally, in App. D.6, we include concluding thoughts delivered by the expert, highlighting LLM agents' strong performance in substructure extraction and limits in harder tasks requiring expert intuition.
> * Self-improvement: The explanations directly influence the final grammar by summarizing the reasoning taken to achieve the decomposition. They are the inputs for our LLM-as-a-judge protocol, which pits discrepant decompositions against each other (Sec. 3.5) in a tournament, and decides which decompositions to use on the basis of the explanation quality. See our additional case study in our response to PGBN.

---

### Official Review · Reviewer_uTAc · 2025-03-13

**Overall Recommendation:** 3

**Summary:**

The paper proposes Foundation Molecular Grammar (FMG) using multi-modal foundation models. FMG induces interpretable graph grammars by converting molecules to images and using LLMs to identify the connection between molecular substructures. It outperforms baselines in molecular generation benchmarks, excelling in synthesizability, diversity, and data efficiency. FMG provides chemical interpretability, offering a new approach for automated molecular discovery workflows.

**Claims And Evidence:**

The claims in the paper are well supported by evidence.

**Essential References Not Discussed:**

N/A

**Experimental Designs Or Analyses:**

The metrics and datasets used in the experiments are taken from earlier works on similar topics, and there appears to be no significant flaws in the experiments.

**Methods And Evaluation Criteria:**

The paper uses LLMs to replace heuristics-based rules in molecular grammar methods, which is a novel and interesting way to incorporate domain knowledge into the process of molecular grammar learning. However, the use of rendered molecular images as the input to LLMs, as opposed to text-based molecule representation formats like SMILES or SELFIES, is not sufficiently motivated. The paper should provide a comparison between these molecule representation methods.

**Other Comments Or Suggestions:**

N/A

**Other Strengths And Weaknesses:**

N/A

**Questions For Authors:**

Why did the authors use images to describe the molecules to the LLMs? Did the authors experiment with text-based formats like SMILES and SELFIES?

**Relation To Broader Scientific Literature:**

The paper combines LLMs with molecular grammar learning and brings interpretability to LLM-based molecule generation, which can be valuable for molecular discovery tasks.

**Theoretical Claims:**

The paper does not make theoretical claims.

---

> ### Author Rebuttal · Authors · 2025-04-01
>
> *The paper uses LLMs to replace heuristics-based rules in molecular grammar methods, which is a novel and interesting way to incorporate domain knowledge into the process of molecular grammar learning.*
>
> Thanks for recognizing the novelty, soundness, and intrigue of our work!
>
> *Why did the authors use images to describe the molecules to the LLMs? Did the authors experiment with text-based formats like SMILES and SELFIES?*
>
>  There are three reasons why we choose rendering molecular images as the input to LLMs:
> 1. Literature review of LLM’s chemistry comprehension abilities:
> Foundation models like GPT-4o and Gemini have shown multi-modal comprehension ability in aligning natural language descriptions with corresponding images for advanced reasoning, but there has been less work on aligning formal languages like SMILES with corresponding images. This may be because LLMs find it easier to obtain semantics from natural descriptions rather than SMILES. For instance, studies like [1,Section D] and [2] find LLMs perform great on tasks that require reasoning from the natural description of molecules, but struggle when fed SMILES/SELFIES (e.g. 0% SMILES-to-name success in [1] Section D).
> 2. Technical formulation is in terms of hypergraphs
> Our underlying formulation builds off the history of hyperedge-replacement grammars, which operate at the hyperedge (or substructure) level instead of the atom level. SMILES/SELFIES syntax doesn’t easily support substructure-level annotations and extending it to do so not only is out of the scope of this work but also changes the data used for LLM pre-training. Meanwhile, rdkit has a library of functions for highlighting, color-coding, and rendering substructures.
> 3. Interpretability of the grammar learning process.
> We thought long and hard about how the complex, hierarchical structured representation of hypergraphs can be fed into LLMs. After initial conversations with chemists, we came to the conclusion that highlighting substructures is the most visually meaningful way to consume the information. We, in parallel with a few others [3], identified an opportunity for combining the latest multi-modal understanding and cheminformatics rendering tools.
> We hope these points sufficiently motivate the visual representation input, and a summarized discussion will be added to the main text.
>
> [1] White et al. Assessment of chemistry knowledge in large language models that generate code. Digital Discovery, 2(2):368–376, 2023.
>
> [2] Guo et al. What Can Large Language Models Do in Chemistry? A Comprehensive Benchmark on Eight Tasks. NeurIPS, 2023.
>
> [3] Wang, Zixu, et al. "Image-based generation for molecule design with SketchMol." Nature Machine Intelligence (2025): 1-12.

---

> > ### Comment · Reviewer_uTAc · 2025-04-04
> >
> > The author answered my questions, but more experiments may be needed to prove the view he put forward. For example, using the textual representation or graph structure representation of molecules as input for processing, then comparing the effect differences of different types of inputs, and finally selecting the optimal data representation form.

---

> > > ### Author Response · Authors · 2025-04-07
> > >
> > > Dear uTAc,
> > >
> > > Thank you for your thoughtful follow-up and for engaging deeply with our work. To start with, we already benchmark against methods specialized for each input modality—SMILES/SELFIES (MolT5, ChemT5, GPT4-ICL), graphs (MoLeR, RW), and hypergraphs (JT-VAE, Hier-VAE, MHG, DEG). Our results (Tables 1 & 2) and qualitative analyses (see responses to V4pu and PGBN) demonstrate FMG’s superior holistic performance and interpretability.
> > >
> > > To further explore your suggestion, we conducted a new ablation study where we modified FMG to use a text-based encoding—FMG-Text—instead of molecular images. Since GPT-4o supports only text and image inputs, direct graph input is currently infeasible, though we see it as a promising future direction.
> > >
> > > **Background: MMFM inputs.** Each FMG input is a grid of cells showing:
> > >
> > > 1. A molecule,
> > > 2. A molecule with one substructure highlighted,
> > > 3. A molecule with two substructures highlighted in different colors (Sec. 3.4).
> > >
> > > While (1) can be replaced by SMILES, (2) and (3) require visual emphasis of substructures, which text-based formats struggle to express.
> > >
> > > **Attempt 1: Tagged SMILES**
> > >
> > > We added tags (e.g. <>) into SMILES to denote motif boundaries. For example:
> > >
> > > Original SMILES: C=CC(=O)OC1CC2CC1C2
> > > Motif: C=CC(=O)O\<C\>1\<CC\>2C\<C\>1\<C\>2
> > >
> > > GPT-4o failed basic comprehension tests. Though it understands that numbers denote ring closures, it became confused about which numbers were relevant to the motif’s ring.As a result, it interpreted the tagged string via straightforward character concatenation—e.g., as CCCCC, a straight-chain alkane—rather than recognizing the cyclopentane motif. We attempted to add ring numbers within the tags to clarify the motif's ring structure, but this only obfuscated the global context and further degraded the model’s understanding.
> > >
> > > **Attempt 2: Atom-number encoding** We then tried a verbose but reliable approach: number all atoms and list motif atoms. For example:
> > >
> > > Original SMILES: [C:1]=[C:2]\[C:3\](=[O:4])[O:5][C:6]1[C:7][C:8]2[C:9][C:10]1[C:11]2
> > >
> > > - (optional) Motif 1: 6,7,8,10,11
> > > - (optional) Motif 2: 8,9,10,11
> > >
> > > This improved parsing and allowed us to fully re-implement FMG with text inputs (FMG-Text) by swapping each visual cell for its text-based counterpart.
> > >
> > > We made sure everything worked as expected on a few case studies. Then, we ran our full evaluation protocol. The downstream generation results are as follows:
> > >
> > > **Results: Small Datasets**
> > > |Method|Valid (Avg.)|Unique|||Div.|||RS|||Memb.|||
> > > |-|-|-|-|-|-|-|-|-|-|-|-|-|-|
> > > |FMG-Text (Best)|100\%|100\%|100\%|100\%|0.73|0.46|0.84|33.1\%|87.1\%|98.5\%|99.6\%|100\%|99.6\%|
> > > |FMG (Best)|100\%|100\%|100\%|100\%|0.73|0.46|**0.85**|**61.7\%**|**93.0\%**|**99.1\%**|99.6\%|100\%|**99.8\%**|
> > >
> > > **Results: Real-World Datasets**
> > > |Method|Valid (Avg.)|Unique||Novelty||Div.||RS||Memb.||
> > > |-|-|-|-|-|-|-|-|-|-|-|-|
> > > |FMG-Text (Best)|100\%|100\%|100\%|100\%|91\%|0.92|0.93|69\%|77\%|**43\%**|33\%|
> > > |FMG (Best)|100\%|100\%|100\%|100\%|**92\%**|**0.93**|0.93|**70\%**|**78\%**|38\%|**46\%**|
> > >
> > > For completeness, we replicated Sec. 5.1's study for FMG-Text too. Full results: https://ibb.co/4ZpLMywg
> > >
> > > **Findings.**
> > >  - FMG with text only inputs is a competitive baseline when images cannot be used. Barring FMG, FMG-Text leads in 13/24 columns of Tables 1 & 2.
> > >  - FMG with image inputs still performs best overall—especially on diversity and synthesizability, which are crucial for practical molecule generation.
> > >
> > > **Technical Note: SMILES/SELFIES compatibility.**
> > >
> > > A key modeling consideration is that SMILES/SELFIES are incompatible with our hypergraph formulation, which treats **bonds** (not atoms) as the fundamental units. This ensures that each clique’s atoms (edges) are disjoint, which is essential for our grammar induction and molecule reconstruction.
> > >
> > > Because SMILES/SELFIES encode atoms explicitly but treat bonds implicitly, there’s no straightforward way to annotate bonds or define disjoint cliques. Enforcing compatibility would require taking the bond-atom dual of (1) our grammar formulation or (2) the SMILES/SELFIES syntax. This mismatch, discussed in Sec. 3.1, further motivates our use of images, where we can easily highlight constituent bonds.
> > >
> > > **Analysis.**
> > > We find that text inputs make small motif identification (membership) easier, while image inputs better support global substructure reasoning, which drives higher synthesizability. This matches known strengths of vision-language models and aligns with your intuition that representation format shapes performance.
> > >
> > > We will include these new results in the revised manuscript. In particular, we will incorporate the full FMG-Text results and supporting analyses into the main text, further clarifying how representation format affects performance.
> > >
> > > We appreciate your suggestions, which have meaningfully strengthened the paper. We hope this additional analysis provides the clarity you were looking for—and would be deeply grateful if you would consider championing the work!

---

### Decision · Program_Chairs · 2025-05-01

**Decision:**

Accept (poster)

**Comment:**

This paper proposes an approach to induce interpretable graph grammars by processing image representation of molecules using vision language models. All the reviewers agree that the proposed idea is interesting, with convincing experiments. Some reviewers expressed concerns about interpretability, which was alleviated during the rebuttal.

A reviewer expressed concern on the lack of experiments for the popular molecule generation benchmarks such as MOSES and ZINC. I do not think this is a large problem since this paper specifically targets data-expensive problems which are actually quite important in some applications.

I do have some concerns on applicability of the method to datasets larger than 1k, e.g., how about 10k? It would be meaningful if the authors manage to scale their experiments. Additionally, the authors could consider a stronger baseline such as finetuning an unconditional molecule generator pretrained on a large corpus of molecules.

Overall, I recommend acceptance for this paper.